# Semi-Random Matrix Completion via Flow-Based Adaptive Reweighting

Jonathan A. Kelner[*]     Jerry Li[†]     Allen Liu[‡]     Aaron Sidford[§]     Kevin Tian[¶]

## Abstract

We consider the well-studied problem of completing a rank-$r$, $\mu$-incoherent matrix $\mathbf{M} \in \mathbb{R}^{d \times d}$ from incomplete observations. We focus on this problem in the *semi-random setting* where each entry is independently revealed with probability at least $p = \frac{\text{poly}(r,\mu,\log d)}{d}$. Whereas multiple nearly-linear time algorithms have been established in the more specialized *fully-random setting* where each entry is revealed with probablity exactly $p$, the only known nearly-linear time algorithm in the semi-random setting is due to [17], whose sample complexity has a polynomial dependence on the inverse accuracy and condition number and thus cannot achieve high-accuracy recovery. Our main result is the first high-accuracy nearly-linear time algorithm for solving semi-random matrix completion, and an extension to the noisy observation setting. Our result builds upon the recent short-flat decomposition framework of [42, 43] and leverages fast algorithms for flow problems on graphs to solve adaptive reweighting subproblems efficiently.

## 1 Introduction

How can we ensure learning algorithms do not overfit to generative assumptions on their input data? Since worst-case statistical inference problems are often intractable (i.e., without distributional assumptions), to develop efficient algorithms, we often introduce a generative model as a proxy for average-case, real-world behavior. However, if these algorithms are to be useful practically, we must ensure they do not depend too strongly on any artificial properties of the generative model. Ideally, they should require only what is statistically and algorithmically necessary, and no more.

In this paper, we consider this question in the context of matrix completion, one of the most fundamental and well-studied linear inverse problems. In matrix completion, the goal is to recover a symmetric rank-$r$ matrix $\mathbf{M} \in \mathbb{R}^{d \times d}$ where $r \ll d$,[6] given a small (typically sublinear) number of entries revealed independently and at random. In a seminal line of work [13, 11, 12, 61], it was shown that under natural structural assumptions on $\mathbf{M}$, one can fully recover $\mathbf{M}$ given $\approx dr$ randomly revealed entries in polynomial time, via semidefinite programming (SDP). Subsequently, there has been extensive work attempting to improve upon statistical and computational guarantees for matrix completion. In an oversimplification of the literature (see Section 1.3 for a more detailed and nuanced discussion), this has resulted in two categories of matrix completion algorithms.

- **SDP-based algorithms** achieve the nearly-optimal sample complexity, but are based on solving SDPs, which requires time $\Omega(d^{3.5})$ even with state of the art solvers [39, 34].

- **First-order iterative methods** require a slightly higher sample complexity, namely, they require $m \approx pd^2$ revealed entries for an observation probability $p = \Omega(\text{poly}(r, \log d)) \cdot \frac{1}{d}$. However,

---

[*]MIT, `kelner@mit.edu`.

[†]Microsoft Research, `jerrl@microsoft.com`.

[‡]MIT, `cliu568@mit.edu`.

[§]Stanford University, `sidford@stanford.edu`.

[¶]University of Texas at Austin, `kjtian@cs.utexas.edu`.

[6]Our results also hold for asymmetric, rectangular matrices. See the discussion at the end of Appendix A.

38th Conference on Neural Information Processing Systems (NeurIPS 2024).

they use faster optimization primitives and can be implemented in time $\widetilde{O}(m \cdot \mathrm{poly}(r, \log d))$. Throughout the introduction, we will somewhat informally refer to such algorithms as *fast*.[7]

Because matrix completion is particularly interesting to study when the target rank $r$ is substantially smaller than the dimension $d$ (as information-theoretically, we require $\gtrsim dr$ observations), it may appear that fast algorithms are able to nearly match the statistical guarantees of the SDP-based methods, while offering substantially lower runtimes. In light of this, naïvely one might expect that first-order iterative methods ought to be always preferred over the alternative. However, it has been noted that in many real-world situations, these first-order iterative methods actually fail to perform reliably, see e.g. [66, 70, 26, 68]. An oft-cited reason for this failure is exactly the tendency for these sorts of methods to demonstrate the type of generative overfitting described above [66, 70, 4]. Indeed, in real-world applications, the distribution of revealed entries is usually very far from uniformly random [4], which can cause iterative methods that rely too heavily on this assumption to fail dramatically. This begs the following natural question:

*Can we design fast algorithms for matrix completion that are robust to generative overfitting?*

In light of this discussion, we consider a *semi-random* variant of matrix completion, introduced in [17]. Recall that in standard matrix completion, we assume every entry $(i, j) \in [d] \times [d]$ is revealed independently with probability $p$. In the *p-semi-random model* (Definition 2), we only know $p \in (0, 1)$ such that every entry is revealed independently with an *unknown* probability $p_{ij} \geq p$. Given that real-world data is unlikely to be generated exactly uniformly, it is reasonable to believe that the semi-random model is a more faithful model of matrix completion in practice.

In this model, the algorithm is given strictly more information than in the standard (uniform) matrix completion setup. This is because the semi-random model is equivalent to the setting where we are first given a standard matrix completion instance, and then additional (but potentially adversarially chosen) entries are revealed.[8] It is straightforward to show that the SDP-based approaches generalize to succeed in the semi-random setting, since the adversary's revealed entries just translate to additional constraints in the convex program satisfied by the ground truth. However, as demonstrated in [17], Appendix A, previously known first-order iterative methods can fail to converge in this semi-random setting. We view this as strong theoretical evidence that such algorithms exhibit large amounts of generative overfitting. We can now more concretely rephrase the previous question:

*Can we design fast algorithms for semi-random matrix completion?*

This paper focuses on the theoretical complexity of the problem; the relationship of the techniques in this work to efficient matrix completion in practice is an interesting direction for future research. Precisely, we seek to design an algorithm for semi-random matrix completion which works for the same (or qualitatively similar) $p$ as in the standard setting, and which runs in time $\widetilde{O}(m \cdot \mathrm{poly}(r, \log d))$? Here $m$ denotes the total number of revealed entries, a proxy for the input size; note that in the semi-random setting, it could be that $m \gg pd^2$.

The main result of [17] is an efficient algorithm for semi-random matrix completion, albeit with some important caveats, as we discuss shortly. Roughly speaking, for $\epsilon \in (0, 1)$, [17] gave an algorithm which, given observations of rank-$r$ $\mathbf{M}$ with condition number $\kappa$ from the $p$-semi-random model for $p = \frac{1}{d} \cdot \widetilde{O}(\mathrm{poly}(r, \kappa, \frac{1}{\epsilon}))$, outputs $\mathbf{M}'$ so that $\|\mathbf{M}' - \mathbf{M}\|_{\mathrm{F}}^2 \leq \epsilon \|\mathbf{M}\|_{\mathrm{F}}^2$ with high probability. Moreover, it runs in nearly-linear time $\widetilde{O}(m \cdot \mathrm{poly}(r, \kappa, \frac{1}{\epsilon}))$, where $m$ is the number of revealed entries. While this is a promising first step towards matrix completion in the semi-random model, it leaves several important questions unanswered.

- **High-accuracy recovery.** Both the sample complexity and runtime of the algorithm in [17] depend polynomially on $\epsilon^{-1}$. In particular, the algorithm cannot recover the matrix $\mathbf{M}$ to $\epsilon^{-1} = \mathrm{poly}(d)$ levels of precision in nearly-linear time. Thus, even in the standard setting where we assume that the bit complexity of the entries of $\mathbf{M}$ are polynomially bounded, [17] cannot achieve exact

---

[7]Here, and throughout the paper, we use the notation $\widetilde{O}(f) := O(f \log^{O(1)} f)$ and similarly define $\widetilde{\Omega}$.

[8]As clarified in Definition 2, it is important in this equivalence that the additionally revealed entries are not adaptively chosen, i.e., do not depend on which random entries were observed. This manifests in our analysis because an adaptive adversary causes significant difficulties with independent sample splitting (see Lemma 2). There is precedent from the literature on fully-random matrix completion that the inability to sample split causes analysis issues (see, e.g., Section 3 of [61]); extending to an adaptive adversary is a challenging open direction.

recovery of the matrix in nearly-linear time. Unfortunately, this dependence seems inherent to the techniques therein (see the discussion in Section 1.2).

- **Condition number dependence.** Another important downside of the [17] algorithm's guarantee is its polynomial dependences on $\kappa$ in both the sample complexity and runtime. No dependence on $\kappa$ is required information-theoretically, or by the SDP-based methods. This aspect also seems somewhat inherent to the non-convex iterative method used in [17].

- **Noise tolerance.** Finally, the guarantees in [17] require that $\mathbf{M}$ is exactly low-rank, i.e. it does not generalize to handle noise in observations. In practice, this is unlikely, and guarantees should ideally hold even when we only assume that the observed $\mathbf{M}$ is close to low-rank.

## 1.1 Our results

Our main result is a new fast algorithm for semi-random matrix completion that addresses all three of the aforementioned issues. Specifically, we show the following main result.

**Theorem 1** (informal, see Theorem 2). *Let $\mathbf{M}^\star \in \mathbb{R}^{d \times d}$ be a symmetric, rank-$r$, $\mu$-incoherent matrix, let symmetric $\mathbf{N} \in \mathbb{R}^{d \times d}$ satisfy $\|\mathbf{N}\|_\infty \leq \Delta$, and let $\widehat{\mathbf{M}} = \mathbf{M}^\star + \mathbf{N}$. Let $\epsilon \in (0, 1)$. There is an algorithm that takes observations from $\widehat{\mathbf{M}}$ in the $p$-semi-random model for $p = \frac{Q}{d}$, where $Q = \mathrm{poly}(r, \mu, \log(\frac{d}{\epsilon}))$, and outputs $\mathbf{M}$ as a rank-$Q$ factorization such that with high probability,*

$$\|\mathbf{M} - \mathbf{M}^\star\|_\infty \leq Q \left( \Delta + \epsilon \|\mathbf{M}^\star\|_\infty \right).$$

*The algorithm runs in nearly-linear time $O(mQ)$ where $m$ is the number of revealed entries.*

Here, $\|\cdot\|_\infty$ refers to the elementwise $\ell_\infty$ norm of a matrix, and $\|\cdot\|_F$ refers to the Frobenius norm. We pause to comment on Theorem 2. First, in the noiseless setting, i.e., when $\Delta = 0$, our algorithm has a polylogarithmic dependence on $\epsilon^{-1}$, and thus achieves high-accuracy recovery in nearly-linear time. In particular, as long as the bit complexity of the underlying matrix is polynomially bounded, we achieve exact recovery in nearly-linear time. Moreover, our rate has no dependence on the condition number $\kappa$. In both of these aspects, our result qualitatively improves upon [17], while retaining a nearly-linear runtime and $\widetilde{O}(d \cdot \mathrm{poly}(r))$ sample complexity. We also remark that the incoherence assumption in Theorem 2 (or a less standard variant thereof), which is formally defined in Definition 1, is used in all low-rank matrix completion algorithms we are aware of, as the problem is intractable without subspace regularity assumptions (see, e.g., discussion in [43]).

Finally, we observe that our guarantee applies to the noisy setting, i.e., $\Delta > 0$. To our knowledge, this is the first such guarantee in the literature for semi-random matrix completion, even for SDP-based methods. We achieve guarantees in terms of the $\ell_\infty$ norm of the noise matrix $\mathbf{N}$; moreover, the overhead is a polynomial of only $r \log(\frac{d}{\epsilon})$, a preferable guarantee to the $\mathrm{poly}(d)$ overheads known to be achievable in the standard, fully random setting by semidefinite programming [11].

We do note that in the standard matrix completion setting, error guarantees are typically phrased in terms of the Frobenius norm of the noise $\mathbf{N}$ [11, 43]. However, in the semi-random setting, some measure of the "element-wise" noise level is unavoidable, because the semi-random noise could be chosen to only reveal unusually large entries of the noise matrix. We believe it is an interesting open question whether or not one can achieve somewhat more global measures of error such as the Frobenius norm of large entries in the revealed noise, which are more comparable to guarantees recently attained for semi-random sparse linear regression [42].

## 1.2 Our techniques

We achieve these results via a fundamentally different approach than the prior work of [17], detailed in Section 2. At a (very) high level, [17] relies on finding a single reweighting of the revealed entries of $\mathbf{M}$ that is good in a global sense. This roughly corresponds to finding a mask of the revealed entries, so that the masked matrix is guaranteed to be a good spectral approximation of the ground truth matrix. Given such a mask, [17] then shows that a non-convex optimization method will recover the true matrix, with high probability. While this is a very natural approach, this also directly results in a $\mathrm{poly}(\frac{1}{\epsilon})$ dependence in the runtime and sample complexity of the algorithm. This is because the quality of the spectral approximation directly goes into the final accuracy of their overall algorithm; ultimately, they need a $(1 + \epsilon)$-multiplicative spectral approximation to the ground truth. However, all known techniques for finding such a high-quality spectral approximation require a $\mathrm{poly}(\frac{1}{\epsilon})$ dependence in both sample complexity and runtime.

We instead devise an iterative method that is guaranteed to make progress, assuming that the iterate satisfies a local progress criterion. We then find a local reweighting of the matrix which allows us to find a step which satisfies this criterion. While this local progress criterion is much more involved to state, and this reweighting must now be done every iteration (rather than just once at the beginning of the algorithm), this framework has an important advantage over the global criterion used previously. Critically, now each step of our method need only seek a reweighting which achieves a constant factor of relative progress to the target matrix. Thus, rather than needing a $(1 + \epsilon)$-multiplicative approximation at every step, we only need a constant multiplicative approximation to make progress. This is the main insight which allows us to achieve polylogarithmic rates in $\frac{1}{\epsilon}$.

Our algorithm builds upon the short-flat decomposition-based approach to designing iterative methods for linear inverse problems introduced in [42], and specialized in [43] to matrix completion in the standard setting. However, there are significant technical challenges in adapting this framework to semi-random models. At a high level, the matrix completion problem is not well-conditioned, so a crucial step in the [43] algorithm is dropping rows and columns of a difference matrix (between an iterate and the target) which are estimated as too large. Random observations (held out via sample splitting) are then used in matrix concentration inequalities to estimate the difference matrix on the remaining indices, and dropped indices are later recovered.

Unfortunately, all of these steps crucially rely on independent sampling and hence break in the presence of semi-random noise. We instead take a fresh look at what certificates of progress are possible in the semi-random setting. Our key technical contribution, Algorithm 2, is a subroutine which makes bicriteria progress: it either finds an adaptive reweighting which accurately estimates the difference on a large submatrix after denoising, or certifies that a few rows and columns are responsible for much of the difference and hence can be dropped. To achieve our nearly-linear runtime, we crucially exploit graphical structure of feasible reweightings present in the randomly-revealed entries, and use flow-based optimization algorithms to solve our reweighting subproblem.

## 1.3 Related work

Here, we review related work on matrix completion and semi-random models. These lines of research are vast and we focus on the research most relevant to our setting. For a more comprehensive review, we refer the interested reader to surveys on these topics, e.g., [41, 60, 59, 63].

**Matrix completion.** Matrix completion was first introduced in [62] in the context of collaborative filtering, e.g., for the Netflix Prize [6]. Since then, it has found applications in various areas, such as signal processing [46, 64, 57], social network analysis [47], causal inference [4], and most recently, AI alignment [1]. In a seminal line of work, [13, 11, 61, 12, 16] demonstrated that SDPs based on nuclear norm minimization can solve matrix completion with sublinear sample complexity, and in polynomial time. Another line of work focuses on first-order iterative methods for matrix completion, see e.g., [44, 31, 32, 33, 36, 74, 67, 18, 72, 30]. These works demonstrate that optimization frameworks, such as alternating minimization, can provably and efficiently recover the true matrix. Finally, yet another line of work seeks to understand the non-convex landscape of descent-based methods for matrix completion [65, 21, 28, 40, 73, 40, 73, 71]. In terms of theoretical runtimes, the state of the art $\widetilde{O}(dr^{3+o(1)})$, is given in [43]. While some of the aforementioned papers have implementations, comprehensively evaluating the practical performance of all of these methods is an interesting direction for future work. As noted in [17], none of these fast methods are known to succeed in the presence of semi-random noise.

**Semi-random models.** Semi-random models were first introduced in a pair of seminal papers [9, 24], in part as a means to test whether learning algorithms which succeed under random models generalize to more realistic generative modeling assumptions. Most early work on semi-random models focused on understanding semi-random variants of various constraint satisfaction problems, see e.g. [25, 24, 45, 15, 48, 50]. More recently, they have also been studied in the context of learning theory, in the context of clustering problems [23, 53, 49, 14, 29, 51, 52, 55], sparse and overcomplete linear regression [37, 42], planted clique [54, 10], and more [27, 8].

Of particular interest to us is the recent line of work on developing fast learning algorithms in semi-random models [17, 42, 43, 27, 8]. The closest and most direct comparison to our work is the aforementioned [17], which we quantitatively improve upon in a number of ways. It is worth noting that the specific $\text{poly}(r) \approx r^6$ dependence in the sample complexity of [17] is lower than in Theorem 1. We view our result as a promising proof-of-concept of the tractability of semi-random

matrix completion via fast algorithms; both our dependence and the dependence of [17] are somewhat large at the moment, limiting immediate practical deployment, but we believe that it is an interest direction for future research to tighten dependencies on problem parameters and seek more practical algorithms.

## 2 Technical overview

At a high level, our algorithm follows the paradigm in [42] for designing fast iterative algorithms that are robust to semi-random models, which used this framework for sparse linear regression. Broadly, the approach is to first find deterministic conditions which guarantee that a step of the iterative method, which descends along a reweighted direction based on semi-random observations, makes progress. We then develop a custom nearly-linear time algorithm for computing such a reweighting, using optimization tools specialized to the regularity of graph-structured polytopes.

**Short-flat decompositions for progress.** We first discuss the verifiable conditions we impose to certify progress of our iterative method. Throughout this section, we let our current iterate be $\mathbf{M} \in \mathbb{R}^{d \times d}$ and let the ground truth be $\mathbf{M}^\star \in \mathbb{R}^{d \times d}$, assume both matrices have rank at most $r$, and focus on the noiseless setting $\mathbf{N} = \mathbf{0}_{d \times d}$ for simplicity. We also assume for normalization purposes that $\|\mathbf{M} - \mathbf{M}^\star\|_{\mathrm{F}} = 1$, and our goal is to take a step $\mathbf{M}' \leftarrow \mathbf{M} + \mathbf{D}$ so that

$$\|(\mathbf{M} + \mathbf{D}) - \mathbf{M}^\star\|_{\mathrm{F}} \le \frac{1}{2}. \tag{1}$$

Our conditions are inspired by the approach in [43], which gives improved iterative algorithms for standard matrix completion (without semi-random noise). Let $p$ be the minimum observation probability, let $\Omega$ be the set of observed entries from $\mathbf{M}^\star$, and let $\Omega^\star$ be the set of truly random observations, i.e., when each entry is revealed with probability exactly $p$ (we can find a coupling so that $\Omega^\star \subseteq \Omega$ always). Inspired by a similar framework in the sparse recovery setting [42], the key certificate for guaranteeing progress of the form (1) in [43] is that of a short-flat decomposition. We say a candidate step $\mathbf{D}$ has a short-flat decomposition if we can write $\mathbf{D} = \mathbf{S} + \mathbf{F}$ where $\|\mathbf{S}\|_{\mathrm{F}} \le 1$ and $\|\mathbf{F}\|_{\mathrm{op}} \le cr^{-\frac{1}{2}}$ for a small constant $c$: here we think of $\mathbf{S}$ as the "short" part and $\mathbf{F}$ as the "flat" part of the decomposition. If we can further ensure that enough signal is captured, i.e., $\langle \mathbf{D}, \mathbf{M}^\star - \mathbf{M} \rangle \ge 1 - c$, then [42, 43] give a short argument based on Hölder's inequality that a step in $\mathbf{D}$, combined with a low-rank projection for denoising, will imply (1).

Thus, it suffices to find a matrix $\mathbf{D}$ that has these properties; for intuition, we would ideally like to take $\mathbf{D} = \mathbf{S} = \mathbf{M}^\star - \mathbf{M}$ and $\mathbf{F} = \mathbf{0}_{d \times d}$, which would complete the matrix in one shot. Naturally, however, we are limited by the fact that $\mathbf{D}$ should be supported on the observations $\Omega$, as otherwise we cannot make any guarantees on $\langle \mathbf{D}, \mathbf{M}^\star - \mathbf{M} \rangle$. Naïvely, one would hope to simply take the empirical observations $\mathbf{D} = \frac{1}{p}[\mathbf{M} - \mathbf{M}^\star]_{\Omega^\star}$ (which zeroes entries outside $\Omega^\star$), an unbiased estimator of $\mathbf{M}^\star - \mathbf{M}$, and appeal to matrix concentration arguments to show existence of a short-flat decomposition. Unfortunately, this is not the case, as entries, rows, or columns of $\mathbf{M}^\star - \mathbf{M}$ which are too large can arbitrarily hinder matrix concentration bounds, and [43] gives simple examples showing this dependence on the imbalance is unavoidable for uniform reweightings.

However, [43] made the important insight that when matrix concentration fails, it must be that $\mathbf{M} - \mathbf{M}^\star$ is overly concentrated on a few heavy rows and columns, and hence dropping a few rows and columns significantly reduces $\|\mathbf{M} - \mathbf{M}^\star\|_{\mathrm{F}}$. These rows and columns can later be recovered using the low-rank assumption, once enough progress has been made. The algorithm in [43] then works by repeatedly estimating heavy rows and columns, explicitly dropping them to form a set $S \subseteq [d]$ of balanced indices, and then taking the step $\mathbf{D} = \frac{1}{p}[\mathbf{M} - \mathbf{M}^\star]_{\Omega^\star \cap (S \times S)}$. This necessity of dropping certain poorly-behaved indices is a crucial difference between the matrix completion setting and simpler semi-random inverse problems such as sparse linear regression [42], which possess stronger global regularity features such as the restricted isometry property (RIP).

**Certifying progress in semi-random models.** In the semi-random matrix completion problem, we need several modifications to the strategies of [42, 43], to handle the interplay between estimating heavy indices and making certifiable progress. Firstly, we cannot accurately estimate the $\ell_2$ norm of rows and columns of the true difference matrix $\mathbf{M} - \mathbf{M}^\star$ from observations, because the semi-random adversary can drastically distort our estimates. However, we show in Algorithm 1 that we can use the $\ell_\infty$ norm of observed entries as a proxy for heaviness.[9] Specifically, after dropping rows and columns

---

[9]This norm distortion is the main source of $\mathrm{poly}(r)$ losses in our guarantees compared to the [43] algorithm.

with the most observed large entries, we argue that all remaining rows and columns are well-balanced except on few sparse errors, which we exclude from our potential.

Next, even if appropriate rows and columns were dropped, we cannot use the empirical observations to construct our step, as semi-random observations are no longer unbiased and the fully-random indices $\Omega^\star$ are unknown. Instead, we use $\Omega^\star$ existentially to set up a convex program over reweightings $w \in \mathbb{R}^\Omega$, to search for a candidate step $\mathbf{D} := \sum_{(i,j) \in \Omega} w_{ij}[\mathbf{M} - \mathbf{M}^\star]_{ij}$. Roughly, our convex program aims to find $w$ satisfying the following constraints (see Lemma 7 for a formal statement).

1. For all $i \in [d]$, $\sum_{j \in [d] | (i,j) \in \Omega} w_{ij}[\mathbf{M} - \mathbf{M}^\star]_{ij}^2 = O(\frac{1}{d})$.

2. There exists $\mathbf{S}, \mathbf{F} \in \mathbb{R}^{d \times d}$ with $\mathbf{D} = \mathbf{S} + \mathbf{F}$, $\|\mathbf{S}\|_{\mathrm{F}} \leq 1$, and $\|\mathbf{F}\|_{\mathrm{op}} = O(r^{-\frac{1}{2}})$.

3. For a sufficiently small constant $c$, $\langle \mathbf{D}, \mathbf{M} - \mathbf{M}^\star \rangle = \sum_{(i,j) \in \Omega} w_{ij}[\mathbf{M} - \mathbf{M}^\star]_{ij}^2 \geq 1 - c$.

Item 1 serves to ensure our step $\mathbf{D}$ is sufficiently spread out amongst its rows and columns, which serves two roles: it bounds how much progress is lost when excluding indices, and enforces crucial problem regularity for our reweighting optimization algorithm. On the other hand, Items 2 and 3 are the standard criteria for making iterative progress through the short-flat framework of [42, 43]. It is straightforward to see that the ideal candidate $\frac{1}{p}[\mathbf{M} - \mathbf{M}^\star]_{\Omega^\star \cap (S \times S)}$ we mentioned earlier is feasible for these constraints, and we show in Lemma 7 that finding any satisfying reweighting is enough to ensure progress (with some mild technical changes). It remains to discuss how to find a candidate $\mathbf{D}$ to ensure rapid convergence through the progress bound (1).

**Flow-based adaptive reweighting.** A key technical contribution of this work is a nearly-linear time algorithm finding a reweighting $\mathcal{D}(w) := \sum_{(i,j) \in \Omega} w_{ij}[\mathbf{M} - \mathbf{M}^\star]_{ij}$ that meets our progress criteria. Due to our focus on fast algorithms, we require this step to run in $O(|\Omega| \cdot \mathrm{poly}(r))$ time. Our strategy is to form a joint objective over $w$ obeying Item 1, and $\|\mathbf{S}\|_{\mathrm{F}} \leq 1$, of the form

$$F_{\text{prog-flat}}(w, \mathbf{S}) := -\langle \mathcal{D}(w), \mathbf{M} - \mathbf{M}^\star \rangle + C \cdot \mathrm{smax}(\mathcal{D}(w) - \mathbf{S}). \qquad (2)$$

Here, the parameter $C$ is used to trade off the progress objective $\langle \mathcal{D}(w), \mathbf{M} - \mathbf{M}^\star \rangle$ (giving Item 3) and the flatness objective $\mathrm{smax}(\mathcal{D}(w) - \mathbf{S})$ (giving Item 2), where $\mathrm{smax}(\mathbf{M}) = \log \mathrm{Tr} \exp(\mathbf{M})$ is a smooth approximation to the operator norm. That is, letting $\mathcal{Z} := \mathcal{W} \times \mathcal{S}$ be the product space between $w \in \mathbb{R}^\Omega$ satisfying Item 1 and short matrices $\mathbf{S} \in \mathbb{R}^{d \times d}$, we show that finding $(w, \mathbf{S}) \in \mathcal{Z}$ with low suboptimality error in $F_{\text{prog-flat}}$ is enough to make progress through Lemma 7.

We optimize (2) by carefully applying a Frank-Wolfe method due to [35]. While naïve Frank-Wolfe analysis requires global smoothness of the objective, this provably does not hold for our $F_{\text{prog-flat}}$ (at least, with a smoothness bound of $\approx \mathrm{poly}(r)$ required for nearly-linear time algorithms). However, we leverage that the analysis in [35] shows that weaker regularity conditions, namely bounds on the curvature constant, suffice for efficient convergence of an iterative method. Specifically, we use the structure of our constraint set $\mathcal{W}$ to show that $F_{\text{prog-flat}}$ is smooth *in a restricted set of directions within* $\mathcal{W}$ which implies suitable bounds on the curvature constant (see, e.g., Lemma 18) to remove a $d$ factor from the naïve smoothness bound. We also observe that $\mathcal{W}$ is exactly a rescaling of a bipartite matching polytope, so that we can use flow-based graph algorithms, e.g., the approximate weighted matching algorithm of [22], to efficiently implement the linear optimization oracles over $\mathcal{W}$ required by Frank-Wolfe methods. We then use techniques from numerical linear algebra and polynomial approximation to implement linear optimization oracles over $\mathcal{S}$, completing our optimization subroutine for finding adaptive reweightings.

One other technical hurdle arises in implementing this framework for finding a reweighting: the existence of $w^\star$ satisfying Items 1, 2, and 3 relies on having a good estimate of the current distance $\|\mathbf{M} - \mathbf{M}^\star\|_{\mathrm{F}}$, after excluding certain rows and columns. In general, we can only upper bound this quantity, but we show that we can also use our optimization subroutine for (2) to certify non-existence of such a $w^\star$. In this latter case, we show that it must have been the case that a few rows and columns were significantly heavier than the rest, and we can run Algorithm 1 to drop them. Our overall iterative method, Algorithm 2, makes bicriteria progress and either terminates with a "drop" step (when no feasible $w^\star$ exists), or a "descent" step (when we find a good solution to (2)).

**Recovering dropped rows and columns.** Finally, we briefly discuss how to post-process an estimate $\mathbf{M}$ that is close to $\mathbf{M}^\star$ on a large submatrix to recover dropped rows and columns. The high-level outline is similar to [43], where we aim to find a "verified" set of rows and columns that

we completed well, and use the low-rank assumption to obtain accurate estimates to the remainder via regression. However, there are several key differences. The main difference is that in the semi-random model, we need to track errors in our estimate entrywise instead of in $\ell_2$, since a semi-randomness could drastically distort the empirical $\ell_2$ error by adversarially revealing entries. We rely on a structural result using tools from the theory on volumetric spanners [5, 7] (see Lemma 27) showing that large entrywise errors must be localized to a small submatrix. At this point, we set up $\ell_\infty$ regression problems to identify verified indices, solvable in nearly-linear time using recent advances in linear programming [69]. Importantly, the overhead in the error guarantee of this recovery step is only $\widetilde{O}(\mathrm{poly}(r))$ (see Proposition 3), allowing us to achieve a recovery guarantee in Theorem 1 that avoids $\mathrm{poly}(d)$ overheads in the presence of noise.

## 3  Preliminaries

We now formally set up preliminaries required for analyzing our solution to the semi-random matrix completion problem. First, we recall the standard definition of incoherence for subspaces.

**Definition 1** (Incoherence). *We say a dimension-$r$ subspace $V \subseteq \mathbb{R}^d$ is $\mu$-incoherent if $\|\mathbf{\Pi}_V e_i\|_2 \le \sqrt{\frac{\mu r}{d}}$ for all $i \in [d]$ (where $e_i$ is the $i$th standard basis vector), and $\mathbf{M} \in \mathbb{R}^{m \times n}$ is $\mu$-incoherent if $\mathbf{M}$ has $\mu$-incoherent row and column spans.*

We will work with symmetric matrices. We denote the space of $d \times d$ symmetric matrices by $\mathbb{S}^{d \times d}$. In Appendix A, we give a reduction from general matrices to symmetric matrices showing that it suffices to work in the symmetric case, up to constant factor loss in parameters. Next, we define the semi-random observation model.

**Definition 2** (Semi-random model). *For $\mathbf{M} \in \mathbb{S}^{d \times d}$, we use $\mathcal{O}_p^{\mathrm{sr}}(\mathbf{M})$ to denote oracle access to $\mathbf{M}$ in the $p$-semi-random model, where we call $\mathcal{O}_p^{\mathrm{sr}}$ a $p$-semi-random oracle. Specifically, in one call to $\mathcal{O}_p^{\mathrm{sr}}$ with input $\mathbf{M}$, for* unknown *observation probabilities $\{p_\iota\}_{\iota \in [d] \times [d]} \subseteq [p, 1]$, each $\iota \in [d] \times [d]$ is independently included in a set $\mathcal{I}$ with probability $p_\iota$. The oracle $\mathcal{O}_p^{\mathrm{sr}}$ then returns the set $\{[\mathbf{M}]_\iota\}_{\iota \in \mathcal{I}}$. When an algorithm requires the ability to query $\mathcal{O}_p^{\mathrm{sr}}(\mathbf{M})$ for various values of $p$ (specified in the algorithm description), we list the input as $\mathcal{O}_{[0,1]}^{\mathrm{sr}}(\mathbf{M})$.*

Note that $\mathcal{O}_p^{\mathrm{sr}}(\mathbf{M})$ and $\mathcal{O}_{[0,1]}^{\mathrm{sr}}(\mathbf{M})$ inherit their definitions in [43] when all $p_\iota = p$, for a fixed $p \in [0, 1]$. Now we can formally define the semi-random matrix completion problem we study.

**Definition 3** (Semi-random matrix completion). *In the* semi-random matrix completion *problem, parameterized by $\mu, \Delta \in \mathbb{R}_{\ge 0}$, $p \in [0, 1]$, $d \in \mathbb{N}$, and $r^\star \in [d]$, there is an unknown rank-$r^\star$ $\mu$-incoherent matrix $\mathbf{M}^\star \in \mathbb{S}^{d \times d}$. For some $\mathbf{N} \in \mathbb{S}^{d \times d}$ satisfying $\|\mathbf{N}\|_\infty \le \Delta$, and $\mathbf{M} := \mathbf{M}^\star + \mathbf{N}$, we receive the output of one call to $\mathcal{O}_p^{\mathrm{sr}}(\mathbf{M})$, and wish to output an approximation to $\mathbf{M}^\star$.*

**Remark 1.** *In Appendix A, we give a sample-splitting reduction which lets us use $\mathcal{O}_p^{\mathrm{sr}}(\mathbf{M})$ to simulate $\approx \frac{p}{q}$ independent accesses to $\mathcal{O}_q^{\mathrm{sr}}(\mathbf{M})$ for smaller values $q \le p$.*

We will also need a few technical definitions to explain the main ingredients in our algorithm.

**Entropy and softmax.** We let $\Delta^{d \times d} := \{\mathbf{M} \in \mathbb{S}_{\succeq \mathbf{0}}^{d \times d} \mid \mathrm{Tr}(\mathbf{M}) = 1\}$ denote the $d \times d$ spectraplex. We let $H : \Delta^{d \times d} \to \mathbb{R}$ denote von Neumann (matrix) entropy, i.e., $H(\mathbf{M}) := \langle \mathbf{M}, \log \mathbf{M} \rangle$, and let $H^* : \mathbb{S}^{d \times d} \to \mathbb{R}$ denote its convex conjugate, which we call matrix softmax, defined by $H^*(\mathbf{M}) := \log \mathrm{Tr} \exp \mathbf{M}$. More generally, for $\eta > 0$, we define

$$H_\eta^*(\mathbf{M}) := \eta \log \mathrm{Tr} \exp \left( \frac{1}{\eta} \mathbf{M} \right). \tag{3}$$

We recall the following standard facts about $H_\eta^*$.

**Fact 1** ([58]). *The function $H_\eta^*$ defined in (3) satisfies the following for all $\eta > 0$.*

1. *For all $\mathbf{M} \in \mathbb{S}^{d \times d}$, $\lambda_1(\mathbf{M}) \le H_\eta^*(\mathbf{M}) \le \lambda_1(\mathbf{M}) + \eta \log d$, where $\lambda_1(\mathbf{M})$ is the maximum eigenvalue of $\mathbf{M}$.*

2. *For all $\mathbf{M} \in \mathbb{S}^{d \times d}$, $\nabla H_\eta^*(\mathbf{M}) = \frac{\exp(\frac{1}{\eta}\mathbf{M})}{\mathrm{Tr}\exp(\frac{1}{\eta}\mathbf{M})}$.*

3. *$H_\eta^*$ is twice-differentiable and $\frac{1}{\eta}$-smooth with respect to the norm $\|\cdot\|_{\mathrm{op}}$.*

To use $H_\eta^*$ to enforce operator norm bounds, we define the signed lift of a matrix $\mathbf{M} \in \mathbb{S}^{d \times d}$ by

$$\text{slift}(\mathbf{M}) := \begin{pmatrix} \mathbf{M} & \mathbf{0}_{d \times d} \\ \mathbf{0}_{d \times d} & -\mathbf{M} \end{pmatrix} \in \mathbb{S}^{2d \times 2d}.$$

Note that slift signs and lifts $\mathbf{M}$ so that its maximum eigenvalue is its operator norm.

**Comparing matrices.** We use the following comparison definition from [43].

**Definition 4** (Closeness on a submatrix)**.** *We say* $\mathbf{M}, \mathbf{M}' \in \mathbb{R}^{m \times n}$ *are* $\Delta$-*close on a* $\gamma$-*submatrix if there exist subsets* $A \subseteq [m]$, $B \subseteq [n]$ *satisfying* $|A| \geq m - \gamma \min(m,n)$, $|B| \geq n - \gamma \min(m,n)$, *and*

$$\left\| [\mathbf{M} - \mathbf{M}']_{A \times B} \right\|_{\mathrm{F}} \leq \Delta.$$

*If* $\gamma = 0$*, we say* $\mathbf{M}, \mathbf{M}'$ *are* $\Delta$-*close. When* $\mathbf{M}, \mathbf{M}'$ *are both symmetric, we require that* $A = B$.

We note that in Definition 4, $A$, $B$ are unknown; our analysis only uses this definition existentially.

# 4 Outline of proof of Theorem 1

In this section, we overview the main components of the proof of Theorem 1, which are developed and proved in full in the appendix sections. Our final matrix completion algorithm, Algorithm 8, alternates between two main subroutines, Algorithm 3 (analyzed in the following Corollary 1) and Algorithm 7 (analyzed in the following Proposition 3). We prove Theorem 2, a formal version of Theorem 1, by combining Corollary 1 and Proposition 3 in Appendix E. We now explain and motivate each of these pieces in the context of our technical overview in Section 2.

The first main component of our final algorithm, Algorithm 8, takes as input parameters $\alpha$ and $\gamma_{\text{tot}}$, as well as an input matrix $\mathbf{M} \in \mathbb{S}^{d \times d}$ satisfying $\|\mathbf{M} - \mathbf{M}^\star\|_{\mathrm{F}} \leq \Delta$ (i.e., $\mathbf{M}$ and the target matrix $\mathbf{M}^\star$ are $\Delta$-close), where $\Delta$ is sufficiently larger than the noise level $\|\mathbf{N}\|_\infty$. Its goal is to use semi-random observations from $\widehat{\mathbf{M}} = \mathbf{M}^\star + \mathbf{N}$ to return another matrix $\mathbf{M}' \in \mathbb{S}^{d \times d}$ such that $\widehat{\mathbf{M}}$ and $\mathbf{M}$ are $\frac{\Delta}{\alpha}$-close on a $\gamma_{\text{tot}}$-submatrix (Definition 4). In other words, we are willing to give up on a $\gamma_{\text{tot}}$ fraction of rows and columns to make an $\alpha$ factor progress on the remaining submatrix. To achieve this result, we first provide a helper claim in Proposition 1, an analysis of Algorithm 2 (DropOrDescent).

**Proposition 1.** *Let* $\mathbf{M} \in \mathbb{S}^{d \times d}$ *be given as a rank-$r$ factorization, let* $\mathbf{M}^\star \in \mathbb{S}^{d \times d}$ *be rank-$r^\star$, and let* $U \subseteq [d]$*. Suppose we know for* $\gamma_{\text{drop}}, \gamma_{\text{sub}} \in [0,1]$, $\Delta > 0$*, that* $|[d] \setminus U| \leq \gamma_{\text{drop}} d$*, and* $\mathbf{M}_{U \times U}, \mathbf{M}^\star_{U \times U}$ *are* $\Delta$-*close on a* $\gamma_{\text{sub}}$-*submatrix. Let* $\delta, \epsilon \in (0, \frac{1}{10})$, $\theta, \kappa \geq 1$*, and suppose for* $\hat{r} := r + r^\star$ *and an appropriate polynomial,*

$$p \in \left[ \frac{\kappa \hat{r}}{d} \cdot \text{poly}\left( \frac{\log(\frac{d}{\delta})}{\gamma_{\text{sub}} \epsilon} \right), 1 \right], \ \gamma_{\text{sub}} \leq \frac{\epsilon}{36\theta}. \tag{4}$$

*Given one call to* $\mathcal{O}_p^{\text{sr}}(\mathbf{M}^\star + \mathbf{N})$ *for* $\|\mathbf{N}\|_\infty \leq \frac{\epsilon \gamma_{\text{sub}} \Delta}{20 d}$*, the following holds with probability* $\geq 1 - \delta$.

1. *If Algorithm 2 returns on Line 12, then* $\mathbf{M}_{U' \times U'}, \mathbf{M}^\star_{U' \times U'}$ *are* $(1 - \frac{\epsilon}{288})\Delta$-*close on a* $\gamma_{\text{sub}} + \frac{4}{\sqrt{\theta \kappa}}$-*submatrix, and* $|U \setminus U'| \leq 40(\gamma_{\text{sub}} + (\frac{\hat{r}}{\theta})^{1/2}) d \log d$.

2. *If Algorithm 2 returns on Line 16, then* $\mathbf{M}'_{U \times U}$, $\mathbf{M}^\star_{U \times U}$ *are* $10\epsilon^{\frac{1}{4}}\Delta$-*close on a* $2\gamma_{\text{sub}}$-*submatrix, and* $\mathbf{M}'$ *is given as a rank-$(3r + 2r^\star)$ factorization.*

*The runtime of the algorithm is*

$$O\left( m \cdot \text{poly}\left( \frac{\hat{r} \log(\frac{d}{\delta})}{\gamma_{\text{sub}} \epsilon} \right) \right),$$

*where* $m \geq d$ *is the number of observed entries upon calling* $\mathcal{O}_p^{\text{sr}}(\mathbf{M}^\star + \mathbf{N})$.

Proposition 1 is used to analyze one step of Algorithm 2, after we have already taken a few iterations and explicitly removed $\gamma_{\text{drop}} d$ rows and columns $[d] \setminus U$ from consideration, so that our remaining submatrices on $U \times U$ are close on a $\gamma_{\text{sub}}$-submatrix. Our goal will ultimately to have $\gamma_{\text{sub}}, \gamma_{\text{drop}} \leq \frac{\gamma_{\text{tot}}}{2}$ throughout the algorithm. Algorithm 2 provides bicriteria guarantees: in the "drop" case of Item 1, it slightly increases the $\gamma_{\text{drop}}$ parameter, increases $\gamma_{\text{sub}}$ by an even smaller amount (where the tradeoff is

given by a degree of freedom $\kappa$), and makes a small amount of progress (decreasing the closeness by $1 - O(\epsilon)$). In the "descent" case of Item 2, we instead decrease the closeness by a large $O(\text{poly}(\epsilon))$ factor, while doubling $\gamma_{\text{sub}}$ and roughly tripling the rank. By choosing the parameters appropriately, we show that iterating upon Proposition 1 yields Corollary 1, proven in Appendix B.

**Corollary 1.** *Let* $\mathbf{M}, \mathbf{M}^\star \in \mathbb{S}^{d \times d}$ *be rank-$r^\star$, and suppose* $\|\mathbf{M} - \mathbf{M}^\star\|_F \leq \Delta$*. Let* $\alpha \geq 250$ *and* $\delta, \gamma_{\text{tot}} \in (0, \frac{1}{10})$*. Algorithm 3 uses one call to* $\mathcal{O}_p^{\text{sr}}(\mathbf{M}^\star + \mathbf{N})$*, where for appropriate polynomials,*

$$\|\mathbf{N}\|_\infty \leq \frac{\Delta}{d \cdot \text{poly}(\frac{r^\star}{\gamma_{\text{tot}}} \log(\frac{d}{\delta})) \cdot \alpha^{1+o(1)}}, \ p = \frac{\text{poly}(\frac{r^\star}{\gamma_{\text{tot}}} \log(\frac{d}{\delta})) \cdot \alpha^{o(1)}}{d},$$

*and computes a rank-$r^\star \cdot \alpha^{o(1)}$-matrix* $\mathbf{M}'$*, such that with probability* $\geq 1 - \delta$*,* $\mathbf{M}'$ *and* $\mathbf{M}^\star$ *are* $\frac{\Delta}{\alpha}$*-close on a* $\gamma_{\text{tot}}$*-submatrix. The runtime of the algorithm is* $m\alpha^{o(1)} \cdot \text{poly}(\frac{r^\star}{\gamma_{\text{tot}}} \log(\frac{d}{\delta}))$*, where* $m$ *is the number of observed entries upon calling* $\mathcal{O}_p^{\text{sr}}(\mathbf{M}^\star + \mathbf{N})$*.*

We now explain the key computational method we develop to implement Proposition 1. We will solve a subproblem, defined in (5), to attempt to find a reweighting which makes progress. The subproblem enforces the conditions on candidate weights $w$ mentioned in Section 2, and is formally stated below:

$$F_{\text{prog-flat}}(x, \mathbf{S}) := - \langle \mathbb{1}_{\mathcal{I}}, x \rangle + CH_\eta^* \left( \text{slift} \left( \mathcal{D}\left(\frac{x}{v}\right) - \mathbf{S} \right) \right),$$

where $\mathcal{D}(w) := \sum_{\iota \in \mathcal{I}} w_\iota \mathbf{D}_\iota$, $v_\iota = [\mathbf{D}]_\iota^2$ for all $\iota \in \mathcal{I}$, and $\frac{x}{v}$ denotes entrywise division. (5)

The difference between (5) and our informal sketch in (2) is that we reparameterize $x \leftarrow w \circ v$ where $v$ is the squared observations of the difference matrix $\mathbf{D}$, and $\mathcal{I}$ is the set of observations. We also introduce a tradeoff parameter $\eta$ for controlling additive error bounds via Fact 1. Thus, the component $\langle \mathbb{1}_{\mathcal{I}}, x \rangle = \langle v, w \rangle$ captures the progress condition (Item 3), and the component $H_\eta^*(\text{slift}(\mathcal{D}(\frac{x}{v}) - \mathbf{S}))$ captures the flatness condition (Item 2). We optimize (5) over $\|\mathbf{S}\|_F \leq 1 + \epsilon$, the set of short matrices, and $x$ belonging to $\mathcal{X}$ defined in (22) which we observe is a rescaling of a bipartite matching polytope by a parameter $k$ (Fact 3). This polytope is used to enforce the spreadness condition (Item 1).

In the case we find a good solution to the subproblem, we can use the resulting solution to take a descent step and give the guarantee in Item 2 of Proposition 3. Otherwise, we have certified that excluding a few heavy rows and columns removed much of the progress that could be made (which we prove is the only way a good solution does not exist), and hence obtain the guarantee in Item 1. Our main guarantee for solving (5) at a fast rate is the following, proven in Appendix C.

**Proposition 2.** *Let* $\epsilon \in [0, C]$ *and* $\delta \in (0, 1)$*, and suppose* $|[\mathbf{D}]_\iota| \in [\ell, u]$ *for all* $\iota \in \mathcal{I}$*. There is an algorithm computing* $(x, \mathbf{S})$*, an* $\epsilon$*-approximate minimizer to (5) with probability* $\geq 1 - \delta$*, in time*

$$O\left( (|\mathcal{I}| + d) \, T^2 \cdot \text{poly}\left( \frac{CkR_\eta}{\epsilon} \log\left(\frac{d}{\delta}\right) \right) \right)$$

*for* $R_\eta := \eta^{-1}(2k + \frac{ku}{\ell^2 d})$ *and* $T = \lceil \frac{8C\eta R_\eta^2}{\epsilon} \rceil$*, and* $\mathbf{S}$ *is given as a rank-$r = O(\frac{C^2}{\epsilon^2} \cdot T)$ factorization.*

The last component in the proof of Theorem 1 is Algorithm 7, a "rounding" step where we take the output of Corollary 1, say $\mathbf{M}$, which is guaranteed to be close to $\mathbf{M}^\star$ on a large submatrix, and then postprocess it to a matrix $\widetilde{\mathbf{M}}$ that is guaranteed to be close to $\mathbf{M}^\star$ everywhere with some $\text{poly}(r, \mu, \log d)$-factor loss in the closeness. As long as the progress $\alpha$ made by Corollary 1 is larger than this factor, we can simply alternate Corollary 1 and the rounding step to still make constant-factor progress in each iteration. We prove the following result in Appendix D.

**Proposition 3.** *Let* $\mathbf{M}^\star \in \mathbb{S}^{d \times d}$ *be rank-$r^\star$ and* $\mu$*-incoherent, and let* $\widehat{\mathbf{M}} = \mathbf{M}^\star + \mathbf{N}$ *for* $\mathbf{N} \in \mathbb{S}^{d \times d}$ *with* $\|\mathbf{N}\|_\infty \leq \Delta$ *for some* $\Delta \geq 0$*. Let* $\mathbf{M} \in \mathbb{S}^{d \times d}$ *be given as a rank-$r$ decomposition, with* $r \geq r^\star$*. Further, for* $\gamma \in (0, 1)$ *suppose* $\mathbf{M}$ *and* $\mathbf{M}^\star$ *are* $d\Delta$*-close on a* $\gamma$*-submatrix. Finally, assume*

$$\gamma \leq \frac{1}{10^4 \mu r \log(d)}.$$

*Then for any* $\delta \in (0, 1)$ *if* $p \geq \frac{1}{d} \cdot \text{poly}(\mu r \log(\frac{d}{\delta}))$ *for an appropriate polynomial, Algorithm 7 uses one call to* $\mathcal{O}_p(\widehat{\mathbf{M}})$ *and with probability* $\geq 1 - \delta$*, outputs* $\widetilde{\mathbf{M}} \in \mathbb{S}^{d \times d}$ *such that*

$$\left\| \widetilde{\mathbf{M}} - \mathbf{M}^\star \right\|_\infty \leq \text{poly}\left( \mu r \log\left(\frac{d}{\delta}\right) \right) \Delta. \tag{6}$$

*Also, $\widetilde{\mathbf{M}}$ is given as a rank-$\mathrm{poly}(\mu r \log(\frac{d}{\delta}))$ factorization. The algorithm runs in $m \cdot \mathrm{poly}(\mu r \log(\frac{d}{\delta}))$ time, where $m$ is the number of observed entries upon calling $\mathcal{O}_p^{\mathrm{sr}}(\widehat{\mathbf{M}})$.*

## Acknowledgments and Disclosure of Funding

AL was supported in part by an NSF GRFP and a Hertz Fellowship. AS was supported in part by a Microsoft Research Faculty Fellowship, NSF CAREER Grant CCF-1844855, NSF Grant CCF-1955039, and a PayPal research award. JK was supported in part by NSF awards CCF-1955217, CCF-1565235, and DMS-2022448. Part of this work was conducted while visiting the Simons Institute for the Theory of Computing.

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

# A  Additional preliminaries

In this section, we formalize some additional preliminaries and helpful notation which will be used in the remainder of the appendices.

**General notation.** For $n \in \mathbb{N}$ we denote $[n] := \{i \in \mathbb{N} \mid i \le n\}$. We say a vector is $s$-sparse if it has $\le s$ nonzero entries. When $S \subseteq T$ and $T$ is clear from context, we let $S^c := T \setminus S$. The all-zeroes and all-ones vectors of dimension $d$ are denoted $\mathbb{0}_d$ and $\mathbb{1}_d$. Applied to a vector, $\|\cdot\|_p$ denotes the $\ell_p$ norm for $p \ge 1$. Applied to a matrix, $\|\cdot\|_p$ denotes the $\ell_p$ norm of the vector corresponding to the flattened matrix. We use $\mathrm{med}(S)$ to denote the median of a set $S$. For $t \in \mathbb{R}$ we let $\mathrm{sign}(t) = 1$ if $t \ge 0$ and $\mathrm{sign}(t) = -1$ otherwise. We let $\circ$ denote the entrywise product of two vectors.

**Matrices.** Matrices are denoted in boldface. The number of nonzero entries of a matrix $\mathbf{M}$ is denoted $\mathrm{nnz}(\mathbf{M})$, and the set of indices corresponding to these nonzero entries, i.e., the support, is denoted $\mathrm{supp}(\mathbf{M})$. We define $\mathcal{T}_{\mathrm{mv}}(\mathbf{M})$ as the amount of time it takes to compute $\mathbf{M}v$ for any vector $v$; note $\mathcal{T}_{\mathrm{mv}}(\mathbf{M}) = O(\mathrm{nnz}(\mathbf{M}))$ if $\mathbf{M}$ has explicit entries, and if $\mathbf{M} \in \mathbb{R}^{m \times n}$ is given as a rank-$r$ factorization then $\mathcal{T}_{\mathrm{mv}}(\mathbf{M}) = O((m + n)r)$. We equip $\mathbb{R}^{m \times n}$ with the inner product $\langle \mathbf{A}, \mathbf{B} \rangle := \mathrm{Tr}(\mathbf{A}^\top \mathbf{B})$, denote the $d \times d$ identity by $\mathbf{I}_d$, and the all-zero $m \times n$ matrix by $\mathbf{0}_{m \times n}$. The $i^{\text{th}}$ row and $j^{\text{th}}$ column of a matrix $\mathbf{M}$ are respectively denoted $\mathbf{M}_{i:}$ and $\mathbf{M}_{:j}$. For a collection of matrices $\{\mathbf{A}_i\}_{i \in [k]} \in \mathbb{R}^{m \times n}$ with an associated operator $\mathcal{A}$ (clear from context), we denote $\mathcal{A}(x) := \sum_{i \in [k]} x_i \mathbf{A}_i$ for $x \in \mathbb{R}^k$. We say $\mathbf{M}$ is $s$-row column sparse (RCS) if its rows and columns are $s$-sparse. For $S \subseteq [m]$, $T \subseteq [n]$, we let the submatrix of $\mathbf{M} \in \mathbb{R}^{m \times n}$ indexed by $S, T$ be denoted $\mathbf{M}_{S \times T}$. When $S$ is a singleton $\{i\}$ for $i \in [m]$, we denote this $\mathbf{M}_{i,T}$ and similarly define $\mathbf{M}_{S,j}$ for $j \in [n]$.

The Frobenius, operator, and trace (nuclear) norms of a matrix are denoted $\|\cdot\|_{\mathrm{F}}$, $\|\cdot\|_{\mathrm{op}}$, and $\|\cdot\|_{\mathrm{tr}}$ respectively, corresponding to the 2-norm, $\infty$-norm, and 1-norm of the singular values of the matrix. We denote the symmetric $d \times d$ matrices by $\mathbb{S}^{d \times d}$ and the positive semidefinite cone by $\mathbb{S}_{\succeq \mathbf{0}}^{d \times d}$. We say $\mathbf{M} \in \mathbb{R}^{m \times n}$ has singular value decomposition (SVD) $\mathbf{M} = \mathbf{U}\boldsymbol{\Sigma}\mathbf{V}^\top$ if $\mathbf{M}$ is rank-$r$, $\boldsymbol{\Sigma} \in \mathbb{R}^{r \times r}$ is diagonal, and $\mathbf{U} \in \mathbb{R}^{m \times r}$, $\mathbf{V} \in \mathbb{R}^{n \times r}$ have orthonormal columns. The ordered singular values of $\mathbf{M} \in \mathbb{R}^{m \times n}$ are denoted $\{\sigma_i(\mathbf{M})\}_{i \in [\min(m,n)]}$ where $\sigma_1(\mathbf{M})$ is largest. Similarly we define the ordered eigenvalues of $\mathbf{M} \in \mathbb{S}^{d \times d}$ by $\{\lambda_i(\mathbf{M})\}_{i \in [d]}$. For $\mathbf{M} \in \mathbb{S}^{d \times d}$ with eigendecomposition $\mathbf{U}\boldsymbol{\Lambda}\mathbf{U}^\top$, we define $\exp(\mathbf{M}) := \mathbf{U}\exp(\boldsymbol{\Lambda})\mathbf{U}^\top$ where on the right-hand side, $\exp$ is applied entrywise on the diagonal. We also define $\log \mathbf{M}$ for $\mathbf{M} \in \mathbb{S}_{\succeq \mathbf{0}}^{d \times d}$ and $|\mathbf{M}|$ for $\mathbf{M} \in \mathbb{S}^{d \times d}$ so the functions are applied to the eigenvalues in the appropriate basis. Finally, we let $\mathrm{span}(\mathbf{A})$ denote the column span (or image) of a matrix $\mathbf{A}$, and for a subspace $V \subset \mathbb{R}^d$, we let $\boldsymbol{\Pi}_V$ denote the orthogonal projection matrix onto $V$.

**Index sets.** For $d \in \mathbb{N}$, we let $S(d)$ contain all symmetrized indices of the form:

$$\{(i,i)\} \text{ for } i \in [d], \text{ or } \{(i,j),(j,i)\} \text{ for } i,j \in [d], i \ne j.$$

We use $(i,j)^\sharp$ as shorthand for the singleton $\{(i,i)\}$ if $i = j$, and otherwise as shorthand for $\{(i,j),(j,i)\}$. When $\mathcal{I}$ is a set of indices in $[d] \times [d]$, we let $\mathbf{M}_{\mathcal{I}}$ be the $d \times d$ matrix which zeroes out all entries of $\mathbf{M}$ except those in $\iota$. For a symmetrized index $\iota \in S(d)$, we denote the scalar value of the $\iota$ entry by $[\mathbf{M}]_\iota$ (note that this is well-defined even when $\iota$ contains two indices due to symmetry). The zero-one matrix whose only nonzero entries are $\iota$ is denoted $\mathbf{E}_\iota$, so that $\mathbf{M}_\iota = [\mathbf{M}]_\iota \mathbf{E}_\iota$ for all $\iota \in S(d)$. For $\mathcal{I} \subseteq S(d)$ clear from context, we let $\mathcal{E}$ be the operator on $w \in \mathbb{R}^{\mathcal{I}}$ which forms the associated $d \times d$ matrix whose indices on $\mathcal{I}$ agree with $w$ and is otherwise zero entrywise, i.e.,

$$\mathcal{E}(w) := \sum_{\iota \in \mathcal{I}} w_\iota \mathbf{E}_\iota.$$

**Optimization.** For convex $f : \mathcal{X} \to \mathbb{R}$ with $\mathcal{X} \subseteq \mathbb{R}^d$, we let $\partial f(x)$ denote the subgradient set at $x \in \mathcal{X}$. We let $\|y\|_* := \sup_{\|x\| \le 1} x^\top y$ denote the dual norm to a norm $\|\cdot\|$. When $\mathbf{M}$ is a $d \times d$ linear operator we use the notation $\mathbf{M}[u,v] := \langle \mathbf{M}u, v \rangle$. This becomes relevant when $\mathcal{M}$ is a Hessian of a matrix function (a linear operator on $d \times d$ matrices), so $\mathcal{M}[\mathbf{U}, \mathbf{V}]$ evaluates this operator on $\mathbf{U}, \mathbf{V}$.

**Reduction to symmetric matrices.** We briefly justify our focus on symmetric matrices in Definition 3. We begin by recalling two facts.

**Fact 2.** *Let $m, n \in \mathbb{N}$ with $m \geq n$. There exists $k \in \mathbb{N}$ such that $kn \in [\frac{m}{2}, m]$.*

**Lemma 1.** *Let $m, n, k \in \mathbb{N}$ and let $\mathbf{M} \in \mathbb{R}^{m \times n}$ be rank-$r$ with SVD $\mathbf{M} = \mathbf{U\Sigma V}^\top$. Suppose that both $U := \mathrm{span}(\mathbf{U})$ and $V := \mathrm{span}(\mathbf{V})$ are $\mu$-incoherent. Then, letting $\mathbf{M}' \in \mathbb{R}^{m \times kn}$ be defined as*

$$\mathbf{M}' := \underbrace{(\mathbf{M} \quad \mathbf{M} \quad \ldots \quad \mathbf{M} \quad \mathbf{M})}_{k \text{ times}},$$

*and letting $\mathbf{M}'$ have SVD $\mathbf{M}' = \mathbf{U\Sigma' W}^\top$, then $W := \mathrm{span}(\mathbf{W})$ is $\mu$-incoherent.*

*Proof.* It is straightforward to check that a valid SVD is given by

$$\mathbf{\Sigma}' = \sqrt{k}\mathbf{\Sigma}, \ \mathbf{W}^\top = \frac{1}{\sqrt{k}} \underbrace{(\mathbf{V}^\top \quad \mathbf{V}^\top \quad \ldots \quad \mathbf{V}^\top \quad \mathbf{V}^\top)}_{k \text{ times}}.$$

Because incoherence is invariant to the choice of SVD, the conclusion holds. $\square$

Consider an asymmetric variant of the problem in Definition 3, where $\mathbf{M}^\star \in \mathbb{R}^{m \times n}$ for $m \geq n$ is rank-$r^\star$ and $\mu$-incoherent, and we have $p$-semi-random observation oracle access (defined analogously to Definition 2) to $\mathbf{M}^\star$. Let $k$ be the integer from Fact 2, and let $\widehat{\mathbf{M}^\star} \in \mathbb{R}^{m \times kn}$ be constructed as in Lemma 1 by concatenating $\mathbf{M}^\star$ $k$ times. The row and column spans of $\widehat{\mathbf{M}^\star}$ are $\mu$-incoherent, by Lemma 1. Moreover, we can simulate $p$-semi-random observation oracle access to

$$\begin{pmatrix} \mathbf{0}_{m \times m} & \widehat{\mathbf{M}^\star} \\ \left[\widehat{\mathbf{M}^\star}\right]^\top & \mathbf{0}_{kn \times kn} \end{pmatrix} \in \mathbb{S}^{(m+kn) \times (m+kn)}$$

in the symmetric sense of Definition 3, given asymmetric oracle access to $\mathbf{M}^\star$. It is straightforward to check that because $\widehat{\mathbf{M}^\star}$ has dimensions within a constant factor of each other, the incoherence parameter is only affected by a constant in this lifting process. For simplicity in the remainder of the paper, we focus on the case of symmetric matrices, i.e., the problem statement in Definition 3.

**Sample splitting.** Finally, we recall a convenient subsampling claim from [43] that extends to the semi-random model, which lets us use $\mathcal{O}_p^{\mathrm{sr}}(\mathbf{M})$ to simulate access to $\mathcal{O}_q^{\mathrm{sr}}(\mathbf{M})$ for smaller values $q \leq p$.

**Lemma 2.** *Let $\{p_k\}_{k \in [K]} \subset (0, 1)$ satisfy $p_k \leq p \leq \frac{1}{K}$ for all $k \in [K]$, and let $\mathbf{M}^\star \in \mathbb{S}^{d \times d}$. We can simulate sequential access to $\mathcal{O}_{p_k}^{\mathrm{sr}}(\mathbf{M}^\star)$ for all $k \in [K]$ with access to $\mathcal{O}_{Kp}^{\mathrm{sr}}(\mathbf{M}^\star)$.*

*Proof.* The claim is immediate from the observation that the proof of Lemma 1, [43] is monotone in $p$. In particular, the same subsampling process in [43] preserves monotonicity of observation probabilities and hence produces valid semi-random observation oracles. $\square$

## B  Semi-random partial matrix completion

In this section, we develop Algorithm 3, our main subroutine for partial matrix completion in the semi-random model (Definition 2). Algorithm 3 uses recursive calls to Algorithm 2, DropOrDescent, which implements a single step of our partial completion method.

At a high level, one application of DropOrDescent is parameterized by a current number of dropped rows $\gamma_{\mathrm{drop}}$, a current number of excluded rows $\gamma_{\mathrm{sub}}$, a subset of undropped rows $U = [d] \setminus D$ satisfying $|D| \leq \gamma_{\mathrm{drop}}d$, and a rank-$r$ matrix $\mathbf{M}$ such that $\mathbf{M}_{U \times U}$ is promised to be close to our target $\mathbf{M}_{U \times U}^\star$ on a $\gamma_{\mathrm{sub}}$-submatrix (Definition 4). It then distinguishes between the following cases from semi-random, noisy observations of $\mathbf{M}^\star$, by solving a testing problem via optimization.

1. In the first case, dropping an additional $\gamma$ fraction of rows and columns decreases the closeness parameter $\Delta$ by a small amount, i.e. a small subset of rows or columns are responsible for a nontrivial portion of the mass. We call this case the "drop" case.

2. In the second case, after removing a few rows and columns, the remaining mass is still large. We show that in this case, by deleting a particular subset of problematic rows or columns, we can use well-conditionedness properties of the remaining submatrix to decrease $\Delta$ by a constant factor. We do so by solving an optimization problem, which finds a spread reweighting of observations with a short-flat decomposition. We call this case the "descent" case.

We analyze the descent case in Appendix B.1, where we demonstrate sufficient conditions on a proposed reweighted step which guarantee progress under submatrix closeness. The candidate steps we analyze in this section are all reweightings of truncated observations, which are found by solving an appropriate optimization problem (see (20)). We then analyze the drop case in Appendix B.2, where we show that either our current difference matrix has Frobenius norm which is unstable to dropping few rows and columns, or a good solution exists which is compatible with our descent algorithm. We give an explicit dropping procedure for handling the former case in Appendix B.3. We finally put these cases together and give a win-win analysis of DropOrDescent in Appendix B.4.

## B.1 Descent steps

In this section, we give our progress analysis in the case where we can find sufficient signal in our observations. Specifically, Lemma 7 gives sufficient conditions on a reweighted, truncated observation matrix such that we can improve our submatrix closeness parameter by a constant factor.

For $\mathbf{D} \in \mathbb{S}^{d \times d}$, we first define a clipping operation which will be used to control our reweightings. We let $\mathrm{clip}_u(\mathbf{D})$ be the matrix which clips all entries to the range $[-u, u]$, formally defined by

$$[\mathrm{clip}_u(\mathbf{D})]_{ij} = \mathrm{med}\left(-u, \mathbf{D}_{ij}, u\right). \tag{7}$$

We also will use the following claim which follows from straightforward casework.

**Lemma 3.** Let $u \geq 0$, $|a| \leq u$, and $a' := \mathrm{med}(-u, a + b, u)$. Then $|a' - a| \leq |b|$.

*Proof.* If $|a + b| \leq u$ the claim is obvious. Otherwise, if $a' = u$ it must be the case that $b \geq 0$, and $0 \leq a' - a \leq b$. Similarly, if $a' = -u$ we must have $0 \geq a' - a \geq -b$. □

We now introduce some additional notation. Let $\mathcal{I} \subset S(d)$ be a set of symmetrized indices under consideration (corresponding to observations). For $w \in \mathbb{R}^{\mathcal{I}}$ and $\widehat{\mathbf{D}} \in \mathbb{S}^{d \times d}$ clear from context, we let

$$\widehat{\mathcal{D}}(w) := \sum_{\iota \in \mathcal{I}} w_\iota \widehat{\mathbf{D}}_\iota. \tag{8}$$

For a parameter $\rho$, we define the set of "spread" reweightings of the observations $\widehat{\mathbf{D}}_{\mathcal{I}}$ by

$$\mathcal{W}_\rho := \left\{ w \in \mathbb{R}^{\mathcal{I}}_{\geq 0} \mid \max_{i \in [d]} \sum_{j \in [d] | \iota = (i,j)^\sharp \in \mathcal{I}} w_\iota [\widehat{\mathbf{D}}]_\iota^2 \leq \rho \right\}. \tag{9}$$

Further, we require the following result on low-rank approximation.

**Lemma 4** (Theorem 1, [56]). *Let $\delta, \epsilon \in (0, 1)$, let $\mathbf{M} \in \mathbb{S}^{d \times d}$, and let $q \in [d]$. There is an algorithm which returns $\mathbf{Z} \in \mathbb{R}^{d \times q}$ with orthonormal columns such that*

$$\left\| \mathbf{Z}\mathbf{Z}^\top \mathbf{M} \mathbf{Z} \mathbf{Z}^\top - \mathbf{M} \right\|_{\mathrm{op}} \leq (1 + \epsilon)\sigma_{q+1}(\mathbf{M})$$

*with probability $\geq 1 - \delta$, in time*

$$O\left( \mathcal{T}_{\mathrm{mv}}(\mathbf{M}) \cdot \frac{q \log \frac{d}{\delta}}{\sqrt{\epsilon}} \right).$$

*Moreover, we have for all $i \in [q]$, letting $z_i$ be the $i^{th}$ column of $\mathbf{Z}$,*

$$\sqrt{z_i^\top \mathbf{M}^2 z_i} \in \left[(1 - \epsilon)\sigma_i(\mathbf{M}),\ (1 + \epsilon)\sigma_i(\mathbf{M})\right].$$

Finally, we provide ways to bound the singular values of an approximately low-rank matrix.

**Lemma 5.** *Let $\mathbf{M}, \mathbf{N} \in \mathbb{S}^{d \times d}$ have eigendecompositions $\mathbf{M} = \mathbf{U}\boldsymbol{\Lambda}\mathbf{U}^\top$ and $\mathbf{N} = \mathbf{V}\boldsymbol{\Sigma}\mathbf{V}^\top$, where $\boldsymbol{\Lambda} = \mathbf{diag}\,(\lambda)$, $\boldsymbol{\Sigma} = \mathbf{diag}\,(\sigma)$, and $\lambda, \sigma \in \mathbb{R}^d$ are in nonincreasing order. Then, $\|\mathbf{M} - \mathbf{N}\|_{\mathrm{F}} \geq \|\lambda - \sigma\|_2$.*

*Proof.* Since $\|\mathbf{M}\|_{\mathrm{F}}^2 = \|\lambda\|_2^2$ and $\|\mathbf{N}\|_{\mathrm{F}}^2 = \|\sigma\|_2^2$, expanding the squares of both sides shows that it suffices to prove that $\langle \mathbf{M}, \mathbf{N} \rangle \leq \langle \lambda, \sigma \rangle$, which follows from the von Neumann trace inequality. $\square$

**Lemma 6.** *Let $\epsilon \in [0, 1]$, let $\mathbf{M} \in \mathbb{S}^{d \times d}$ be rank-$\hat{r}$, and suppose $\|\mathbf{M} - \mathbf{S}\|_{\mathrm{F}} \leq \epsilon$ for $\mathbf{S} \in \mathbb{S}^{d \times d}$. Then for any $s \in \mathbb{N}$, we have $\sigma_{\hat{r}+s+1}(\mathbf{S}) \leq \frac{\epsilon}{s^{1/2}}$.*

*Proof.* Let $\lambda \in \mathbb{R}^d$ be the set of eigenvalues of $\mathbf{M}$ in nonincreasing order, and similarly let $\sigma \in \mathbb{R}^d$ be the nonincreasing eigenvalues of $\mathbf{S}$. By assumption, $\lambda_i = 0$ for all $i \geq \hat{r} + 1$, and $\|\lambda - \sigma\|_2 \leq \epsilon$ by Lemma 5. Thus, at most $\hat{r} + s$ eigenvalues of $\mathbf{S}$ can be larger than $\frac{\epsilon}{s^{1/2}}$, giving the conclusion. $\square$

We are now ready to give our main progress guarantee in Lemma 7. Specifically, we show that a reweighting of truncated observations which has enough signal and admits a short-flat decomposition is guaranteed to make significant progress, as long as it is sufficiently spread in the sense of (9).

**Lemma 7.** *Let $\epsilon, \gamma \in (0, \frac{1}{10})$, $\ell, u \geq 0$, and $T \subseteq S \subseteq U \subseteq [d]$ such that $|U \setminus S| \leq \gamma d$, $|S \setminus T| \leq \gamma d$, and $\ell \leq \frac{\epsilon}{d}$. Suppose that for $\mathbf{D}, \mathbf{D}^\star, \mathbf{N} \in \mathbb{S}^{d \times d}$, we have $\mathbf{D} = \mathbf{D}^\star + \mathbf{N}$, for rank-$\hat{r}$ $\mathbf{D}^\star$ with $\|\mathbf{D}^\star_{S \times S}\|_{\mathrm{F}} \leq 1$, $\|\mathbf{D}^\star_{T \times T}\|_\infty \leq u$, and $\|\mathbf{N}\|_\infty \leq \gamma\ell$. Let*

$$\widehat{\mathbf{D}} := \mathrm{clip}_u(\mathbf{D}),$$

*and for $\mathcal{I} \subseteq S(d) \cap (U \times U)$, $w \in \mathbb{R}^{\mathcal{I}}_{\geq 0}$, suppose the following conditions hold for $\mathbf{S}, \mathbf{F} \in \mathbb{S}^{d \times d}$ supported on $\mathcal{I} \cap (U \times U)$ with $\widehat{\mathcal{D}}(w) = \mathbf{F} + \mathbf{S}$.*

1. *$w \in \mathcal{W}_\rho$ for $\rho = \frac{\epsilon}{2\gamma d}$, recalling the definition (9).*
2. *$\langle \widehat{\mathbf{D}}, \widehat{\mathcal{D}}(w) \rangle \geq 1 - \epsilon$.*
3. *$\|\mathbf{S}\|_{\mathrm{F}} \leq 1 + \epsilon$.*
4. *$\|\mathbf{F}\|_{\mathrm{op}} \leq \frac{\epsilon}{\hat{r}^{1/2}}$.*
5. *$\min_{\iota \in \mathcal{I}} |[\widehat{\mathbf{D}}]_\iota| \geq \ell$.*

*Then, letting $\mathbf{Z} \in \mathbb{R}^{d \times 2\hat{r}}$ be the output of Lemma 4 with parameters $\delta \in (0, 1)$, $\epsilon \leftarrow \frac{1}{2}$, $\mathbf{M} \leftarrow \mathbf{S}$, and $q \leftarrow 2\hat{r}$, we have with probability $\geq 1 - \delta$ that*

$$\left\| \left[ \mathbf{D}^\star - \mathbf{Z}\mathbf{Z}^\top \mathbf{S}\mathbf{Z}\mathbf{Z}^\top \right]_{T \times T} \right\|_{\mathrm{F}} \leq 10\epsilon^{\frac{1}{4}}.$$

*Proof.* As a starting point, we claim that

$$\|\mathbf{D}^\star_{T \times T} - \mathbf{S}\|_{\mathrm{F}} \leq \sqrt{13\epsilon}. \tag{10}$$

To see this, we expand to show that

$$
\begin{aligned}
\left\| \mathbf{D}^\star_{T \times T} - \mathbf{S} \right\|_{\mathrm{F}}^2 &\leq 2 + 3\epsilon - 2\left\langle \mathbf{D}^\star_{T \times T}, \mathbf{S}_{T \times T} \right\rangle \\
&\leq 2 + 3\epsilon - 2\left\langle \mathbf{D}^\star_{T \times T}, \left[\widehat{\mathcal{D}}(w)\right]_{T \times T} \right\rangle + 2\left\| \mathbf{D}^\star_{T \times T} \right\|_{\mathrm{tr}} \|\mathbf{F}\|_{\mathrm{op}} \\
&\leq 2 + 5\epsilon - 2\left\langle \widehat{\mathbf{D}}_{T \times T}, \left[\widehat{\mathcal{D}}(w)\right]_{T \times T} \right\rangle + 2\left\| \left[\mathbf{D}^\star - \widehat{\mathbf{D}}\right]_{T \times T} \right\|_\infty \left\| \widehat{\mathcal{D}}(w) \right\|_1 \\
&\leq 2 + 7\epsilon - 2\left\langle \widehat{\mathbf{D}}_{T \times T}, \left[\widehat{\mathcal{D}}(w)\right]_{T \times T} \right\rangle.
\end{aligned}
\tag{11}
$$

The first line used Item 3 and $\|\mathbf{D}^\star_{S \times S}\|_{\mathrm{F}} \leq 1$, and the second used the (matrix) Hölder's inequality. The third used Item 4 and that $\|\widehat{\mathbf{D}}_{T \times T}\|_{\mathrm{tr}} \leq \sqrt{\hat{r}}\|\widehat{\mathbf{D}}_{T \times T}\|_{\mathrm{F}}$, and the (vector) Hölder's inequality. The last line used $\|\mathbf{D}^\star_{T \times T}\|_\infty \leq u$, the definition of $\widehat{\mathbf{D}}$, and Lemma 3 to conclude

$$\left\| \left[\mathbf{D}^\star - \widehat{\mathbf{D}}\right]_{T \times T} \right\|_\infty \leq \|\mathbf{N}_{T \times T}\|_\infty \leq \gamma\ell,$$

and also bounded $\|\widehat{\mathcal{D}}(w)\|_1$ via Items 1 and 5, which show

$$\|\widehat{\mathcal{D}}(w)\|_1 \leq \left\langle \widehat{\mathbf{D}}, \widehat{\mathcal{D}}(w) \right\rangle \cdot \left( \max_{\iota \in \mathcal{I}} \frac{1}{\left| [\widehat{\mathbf{D}}]_\iota \right|} \right) \leq \frac{\rho d}{\ell} = \frac{\epsilon}{2\gamma\ell}.$$

Finally, since $w$ is supported on $U \times U$, applying $|U \setminus T| \leq 2\gamma d$, Item 1, and Item 2 yields

$$\left\langle \widehat{\mathbf{D}}_{T \times T}, \left[ \widehat{\mathcal{D}}(w) \right]_{T \times T} \right\rangle \geq \left\langle \widehat{\mathbf{D}}, \widehat{\mathcal{D}}(w) \right\rangle - 4\gamma\rho d \geq 1 - 3\epsilon. \tag{12}$$

Then, (10) follows by combining (11) and (12). Next, applying Lemma 6 using (10) shows that

$$\sigma_{2\hat{r}+1}(\mathbf{S}) \leq \sqrt{\frac{13\epsilon}{\hat{r}}}.$$

Therefore, assuming the algorithm in Lemma 4 succeeds, we have

$$\underbrace{\mathbf{ZZ}^\top \mathbf{SZZ}^\top}_{:=\mathbf{S}'} = \mathbf{S} + \underbrace{(\mathbf{ZZ}^\top \mathbf{SZZ}^\top - \mathbf{S})}_{:=\mathbf{F}'}, \text{ where } \|\mathbf{F}'\|_{\mathrm{op}} \leq \frac{6\sqrt{\epsilon}}{\sqrt{\hat{r}}}.$$

At this point, we can write $\mathcal{D}(w) = \mathbf{S}' + (\mathbf{F} + \mathbf{F}')$, where by the triangle inequality

$$\|\mathbf{F} + \mathbf{F}'\|_{\mathrm{op}} \leq 7\sqrt{\epsilon} \cdot \frac{1}{\sqrt{\hat{r}}}.$$

Since $\|\mathbf{S}'\|_{\mathrm{F}} \leq \|\mathbf{S}\|_{\mathrm{F}} \leq 1 + \epsilon$, the remainder of the argument is identical to the proof of (10) with the worse flatness parameter, where we use that $\|\mathbf{S}'_{T \times T}\|_{\mathrm{F}} \leq \|\mathbf{S}'\|_{\mathrm{F}}$ and $\|\mathbf{S}_{T \times T}\|_{\mathrm{F}} \leq \|\mathbf{S}\|_{\mathrm{F}}$. □

### B.2 Existence of good solutions

In this section, we define a stability condition (14) under which implies a solution exists which satisfies the conditions in Lemma 7. We use this existence claim to certify progress in one of two ways in our final algorithm. In Lemma 11, we show that under (14), there exists a reweighting (with high probability over the semi-random observations) which is compatible with Lemma 7's requirements. We then show in Lemma 12 that in the absence of (14), one can increase the drop parameter to improve the submatrix closeness bound. We will analyze an explicit dropping procedure we use in this latter case in Appendix B.3. We first require a helper guarantee from prior work on controlling entries of a low-rank matrix after dropping a few problematic rows and columns.

**Lemma 8** (Lemma 6, [43]). *Let $\mathbf{A} \in \mathbb{S}^{d \times d}$ be a rank-$\hat{r}$ matrix with $\|\mathbf{A}\|_{\mathrm{F}} \leq 1$, and let $\gamma \in [0, 1]$. There is $S' \subseteq [d]$ with $|S'| \geq (1 - \gamma)d$, such that*

$$\|\mathbf{A}_{S' \times S'}\|_\infty \leq \frac{\sqrt{\hat{r}}}{\gamma d}. \tag{13}$$

We also require a bound on the sampling error of random observations from a matrix with bounded entries, rows, and columns, which is an application of the matrix Bernstein inequality.

**Lemma 9** (Lemma 7, [43]). *Let $p, \delta \in [0, \frac{1}{2}]$, $\rho, \tau \geq 0$, and let $\mathbf{A} \in \mathbb{S}^{d \times d}$ satisfy*

$$\|\mathbf{A}_{i:}\|_2 \leq \rho \text{ for all } i \in [d], \text{ and } |\mathbf{A}_{ij}| \leq \tau \text{ for all } (i, j) \in [d] \times [d].$$

*Let $\Omega \subseteq S(d)$ be a random subset where each $\iota \in S(d)$ is included in $\Omega$ with probability $p$, and let $\widetilde{\mathbf{A}} := \frac{1}{p}\mathbf{A}_\Omega$. Then with probability at least $1 - \delta$,*

$$\left\| \mathbf{A} - \widetilde{\mathbf{A}} \right\|_{\mathrm{op}} \leq 4 \max \left( \frac{\rho}{\sqrt{p}} \sqrt{\log\left(\frac{d}{\delta}\right)}, \frac{\tau}{p} \log\left(\frac{d}{\delta}\right) \right).$$

Finally, we need a guarantee on the concentration of empirical estimates of squared $\ell_2$ norms.

**Lemma 10** (Lemma 3, [43]). *Let $p, \delta, \alpha \in (0, 1)$, let $v \in \mathbb{R}^d$ have $\|v\|_\infty \leq \tau$, and let $\tilde{v} \in \mathbb{R}^d$ have each $\tilde{v}_i$ independently set to $v_i$ with probability $p$, and $0$ otherwise. Then with probability $\geq 1 - \delta$,*

$$\left| \|v\|_2^2 - \frac{1}{p} \|\tilde{v}\|_2^2 \right| \leq \max \left( \alpha \|v\|_2^2, \frac{3\tau^2 \log \frac{2}{\delta}}{\alpha p} \right).$$

We are now ready to give our main result on sufficient conditions for a good solution to exist.

**Lemma 11.** *Let $\delta, \epsilon, \gamma, p \in (0, \frac{1}{10})$, and $S \subseteq U \subseteq [d]$ such that $|U \setminus S| \leq \gamma d$. Let*

$$\ell := \frac{\epsilon}{20d}, \; u := \frac{2\sqrt{\hat{r}}}{\gamma d}, \; p \geq \frac{240\hat{r}\log(\frac{6d}{\delta})}{\gamma^3\epsilon^2 d}.$$

*Suppose that for $\mathbf{D}, \mathbf{D}^\star, \mathbf{N} \in \mathbb{S}^{d \times d}$, we have $\mathbf{D} = \mathbf{D}^\star + \mathbf{N}$, for rank-$\hat{r}$ $\mathbf{D}^\star$ with $\|\mathbf{D}^\star_{S \times S}\|_{\mathrm{F}} \leq 1$ and $\|\mathbf{N}\|_\infty \leq \gamma\ell$. Further, for $\widehat{\mathbf{D}} := \mathrm{clip}_u(\mathbf{D})$, let $\Omega^\star := \{\iota \in S(d) \cap (U \times U) \mid |[\widehat{\mathbf{D}}]_\iota| \geq \ell\}$. Finally, let $S'$ be the result of applying Lemma 8 with $\mathbf{A} \leftarrow \mathbf{D}^\star_{S \times S}$, $\gamma \leftarrow \frac{\gamma}{2}$, so $\|\mathbf{D}^\star_{S' \times S'}\|_\infty \leq u$, and let $T := S' \setminus L$ where $L$ is the indices of the $\lfloor \frac{\gamma d}{2} \rfloor$ rows of $\widehat{\mathbf{D}}_{S' \times S'}$ with largest $\ell_2$ norms, so $|S \setminus T| \leq \gamma d$. Suppose*

$$\left\|\widehat{\mathbf{D}}_{T \times T}\right\|_{\mathrm{F}} \geq 1 - \frac{\epsilon}{30}. \tag{14}$$

*Letting $\Omega \subseteq S(d)$ have each $\iota \in S(d)$ independently included with probability $p$, and $w^\star \in \{0, \frac{1}{p}\}^\Omega$ have $[w^\star]_\iota = \frac{1}{p}$ if and only if $\iota \in (T \times T) \cap \Omega^\star$, the following conditions all hold with probability $\geq 1 - \delta$ over the randomness of $\Omega$, and $\mathbf{S}^\star, \mathbf{F}^\star \in \mathbb{S}^{d \times d}$ supported on $(T \cap T) \cap \Omega^\star$ with $\widehat{\mathcal{D}}(w^\star) = \mathbf{F}^\star + \mathbf{S}^\star$.*

1. *$w^\star \in \mathcal{W}_\rho$ for $\rho = \frac{\epsilon}{2\gamma d}$, recalling the definition (9).*

2. *$\langle \widehat{\mathbf{D}}, \widehat{\mathcal{D}}(w^\star) \rangle \geq 1 - \frac{\epsilon}{2}$.*

3. *$\|\mathbf{S}^\star\|_{\mathrm{F}} \leq 1 + \epsilon$.*

4. *$\|\mathbf{F}^\star\|_{\mathrm{op}} \leq \frac{\gamma\epsilon}{4\hat{r}^{1/2}}$.*

*Proof.* For notational convenience, let $\mathcal{L}^\star := S(d) \setminus \Omega^\star$ be the set of indices $\iota \in S(d)$ with $|[\widehat{\mathbf{D}}]_\iota| < \ell$. We define our decomposition of $\widehat{\mathcal{D}}(w^\star)$ as follows, where we recall $w^\star$ is supported on $(T \times T) \cap \Omega^\star$:

$$\widehat{\mathcal{D}}(w^\star) := \underbrace{\widehat{\mathbf{D}}_{(T \times T) \cap \Omega^\star}}_{:=\mathbf{S}^\star} + \underbrace{\widehat{\mathcal{D}}(w^\star) - \widehat{\mathbf{D}}_{(T \times T) \cap \Omega^\star}}_{:=\mathbf{F}^\star}.$$

Now, clearly $\|\mathbf{N}\|_{\mathrm{F}} \leq d\|\mathbf{N}\|_\infty \leq \frac{\epsilon}{200}$, so $\|\mathbf{S}^\star\|_{\mathrm{F}} \leq \|\mathbf{D}_{S \times S}\|_{\mathrm{F}} \leq 1 + \frac{\epsilon}{200}$ where we used that every entry of $\mathbf{S}^\star$ is smaller in magnitude than the corresponding entry in $\mathbf{D}_{S \times S}$ (it is zeroed out if it does not belong to $(T \times T) \cap \Omega^\star$, and otherwise is clipped to a smaller range). This establishes Item 3. The same logic shows that $\|\widehat{\mathbf{D}}_{S' \times S'}\|_{\mathrm{F}} \leq 1 + \frac{\epsilon}{200}$. Combined with (14) and the definition of $T$,

$$\max_{i \in T} \left\|\left[\widehat{\mathbf{D}}_{T \times T}\right]_{i:}\right\|_2^2 \leq \frac{2}{\gamma d} \cdot \frac{\epsilon}{10} = \frac{\epsilon}{5\gamma d}. \tag{15}$$

since $\sum_{i \in L} \|[\widehat{\mathbf{D}}_{T \times T}]_{i:}\|_2^2 \leq \|\widehat{\mathbf{D}}_{S' \times S'}\|_{\mathrm{F}}^2 - \|\widehat{\mathbf{D}}_{T \times T}\|_{\mathrm{F}}^2 \leq \frac{\epsilon}{10}$, and $L$ consisted of the largest rows of $S'$ by $\ell_2$ norm. Therefore, applying Lemma 9 with $\mathbf{A} \leftarrow \widehat{\mathbf{D}}_{(T \times T) \cap \Omega^\star}$, $\tau \leftarrow u$, and $\rho^2 \leftarrow \frac{\epsilon}{5\gamma d}$ establishes Item 4 with probability $\geq 1 - \frac{\delta}{3}$ using the lower bound on $p$. We further have by (14) that

$$\left\|\widehat{\mathbf{D}}_{(T \times T) \cap \Omega^\star}\right\|_{\mathrm{F}} \geq 1 - \frac{\epsilon}{30} - \left\|\widehat{\mathbf{D}}_{\mathcal{L}^\star}\right\|_{\mathrm{F}} \geq 1 - \frac{\epsilon}{30} - d\ell \geq 1 - \frac{\epsilon}{12}.$$

Therefore, Lemma 10 with $v$ set to the vectorized $\widehat{\mathbf{D}}_{(T \times T) \cap \Omega^\star}$, $\tau \leftarrow u$, and $\alpha \leftarrow \frac{\epsilon}{5}$ shows

$$\left\|\widehat{\mathbf{D}}_{(T \times T) \cap \Omega^\star}\right\|_{\mathrm{F}}^2 - \left\langle \widehat{\mathbf{D}}, \widehat{\mathcal{D}}(w^\star) \right\rangle \leq \max\left(\alpha \left\|\widehat{\mathbf{D}}_{(T \times T) \cap \Omega^\star}\right\|_{\mathrm{F}}^2, \frac{3u^2\log\left(\frac{6}{\delta}\right)}{\alpha p}\right) \leq \frac{\epsilon}{4}$$

with probability $\geq 1 - \frac{\delta}{3}$; we again used our lower bound on $p$. Combining the above two displays establishes Item 2. Finally, to see Item 1, fix a row $i \in T$. We apply Lemma 10 once more with $v$ set to the vectorized $[\widehat{\mathbf{D}}_{(T \times T) \cap \Omega^\star}]_{i:}$ and $\alpha = \frac{1}{2}$ to obtain that with probability $\geq 1 - \frac{\delta}{3}$,

$$\left\|\left[\widehat{\mathcal{D}}(w^\star)\right]_{i:}\right\|_2^2 \leq \frac{3}{2}\left\|\left[\widehat{\mathbf{D}}_{(T \times T) \cap \Omega^\star}\right]_{i:}\right\|_2^2 + \frac{6u^2\log(\frac{6}{\delta})}{p} \leq 2 \cdot \frac{\epsilon}{5\gamma d} \leq \frac{\epsilon}{2\gamma d},$$

where we used (15) and our lower bound on $p$. $\qquad\square$

We also provide a helper lemma for making progress if (14) fails to hold.

**Lemma 12.** *In the setting of Lemma 11, if (14) is false, $\left\|\mathbf{D}_{T\times T}^\star\right\|_{\mathrm{F}} \leq 1 - \frac{\epsilon}{36}$.*

*Proof.* This follows from Lemma 3 and the assumption on (14), which lets us bound

$$\left\|\mathbf{D}_{T\times T}^\star\right\|_{\mathrm{F}} \leq \left\|\widehat{\mathbf{D}}_{T\times T}\right\|_{\mathrm{F}} + \left\|\left[\mathbf{D}^\star - \widehat{\mathbf{D}}\right]_{T\times T}\right\|_{\mathrm{F}}$$

$$\leq \left\|\widehat{\mathbf{D}}_{T\times T}\right\|_{\mathrm{F}} + d\left\|\mathbf{N}\right\|_\infty \leq 1 - \frac{\epsilon}{30} + \frac{\epsilon}{200} \leq 1 - \frac{\epsilon}{36}.$$

$\square$

## B.3 Drop steps

In the previous Appendix B.2, we developed Lemmas 11 and 12, which together say that either there exists a good solution compatible with the progress analysis in Lemma 7 (if (14) holds), or there are an $\approx \gamma$ fraction of rows and columns which are responsible for an $\approx \epsilon$ fraction of the Frobenius norm of the difference matrix (if (14) does not hold). In this section, we show in the latter scenario, there is a procedure which explicitly drops a subset of rows and columns, such that on the remainder, a slightly smaller submatrix makes an $\approx \epsilon$ fraction of progress. Importantly, we can ensure that the submatrix parameter (i.e. $\gamma$ in the sense of Definition 4) grows slowly using a tunable parameter, which lets us iterate drop steps without losing too many rows and columns.

We begin by giving a subroutine (Algorithm 1), analogous to Section 4.1 of [43], which takes semi-random observations from an appropriately-bounded difference matrix and explicitly drops a small number of rows and columns, estimated from observations. We show that all remaining rows have few large entries with high probability. Our drop step simply applies this subroutine.

---

**Algorithm 1:** $\mathsf{Sparsify}(\mathcal{O}_{[0,1]}^{\mathrm{sr}}(\mathbf{D}), U, \tau, \gamma, p, \delta)$

---

1 **Input:** $\mathcal{O}_{[0,1]}^{\mathrm{sr}}(\mathbf{D}), U \subseteq [d], \tau \geq 0, \gamma, p, \delta \in (0,1)$
2 $S_0 \leftarrow U$
3 $t_{\max} \leftarrow \lceil 3\log d\rceil + 1$
4 **for** $0 \leq t < t_{\max}$ **do**
5 $\quad\quad \mathbf{D}_t \leftarrow \mathcal{O}_p^{\mathrm{sr}}(\mathbf{D}_{S_t \times S_t})$
6 $\quad\quad$ **for** $i \in S_t$ **do** $r_{i,t} \leftarrow |\{j \in S_t \mid [\mathbf{D}_t]_{ij}$ was observed, and $|[\mathbf{D}_t]_{ij}| \geq \tau\}|$
7 $\quad\quad S_{t+1} \leftarrow S_t \setminus R_t$ where $R_t \subset S_t$ corresponds to the $\lfloor\gamma d\rfloor$ indices $i$ with largest $r_{i,t}$
8 **end**
9 **return** $U' \leftarrow S_{t_{\max}}$

---

**Lemma 13.** *Let $\mathbf{D} \in \mathbb{S}^{d\times d}$, such that there is $A \subseteq U \subseteq [d]$ with $|U\setminus A| \leq \gamma d$, $\left\|\mathbf{D}_{A\times A}\right\|_\infty \leq \tau$. Let $s \in \mathbb{N}$ and $p \geq \frac{800}{s}\log(\frac{d}{\delta})$. Then with probability $\geq 1-\delta$, Algorithm 1 returns $U' \subseteq U$ with $|U \setminus U'| \leq 4\gamma d\log d$, and $\mathbf{D}_{U'\times U'}$ has at most $s$ entries with magnitudes $\geq \tau$ per row or column.*

*Proof.* Let $\mathbf{P} \in [0,1]^{d\times d}$ be the true (symmetric) matrix of reveal probabilities of entries for calls to $\mathcal{O}_p^{\mathrm{sr}}$, following Definition 2. Note that all entries of $\mathbf{P}$ are at least $p$. We define the potential

$$\Phi_t := \sum_{\substack{(i,j)\in S_t\times S_t \\ |[\mathbf{D}_t^\star]_{ij}|\geq\tau}} \mathbf{P}_{ij}.$$

We also denote the expected number of revealed large entries in each row and column as follows:

$$r_{i,t}^\star := \sum_{\substack{j\in S_t \\ |[\mathbf{D}_t^\star]_{ij}|\geq\tau}} \mathbf{P}_{ij} \text{ for all } i \in S_t.$$

By a standard Chernoff bound, with probability $\geq 1-\delta$, for all $0 \leq t < t_{\max}$ and all $i \in S_t$,

$$\left|r_{i,t} - r_{i,t}^\star\right| \leq \max\left(\frac{1}{4}r_{i,t}^\star,\ 50\log\left(\frac{d}{\delta}\right)\right). \tag{16}$$

Condition on this event in the rest of the proof. Due to our symmetric observation model (Definition 2), and since $A \times A$ has no entries $\geq \tau$ in magnitude, in any iteration, at least half of the contribution to $\Phi_t$ must be due to remaining rows in $B_t := S_t \setminus A$, i.e. we have $|B_t| \leq \gamma d$ such that

$$\sum_{i \in B_t} r_{i,t}^\star \geq \frac{\Phi_t}{2}.$$

Letting $R_{i,t}^\star$ denote the $\lfloor \gamma d \rfloor \geq |B_t|$ indices in $S_t$ with largest $r_{i,t}^\star$, we thus have from (16),

$$\sum_{i \in R_t} r_{i,t} \geq \sum_{i \in R_t^\star} r_{i,t} \geq \frac{3}{4} \sum_{i \in R_t^\star} r_{i,t}^\star - 50\gamma d \log\left(\frac{d}{\delta}\right) \geq \frac{3}{8}\Phi_t - 50\gamma d \log\left(\frac{d}{\delta}\right).$$

Therefore, if $\Phi_t \geq 400\gamma d \log(\frac{d}{\delta})$, we decrease $\Phi_t$ by at least a $\frac{1}{4}$ factor, so by our choice of $t_{\max}$, we have $\Phi_{t_{\max}-1} \leq 400\gamma d(\log\frac{d}{\delta}) \leq \frac{\gamma psd}{2}$. This shows that there are at most $\gamma d$ remaining rows $i \in S_{t_{\max}-1}$ with $r_{i,t_{\max}-1}^\star \geq \frac{ps}{2}$. Now, any remaining row $i \in S_{t_{\max}-1}$ with at least $s$ entries with magnitude $\geq \tau$ has $r_{i,t_{\max}-1}^\star \geq ps$, and therefore will be one of the largest $\gamma d$ remaining rows by observed counts via (16). Hence, it will be dropped in the last iteration $t_{\max}$ as claimed.

$\square$

We conclude by showing that the leftover poorly-behaved rows and columns (i.e. those with any large entries) comprise a negligible fraction of the overall indices, using Lemma 8. This shows that by excluding these few indices from our maintained submatrix (after explicitly dropping indices via Algorithm 2), we make an $\approx \epsilon$ fraction of the progress if the condition in Lemma 12 is met.

**Lemma 14.** *Let $\delta, \epsilon, \gamma, p \in (0, \frac{1}{10})$, $\theta, \kappa \geq 1$, and $S \subseteq U \subseteq [d]$ such that $|U \setminus S| \leq \gamma d$. Suppose that for $\mathbf{D}, \mathbf{D}^\star, \mathbf{N} \in \mathbb{S}^{d \times d}$, we have $\mathbf{D} = \mathbf{D}^\star + \mathbf{N}$ for rank-$\hat{r}$ $\mathbf{D}^\star$ with $\left\|\mathbf{D}_{S \times S}^\star\right\|_{\mathrm{F}} \leq 1$ and $\|\mathbf{N}\|_\infty \leq \frac{\epsilon}{d}$, and that there is a subset $T \subseteq S$ with $\|\mathbf{D}_{T \times T}^\star\|_{\mathrm{F}} \leq 1 - \epsilon$ and $|S \setminus T| \leq \frac{\epsilon d}{\theta}$. Then, applying Algorithm 1 with inputs $\mathcal{O}_{[0,1]}^{\mathrm{sr}}(\mathbf{D}), U, \delta$, and*

$$\tau \leftarrow \frac{\sqrt{\theta}}{5d}, \ \gamma \leftarrow \gamma + 10\sqrt{\frac{\hat{r}}{\theta}}, \ p \geq \frac{800\hat{r}\kappa\log(\frac{d}{\delta})}{d}, \tag{17}$$

*returns a subset $U' \subseteq U$ such that with probability $\geq 1 - \delta$, there exists a subset $S' \subseteq U'$ with*

$$|U \setminus U'| \leq 40\left(\gamma + \sqrt{\frac{\hat{r}}{\theta}}\right) d\log d, \ |U' \setminus S'| \leq \left(\gamma + \frac{4}{\sqrt{\theta\kappa}}\right)d, \ and \ \|\mathbf{D}_{S' \times S'}\|_{\mathrm{F}} \leq 1 - \frac{\epsilon}{8}. \tag{18}$$

*Proof.* First, by Lemma 8, there is a subset $A \subseteq S$ such that $|S \setminus A| \leq 10\sqrt{\frac{\hat{r}}{\theta}}$, so that

$$\left\|\mathbf{D}_{A \times A}\right\|_\infty \leq \left\|\mathbf{D}_{A \times A}^\star\right\|_\infty + \|\mathbf{N}\|_\infty \leq \frac{\sqrt{\theta}}{10d} + \frac{\epsilon}{d} \leq \frac{\sqrt{\theta}}{5d}.$$

Therefore, our application of Algorithm 1 with the parameters in (17) satisfies the hypotheses of Lemma 13. Condition on the conclusion of Lemma 13 holding in the rest of the proof (with probability $\geq 1 - \delta$), giving us the first bound in (18). Also, there are at most $\frac{d}{\hat{r}\kappa}$ entries per row and column of $\mathbf{D}_{U' \times U'}$ with magnitude larger than $\tau$ in (17), so there are at most $\frac{d}{\hat{r}\kappa}$ entries per row and column of $\mathbf{D}_{U' \times U'}^\star$ with magnitude larger than $\tau + \frac{\epsilon}{d} \leq 2\tau$.

Next, if $\|\mathbf{D}_{S \times S}^\star\|_{\mathrm{F}} \leq 1 - \frac{\epsilon}{3}$, the conclusion is obvious (taking $S' \leftarrow S \cap U'$). Otherwise, for $C := S \setminus T$,

$$\left\|\mathbf{D}_{C \times S}^\star\right\|_{\mathrm{F}}^2 \geq \frac{1}{2}\left(\left\|\mathbf{D}_{S \times S}^\star\right\|_{\mathrm{F}}^2 - \left\|\mathbf{D}_{T \times T}^\star\right\|_{\mathrm{F}}^2\right) \geq \frac{1}{2}\left(\left(1 - \frac{\epsilon}{3}\right)^2 - (1 - \epsilon)^2\right) \geq \frac{\epsilon}{2}.$$

By Lemma 8, there is a subset $S' \subseteq S \cap U'$ with

$$|(S \cap U') \setminus S'| \leq \frac{4d}{\sqrt{\theta\kappa}}, \ \left\|\mathbf{D}_{S' \times S'}^\star\right\|_\infty \leq \frac{\sqrt{\hat{r}\theta\kappa}}{4d}. \tag{19}$$

This set satisfies the second bound in (18), so it remains to prove the third, which follows from

$$\left\|\mathbf{D}^\star_{S'\times S'}\right\|_F^2 \le \left\|\mathbf{D}^\star_{S\times S}\right\|_F^2 - \left(\left\|\mathbf{D}^\star_{C\times S}\right\|_F^2 - \left\|\mathbf{D}^\star_{(C\cap S')\times S'}\right\|_F^2\right)$$

$$\le 1 - \frac{\epsilon}{2} + \left\|\mathbf{D}^\star_{(C\cap S')\times S'}\right\|_F^2$$

$$\le 1 - \frac{\epsilon}{2} + \frac{\epsilon d}{\theta}\cdot\left(d\cdot(2\tau)^2 + \frac{d}{\hat r\kappa}\cdot\left(\frac{\sqrt{\hat r\theta}\kappa}{4d}\right)^2\right) \le 1 - \frac{\epsilon}{4}.$$

The third line used that there are at most $\frac{\epsilon d}{\theta}$ rows in $C$, and each such row (restricted to $S'$) has all but $\frac{d}{r\kappa}$ entries bounded in magnitude by $\tau$, with the remainder bounded as in (19). $\qquad\square$

### B.4 Analysis of DropOrDescent and SRPartialCompletion

We now combine Lemma 7 and Lemma 14 to give a win-win analysis of a progress step, DropOrDescent. This progress step either executes a descent step (analyzed via Lemma 7), or a drop step (analyzed via Lemma 14); we use Lemmas 11 and 12 to demonstrate correctness of each of these cases. For the runtime analysis, we rely on Proposition 6, the main result of Appendix C, to solve a relevant optimization problem used by our algorithm, but all other parts of the proof are self-contained.

We provide pseudocode for the algorithm corresponding to Proposition 1 in Algorithm 2.

---

**Algorithm 2:** DropOrDescent($\mathcal{O}^{\mathrm{sr}}_{[0,1]}(\mathbf{M}^\star + \mathbf{N}), \mathbf{M}, U, \gamma_{\mathrm{drop}}, \gamma_{\mathrm{sub}}, \Delta, r, r^\star, \delta, \epsilon, \theta, \kappa$)

---

**1 Input:** $\mathcal{O}^{\mathrm{sr}}_{[0,1]}(\mathbf{M}^\star + \mathbf{N})$ where $\mathbf{M}^\star \in \mathbb{S}^{d\times d}$ is rank-$r^\star$ and $\|\mathbf{N}\|_\infty \le \frac{\epsilon\gamma_{\mathrm{sub}}\Delta}{20d}$, $U \subseteq [d]$ with
$|[d]\setminus U| \le \gamma_{\mathrm{drop}}$, rank-$r$ $\mathbf{M}\in\mathbb{S}^{d\times d}$ such that $\mathbf{M}_{U\times U}, \mathbf{M}^\star_{U\times U}$ are $\Delta$-close on a $\gamma_{\mathrm{sub}}$-submatrix,
$\delta, \epsilon \in (0, \frac{1}{10}), \theta, \kappa \ge 1$
**2** $\hat r \leftarrow r + r^\star$
**3** $p_1 \leftarrow \frac{240\hat r\log(\frac{30d}{\delta})}{\gamma_{\mathrm{sub}}^3\epsilon^2 d}, p_2 \leftarrow \frac{800\hat r\kappa\log(\frac{5d}{\delta})}{d}, u \leftarrow \frac{2\sqrt{\hat r}}{\gamma_{\mathrm{sub}}d}, \ell \leftarrow \frac{\epsilon}{20d}$
**4** $\mathcal{I} \leftarrow \mathrm{supp}(\mathcal{O}^{\mathrm{sr}}_{p_1}(\mathbf{M}^\star + \mathbf{N})) \setminus \{\iota \in S(d)\mid |[\mathbf{M}^\star + \mathbf{N} - \mathbf{M}]_\iota| \le \ell\Delta\} \cap (U\times U)$
**5** $v_\iota \leftarrow \Delta^{-2}\cdot[\mathrm{clip}_{u\Delta}(\mathbf{M}^\star + \mathbf{N} - \mathbf{M})]_\iota^2$ for all $\iota \in \mathcal{I}$
**6** $(w, \mathbf{S}) \leftarrow \frac{\epsilon}{12}$-approximate minimizer to (20) with failure probability $\frac{\delta}{5}$, computed using
   Proposition 6
**7** $z \leftarrow$ output of Lemma 4 with $\mathbf{M} \leftarrow \widehat{\mathcal{D}}(w) - \mathbf{S}, q \leftarrow 1, \epsilon \leftarrow \frac{\gamma_{\mathrm{sub}}}{24}, \delta \leftarrow \frac{\delta}{5}$
**8** $V_2 \leftarrow \Delta^{-1}\cdot(z^\top(\widehat{\mathcal{D}}(w) - \mathbf{S})^2 z)^{\frac{1}{2}}$
**9** $V \leftarrow -\langle v, w\rangle + CV_2$
**10 if** $V > -(1 - \frac{11\epsilon}{12})$ **then**
**11** $\quad$ $U' \leftarrow$ Sparsify($\mathcal{O}^{\mathrm{sr}}_{[0,1]}(\frac{1}{\Delta}(\mathbf{M}^\star + \mathbf{N} - \mathbf{M})), U, \frac{\sqrt{\theta}}{5d}, \gamma_{\mathrm{sub}} + 10\sqrt{\frac{\hat r}{\theta}}, p_2, \frac{\delta}{5}$)
**12** $\quad$ **Return:** ("Drop", $U'$)
**13 end**
**14 else**
**15** $\quad$ $\mathbf{Z} \leftarrow$ output of Lemma 4 with $\mathbf{M}\leftarrow\mathbf{S}, q\leftarrow 2\hat r, \epsilon\leftarrow\frac{1}{2}, \delta\leftarrow\frac{\delta}{5}$
**16** $\quad$ **Return:** ("Descent", $\mathbf{M}' \leftarrow \mathbf{M} + \mathbf{ZZ}^\top\mathbf{SZZ}^\top$)
**17 end**

---

**Proposition 1.** *Let $\mathbf{M}\in\mathbb{S}^{d\times d}$ be given as a rank-$r$ factorization, let $\mathbf{M}^\star\in\mathbb{S}^{d\times d}$ be rank-$r^\star$, and let $U\subseteq[d]$. Suppose we know for $\gamma_{\mathrm{drop}}, \gamma_{\mathrm{sub}}\in[0,1], \Delta > 0$, that $|[d]\setminus U| \le \gamma_{\mathrm{drop}}d$, and $\mathbf{M}_{U\times U}, \mathbf{M}^\star_{U\times U}$ are $\Delta$-close on a $\gamma_{\mathrm{sub}}$-submatrix. Let $\delta, \epsilon\in(0, \frac{1}{10}), \theta, \kappa\ge 1$, and suppose for $\hat r := r + r^\star$ and an appropriate polynomial,*

$$p \in \left[\frac{\kappa\hat r}{d}\cdot\mathrm{poly}\left(\frac{\log(\frac{d}{\delta})}{\gamma_{\mathrm{sub}}\epsilon}\right), 1\right], \quad \gamma_{\mathrm{sub}} \le \frac{\epsilon}{36\theta}. \tag{4}$$

*Given one call to $\mathcal{O}^{\mathrm{sr}}_p(\mathbf{M}^\star + \mathbf{N})$ for $\|\mathbf{N}\|_\infty \le \frac{\epsilon\gamma_{\mathrm{sub}}\Delta}{20d}$, the following holds with probability $\ge 1 - \delta$.*

1. *If Algorithm 2 returns on Line 12, then* $\mathbf{M}_{U' \times U'}, \mathbf{M}^{\star}_{U' \times U'}$ *are* $(1 - \frac{\epsilon}{288})\Delta$-*close on a* $\gamma_{\text{sub}} + \frac{4}{\sqrt{\theta\kappa}}$-*submatrix, and* $|U \setminus U'| \le 40(\gamma_{\text{sub}} + (\frac{\hat{r}}{\theta})^{1/2})d \log d.$

2. *If Algorithm 2 returns on Line 16, then* $\mathbf{M}'_{U \times U}$, $\mathbf{M}^{\star}_{U \times U}$ *are* $10\epsilon^{\frac{1}{4}}\Delta$-*close on a* $2\gamma_{\text{sub}}$-*submatrix, and* $\mathbf{M}'$ *is given as a rank-*$(3r + 2r^{\star})$ *factorization.*

*The runtime of the algorithm is*

$$O\left(m \cdot \text{poly}\left(\frac{\hat{r}\log(\frac{d}{\delta})}{\gamma_{\text{sub}}\epsilon}\right)\right),$$

*where* $m \ge d$ *is the number of observed entries upon calling* $\mathcal{O}_p^{\text{sr}}(\mathbf{M}^{\star} + \mathbf{N}).$

*Proof.* Throughout the proof, without loss we let $\Delta = 1$, since we can scale $\mathbf{M}$ by $\frac{1}{\Delta}$, take the claimed step, and scale up by $\Delta$ (indeed, examination of Algorithm 2 shows it performs this scaling). Further, let $\hat{r} := r + r^{\star}$, $\mathbf{D}^{\star} := \mathbf{M}^{\star} - \mathbf{M}$, $\mathbf{D} := \mathbf{D}^{\star} + \mathbf{N}$, and fix

$$u := \frac{2\sqrt{\hat{r}}}{\gamma_{\text{sub}}d}, \; \ell := \frac{\epsilon}{20d}, \; \rho := \frac{\epsilon}{2\gamma_{\text{sub}}d}.$$

Also, let $\widehat{\mathbf{D}} := \text{clip}_u(\mathbf{D})$, let $\Omega^{\star} := \{\iota \in S(d) \mid |[\widehat{\mathbf{D}}]_{\iota}| \le \ell\}$ and let $\mathcal{L}^{\star} := S(d) \setminus \Omega^{\star}$. Note that Lemma 2 shows we can simulate $p_1$-semi-random observation oracle access to $\widehat{\mathbf{D}} := \text{clip}_u(\mathbf{D})$ (as required by Line 4) and the $\le 4\log(d)$ calls to $p_2$-semi-random observation oracle access to $\mathbf{D}$ (as required by Sparsify in Line 11), given access to $\mathcal{O}_p^{\text{sr}}(\mathbf{M}^{\star} + \mathbf{N})$, proving correctness of $p$ in (4).

Let $\mathcal{I} \subseteq S(d)$ correspond to the output of $\mathcal{O}_{p_1}^{\text{sr}}(\mathbf{M} + \mathbf{N})$ in Line 4, after removing indices in $\mathcal{L}^{\star}$ and outside $U \times U$, which we couple to $\Omega \subseteq \mathcal{I}$, the result of calling $\mathcal{O}_{p_1}([\mathbf{M} + \mathbf{N}]_{U \times U})$ (i.e. a $p_1$-semi-random observation oracle where all probabilities are $p_1$), after removing indices in $\mathcal{L}^{\star}$.

We now describe our algorithm. Define $\mathcal{W}_{\rho}$ as in (9), and for $w \in \mathbb{R}^{\mathcal{I}}$, define $\widehat{\mathcal{D}}(w)$ as in (8). Consider the following optimization problem over $\mathcal{W}_{\rho} \times \mathcal{S}$, where $\mathcal{S} := \{\mathbf{S} \in \mathbb{S}^{d \times d} \mid \|\mathbf{S}\|_{\text{F}} \le 1 + \epsilon\}$:

$$\min_{w \in \mathcal{W}_{\rho}, \mathbf{S} \in \mathcal{S}} F(w, \mathbf{S}) := -\langle v, w \rangle + CH_{\eta}^{*}\left(\text{lift}\left(\widehat{\mathcal{D}}(w) - \mathbf{S}\right)\right), \tag{20}$$

$$\text{where } v \in \mathbb{R}^{\mathcal{I}} \text{ has } v_{\iota} = [\widehat{\mathbf{D}}_{\iota}]^2 \text{ for all } \iota \in \mathcal{I}, \; C := \frac{\sqrt{\hat{r}}}{2\gamma_{\text{sub}}}, \; \eta := \frac{\epsilon}{24C\log(2d)}.$$

We compute $(w, \mathbf{S})$, an $\frac{\epsilon}{12}$-approximate minimizer to $F$, and $V$, satisfying $|V - F(w, \mathbf{S})| \le \frac{\epsilon}{12}$. Pseudocode for computing such a $V$ with failure probability $1 - \frac{\delta}{5}$ is provided in Line 9, and the correctness and runtime of this step are analyzed at the end of this proof.

1. If $V > -(1 - \frac{11\epsilon}{12})$, we call Sparsify on (our semi-random access to) the matrix $\mathbf{D} = \mathbf{M}^{\star} + \mathbf{N} - \mathbf{M}$ with parameters $U, \tau \leftarrow \frac{\sqrt{\theta}}{5d}, \gamma \leftarrow \gamma_{\text{sub}} + 10\sqrt{\frac{\hat{r}}{\theta}}, p \leftarrow p,$ and $\delta \leftarrow \frac{\delta}{5}$. We then return the new row subset $U' \subseteq U$. We call this case the "drop" case.

2. Otherwise, we update to $\mathbf{M} + \mathbf{Z}\mathbf{Z}^{\top}\mathbf{S}\mathbf{Z}\mathbf{Z}^{\top}$, where $\mathbf{Z} \in \mathbb{R}^{d \times 2\hat{r}}$ is the output of Lemma 4 with parameters $\delta \leftarrow \frac{\delta}{5}, \epsilon \leftarrow \frac{1}{2}, \mathbf{M} \leftarrow \mathbf{S},$ and $q \leftarrow 2\hat{r}$. We call this case the "descent" case.

We first analyze the drop case. We claim that if $V > -(1 - \frac{11\epsilon}{12})$, it could not be the case that (14) held. To see this, suppose (14) held. Then, with probability $\ge 1 - \frac{\delta}{5}$ (due to our lower bound (4) on $p$), Lemma 11 provides us $(w^{\star}, \mathbf{S}^{\star}) \in \mathcal{W}_{\rho} \times \mathcal{S}$ satisfying

$$\widehat{\mathcal{D}}(w^{\star}) = \mathbf{S}^{\star} + \mathbf{F}^{\star}, \; \langle v, w^{\star} \rangle \ge 1 - \frac{\epsilon}{2}, \text{ for } \|\mathbf{S}^{\star}\|_{\text{F}} \le 1 + \epsilon, \; \|\mathbf{F}^{\star}\|_{\text{op}} \le \frac{\gamma_{\text{sub}}\epsilon}{4\sqrt{\hat{r}}}.$$

In fact, $w^{\star}$ is only supported on $\Omega$, the fully random coordinates. By using Fact 1, we conclude that

$$F(w^{\star}, \mathbf{S}^{\star}) \le -\left(1 - \frac{\epsilon}{2}\right) + C\left(\|\mathbf{F}^{\star}\|_{\text{op}} + \eta\log(2d)\right)$$

$$\le -\left(1 - \frac{\epsilon}{2}\right) + \frac{C\gamma_{\text{sub}}\epsilon}{\sqrt{\hat{r}}} + \frac{\epsilon}{24} \le -\left(1 - \frac{3\epsilon}{4}\right).$$

Therefore, any $\frac{\epsilon}{12}$-approximate minimizer $(w, \mathbf{S})$ to $F$ will satisfy $F(w, \mathbf{S}) \leq -(1 - \frac{5\epsilon}{6})$. Finally, since we assumed our computed value $V$ satisfies $|V - F(w, \mathbf{S})| \leq \frac{\epsilon}{12}$, this contradicts $V > -(1 - \frac{11\epsilon}{12})$ as claimed. Hence, (14) indeed did not hold, so by Lemma 12 and our bounds in (4), the preconditions of Lemma 14 are met with $\epsilon \leftarrow \frac{\epsilon}{36}$, proving correctness of the "drop" case with the stated parameters.

Next, suppose we are in the descent case, meaning we have $(w, \mathbf{S}) \in \mathcal{W}_\rho \times \mathcal{S}$ satisfying $F(w, \mathbf{S}) \leq -(1 - \epsilon)$. By nonnegativity of the second summand in the definition of $F$, this shows

$$\left\langle \widehat{\mathbf{D}}, \widehat{\mathcal{D}}(w) \right\rangle = \langle v, w \rangle \geq 1 - \epsilon.$$

Moreover, because $\langle v, w \rangle \leq d\rho = \frac{\epsilon}{2\gamma}$ by the definition of $\mathcal{W}_\rho$, we have

$$\left\| \widehat{\mathcal{D}}(w) - \mathbf{S} \right\|_{\mathrm{op}} \leq H_\eta^* \left( \mathrm{slift}\left( \widehat{\mathcal{D}}(w) - \mathbf{S} \right) \right) \leq \frac{F(w, \mathbf{S}) + \frac{\epsilon}{2\gamma}}{C} \leq \frac{\epsilon}{\sqrt{\hat{r}}}, \tag{21}$$

where we used Fact 1 in the first inequality. Lemma 7 then yields correctness of the "descent" case.

We next bound the runtime, which is the sum of three costs: the optimization solver, the cost of obtaining an approximate value $V$, and the cost of Lemma 7 in the descent case. It is straightforward to check that in the context of Proposition 6, which bounds the runtime of our optimization solver,

$$k = \frac{\epsilon}{2\gamma_{\mathrm{sub}}}, \ R_\eta = O\left( \frac{\hat{r} \log(d)}{\gamma_{\mathrm{sub}}^3 \epsilon^2} \right), \ \text{and} \ T = O\left( \frac{\hat{r}^2 \log(d)}{\gamma_{\mathrm{sub}}^6 \epsilon^4} \right).$$

Therefore, by our choices of $C, \eta$ from before, the cost of Proposition 6 for $\frac{\delta}{5}$ failure probability is

$$O\left( \left( m + \frac{d\hat{r}^3 \log(d)}{\gamma_{\mathrm{sub}}^8 \epsilon^6} \right) \cdot \frac{\hat{r}^4 \log(d)}{\gamma_{\mathrm{sub}}^{11.5} \epsilon^8} \log^{1.5}\left( \frac{d}{\delta \gamma_{\mathrm{sub}}} \right) \right).$$

Next, consider the cost of obtaining $V$ satisfying $|V - F(w, \mathbf{S})|$. We can explicitly compute the term $\langle v, w \rangle$, and because $\eta C \log(2d) \leq \frac{\epsilon}{24}$, it suffices to obtain an $\frac{\epsilon}{24C} = \frac{\epsilon \gamma_{\mathrm{sub}}}{12\hat{r}^{1/2}}$-additive approximation to $\|\widehat{\mathcal{D}}(w) - \mathbf{S}\|_{\mathrm{op}}$ by Fact 1. The guarantees of Proposition 6 give that $\|\widehat{\mathcal{D}}(w) - \mathbf{S}\|_{\mathrm{op}} \leq \frac{2\epsilon}{\hat{r}^{1/2}}$ (see (21)). Therefore, we may call Lemma 4 with $q \leftarrow 1$, $\epsilon \leftarrow \frac{\gamma_{\mathrm{sub}}}{24}$, and $\delta \leftarrow \frac{\delta}{6}$, in time which does dominate the above, using the rank-$O(\frac{\hat{r}^{2.5} \log(d)}{\gamma_{\mathrm{sub}}^6 \epsilon^5})$ factorization of $\mathbf{S}$ given to us by Proposition 6. Additionally, the runtime of Lemma 7, i.e. the cost of running Lemma 4 with constant $\epsilon$ and $q = 2\hat{r}$, does not dominate Proposition 6. The claimed runtime follows by combining these bounds.

Finally, the failure probability comes from a union bound over Lemma 11, Lemma 14, the subroutine in Proposition 6, and our two calls to Lemma 4, once for the descent step and once to compute the value $V$. We set each of the failure probabilities of these five randomized steps to $\frac{\delta}{5}$. $\square$

By iterating on Proposition 1, we provide our full partial completion method, Algorithm 3.

**Corollary 1.** *Let* $\mathbf{M}, \mathbf{M}^\star \in \mathbb{S}^{d \times d}$ *be rank-$r^\star$, and suppose* $\|\mathbf{M} - \mathbf{M}^\star\|_{\mathrm{F}} \leq \Delta$. *Let* $\alpha \geq 250$ *and* $\delta, \gamma_{\mathrm{tot}} \in (0, \frac{1}{10})$. *Algorithm 3 uses one call to* $\mathcal{O}_p^{\mathrm{sr}}(\mathbf{M}^\star + \mathbf{N})$, *where for appropriate polynomials,*

$$\|\mathbf{N}\|_\infty \leq \frac{\Delta}{d \cdot \mathrm{poly}(\frac{r^\star}{\gamma_{\mathrm{tot}}} \log(\frac{d}{\delta})) \cdot \alpha^{1+o(1)}}, \ p = \frac{\mathrm{poly}(\frac{r^\star}{\gamma_{\mathrm{tot}}} \log(\frac{d}{\delta})) \cdot \alpha^{o(1)}}{d},$$

*and computes a rank-$r^\star \cdot \alpha^{o(1)}$-matrix $\mathbf{M}'$, such that with probability $\geq 1 - \delta$, $\mathbf{M}'$ and $\mathbf{M}^\star$ are $\frac{\Delta}{\alpha}$-close on a $\gamma_{\mathrm{tot}}$-submatrix. The runtime of the algorithm is $m\alpha^{o(1)} \cdot \mathrm{poly}(\frac{r^\star}{\gamma_{\mathrm{tot}}} \log(\frac{d}{\delta}))$, where $m$ is the number of observed entries upon calling $\mathcal{O}_p^{\mathrm{sr}}(\mathbf{M}^\star + \mathbf{N})$.*

*Proof.* We claim that throughout the course of Algorithm 3, the preconditions of Proposition 1 are always met, i.e. $\gamma_{\mathrm{sub}}$ is always an upper bound on the number of excluded rows $U \setminus S$, $\gamma_{\mathrm{drop}}$ is always an upper bound on the number of dropped rows $[d] \setminus U$, and $\mathbf{M}$ always has rank $\hat{r} - r^\star$. To show correctness of the rank bound, the rank $r$ of our iterate only increases on drop steps, where Proposition 1 ensures $r + r^\star$ at most triples, so $\hat{r} - r^\star$ (after the update on Line 11) remains a correct bound on $r$ throughout the algorithm. Also, $\hat{r} \leq \hat{r}_{\mathrm{tot}}$ throughout, since there are $\leq R$ descent steps.

**Algorithm 3:** SRPartialCompletion($\mathcal{O}^{\mathrm{sr}}_{[0,1]}(\mathbf{M}^\star + \mathbf{N}), \mathbf{M}, \Delta, r^\star, \alpha, \gamma_{\mathrm{tot}}, \delta$)

---

**1 Input:** $\mathcal{O}^{\mathrm{sr}}_{[0,1]}(\mathbf{M}^\star + \mathbf{N})$ where $\mathbf{M}^\star \in \mathbb{S}^{d \times d}$, rank-$r^\star$ $\mathbf{M} \in \mathbb{S}^{d \times d}$ such that $\|\mathbf{M} - \mathbf{M}^\star\|_{\mathrm{F}} \leq \Delta$,
$\alpha \geq 250, \gamma_{\mathrm{tot}}, \delta \in (0, \frac{1}{10})$

**2** $N_{\mathrm{drop}} \leftarrow 0, N_{\mathrm{descent}} \leftarrow 0, R \leftarrow \sqrt{\log \alpha}, \epsilon \leftarrow \exp(-8\sqrt{\log \alpha}), K \leftarrow \frac{288\sqrt{\log \alpha}}{\epsilon}$,
$\hat{r}_{\mathrm{tot}} \leftarrow (2r^\star) \cdot 3^R$

**3** $\hat{r} \leftarrow 2r^\star, U \leftarrow [d], \gamma_{\mathrm{temp}} \leftarrow \frac{\epsilon \gamma_{\mathrm{tot}}^2}{10^6 \hat{r}_{\mathrm{tot}} RK \log(d)}, \gamma_{\mathrm{sub}} \leftarrow \frac{\gamma_{\mathrm{temp}}}{160 RK \log(d)} \cdot 2^{-R}, \gamma_{\mathrm{drop}} \leftarrow 0$

**4** $\theta \leftarrow \frac{25600 \hat{r}_{\mathrm{tot}} (RK \log(d))^2}{\gamma_{\mathrm{tot}}^2}, \kappa \leftarrow \frac{16\gamma_{\mathrm{tot}}^2 R^2 K^2 4^R}{\hat{r}_{\mathrm{tot}} \gamma_{\mathrm{temp}}^2}$

**5 while** $N_{\mathrm{drop}} < RK$ **and** $N_{\mathrm{descent}} < R$ **do**

**6**     **if** DropOrDescent($\mathcal{O}^{\mathrm{sr}}_{[0,1]}(\mathbf{M}^\star + \mathbf{N}), \mathbf{M}, U, \gamma_{\mathrm{drop}}, \gamma_{\mathrm{sub}}, \Delta, \hat{r} - r^\star, r^\star, \frac{\delta}{R(K+1)}, \epsilon, \theta, \kappa) =$
      ("Drop", $U'$) **then**

**7**       $U \leftarrow U', \gamma_{\mathrm{drop}} \leftarrow \gamma_{\mathrm{drop}} + \frac{\gamma_{\mathrm{tot}}}{2RK}, \Delta \leftarrow \exp(-\frac{\epsilon}{288})\Delta, N_{\mathrm{drop}} \leftarrow N_{\mathrm{drop}} + 1$

**8**     **end**

**9**     **else**

**10**       ("Descent", $\mathbf{M}'$) $\leftarrow$
      DropOrDescent($\mathcal{O}^{\mathrm{sr}}_{[0,1]}(\mathbf{M}^\star + \mathbf{N}), \mathbf{M}, U, \gamma_{\mathrm{drop}}, \gamma_{\mathrm{sub}}, \Delta, \hat{r} - r^\star, r^\star, \frac{\delta}{R(K+1)}, \epsilon, \theta, \kappa)$

**11**       $\hat{r} \leftarrow 3\hat{r}, \gamma_{\mathrm{sub}} \leftarrow 2\gamma_{\mathrm{sub}}, \mathbf{M} \leftarrow \mathbf{M}', \Delta \leftarrow 10\epsilon^{\frac{1}{4}}\Delta, N_{\mathrm{descent}} \leftarrow N_{\mathrm{descent}} + 1$

**12**     **end**

**13 end**

**14 Return:** $\mathbf{M}$

---

We next prove correctness of the $\gamma_{\mathrm{sub}}$ parameter throughout the algorithm. By Proposition 1, this parameter doubles each time Algorithm 2 runs a descent step, which is correctly handled by Line 11. Moreover, the only other time $\gamma_{\mathrm{sub}}$ increases is by $\frac{4}{\sqrt{\theta\kappa}}$ each time Algorithm 2 runs a drop step, via the first part of Proposition 1. Note that the algorithm terminates once $RK$ drop steps or $R$ descent steps have run, and by our choices of $\theta, \kappa$, we have

$$\frac{4}{\sqrt{\theta\kappa}} \cdot RK \leq \frac{\gamma_{\mathrm{temp}}}{160 RK \log(d)} \cdot 2^{-R},$$

which is our initial setting of $\gamma_{\mathrm{sub}}$ in Line 3. Therefore, since the worst-case growth of the $\gamma_{\mathrm{sub}}$ parameter over $RK$ drop steps and $R$ descent steps is if we encounter all of the drop steps first (and then repeatedly double $R$ times), the value of $\gamma_{\mathrm{sub}}$ is correct throughout the algorithm.

Next, we show that the $\gamma_{\mathrm{drop}}$ parameter remains correct throughout the algorithm. Note that the $\gamma_{\mathrm{sub}}$ parameter is never larger than $\frac{\gamma_{\mathrm{temp}}}{160 RK \log(d)} \leq \frac{\gamma_{\mathrm{tot}}}{160 RK \log(d)}$. Therefore, the number of dropped rows in each drop step, by Proposition 1 and our choice of $\theta$, is at most

$$40 \left( \frac{\gamma_{\mathrm{tot}}}{160 RK \log d} + \sqrt{\frac{\hat{r}_{\mathrm{tot}}}{\theta}} \right) d \log d \leq \left( \frac{\gamma_{\mathrm{tot}}}{4RK} + \frac{\gamma_{\mathrm{tot}}}{4RK} \right) \cdot d = \frac{\gamma_{\mathrm{tot}} d}{2RK}.$$

Therefore the update to $\gamma_{\mathrm{drop}}$ in Line 7 is correct. We have also shown that over the course of the algorithm, we never drop or exclude more than $\frac{\gamma_{\mathrm{tot}} d}{160 RK \log(d)} + \frac{\gamma_{\mathrm{tot}} d}{2} \leq \gamma_{\mathrm{tot}} d$ rows in total. Additionally, we verify the second condition in (4) holds with the largest value of $\gamma_{\mathrm{sub}}$:

$$\frac{\gamma_{\mathrm{temp}}}{160 RK \log(d)} \leq \frac{\epsilon}{36\theta}.$$

Finally, after $RK$ drop steps Proposition 1 shows that we have made an $\exp(-\frac{\epsilon}{288} \cdot RK) = \frac{1}{\alpha}$ factor progress on the closeness parameter $\Delta$. Similarly, after $R$ descent steps, since $10\epsilon^{\frac{1}{4}} \leq \epsilon^{\frac{1}{8}}$ by our condition $\alpha \geq 250$, we have made an $\epsilon^{\frac{R}{8}} = \frac{1}{\alpha}$ factor progress as well. The failure probability comes from union bounding over the at most $R(K+1)$ calls to DropOrDescent.

For the runtime bound, by observation all of the parameters $\hat{r}, \gamma_{\mathrm{sub}}^{-1}, \epsilon^{-1}$ in Proposition 1 are bounded by a polynomial in $r^\star \cdot \alpha^{o(1)}$ throughout the algorithm. The correctness of the lower bound on $p$ and the upper bound on $\|\mathbf{N}\|_\infty$ similarly follow from Proposition 1. $\qquad\square$

## C Computing adaptive reweightings

In this section, we develop an optimization subroutine for solving the subproblems (20) arising in our partial matrix completion method. We use an approximate Frank-Wolfe framework to reduce our optimization to a sequence of linear optimization problems over our constraint set, which we characterize as a matching polytope over a bipartite graph. We show that for the vector block of our variable, we can cast these linear optimization problems as computing maximum weight bipartite matchings, as explained in Appendix C.4. To efficiently implement the matrix block steps, we apply standard tools from the numerical linear algebra and approximation theory literature in Appendix C.3. We now define the problem we study in a self-contained way.

Throughout this section, we fix a subset $\mathcal{I} \subset S(d)$ and parameters $k \in \mathbb{R}_{>0}$, $\epsilon \in (0,1)$, and define:

$$\mathcal{X} := \left\{ x \in \mathbb{R}^{\mathcal{I}}_{\geq 0} \mid \max_{i \in [d]} \sum_{j \in [d] \mid \iota = (i,j)^{\sharp} \in \mathcal{I}} x_{\iota} \leq \frac{k}{d} \right\},$$
$$\mathcal{S} := \left\{ \mathbf{S} \in \mathbb{S}^{d \times d} \mid \|\mathbf{S}\|_{\mathrm{F}} \leq 1 + \epsilon \right\}, \; \mathcal{Z} := \mathcal{X} \times \mathcal{S}. \tag{22}$$

In other words, $\mathcal{S}$ is the set of "short" (Frobenius-norm bounded) matrices. Further, if $x \in \mathcal{X}$ is viewed as a symmetric $d \times d$ matrix, the constraints on $\mathcal{X}$ enforce that $x$ has bounded row and column sums. Clearly $x \in \mathcal{X} \implies \|x\|_1 \leq k$, so $\mathcal{X}$ is a subset of an $\ell_1$ ball of radius $k$. However, we can more specifically capture $\mathcal{X}$ as a scaled bipartite matching polytope.

**Fact 3.** *Let $\mathcal{I} \subset S(d)$ be identified with a $d \times d$ bipartite graph $G = (L \cup R, E)$, where for $(i,j) \in [d] \times [d]$, we place an edge from the $i^{th}$ vertex in $L$ (denoted $u_i$) to the $j^{th}$ vertex in $V$ (denoted $v_j$) if and only if $(i,j)^{\sharp} \in S(d)$. Let $\mathcal{M}$ be the bipartite matching polytope of $G$, i.e.*

$$\mathcal{M} := \left\{ f \in \mathbb{R}^{E}_{\geq 0} \mid \max_{u_i \in L} \sum_{v_j \in R \mid (u_i, v_j) \in E} f_{(u_i, v_j)} \leq 1 \right\}. \tag{23}$$

*Then $\frac{d}{k} \mathcal{X} = \mathcal{M}$ for $\mathcal{X}$ in (22), where we map coordinates in $\mathcal{I}$ to edges in $E$ in the canonical way.*

By the characterization in Fact 3, we efficiently implement iterations of our Frank-Wolfe method by calling an approximate maximum weight matching algorithm by [22]. We also use the structure of the matching polytope to prove a key regularity property on our objective in Appendix C.2. We finally require the definition of a bounded submatrix, whose entries lie in a given range.

**Definition 5** (Bounded submatrix). *Let $\mathbf{D} \in \mathbb{S}^{d \times d}$ and $\mathcal{I} \subset S(d)$. We say that $(\mathbf{D}, \mathcal{I})$ is an $(\ell, u)$-bounded submatrix if for all $\iota \in \mathcal{I}$, $|[\mathbf{D}]_{\iota}| \in [\ell, u]$.*

When $\mathbf{D}$ and $\mathcal{I}$ are clear from context, we let $\mathcal{D}$ be the operator on $\mathbb{R}^{\mathcal{I}}$ such that

$$\mathcal{D}(w) := \sum_{\iota \in \mathcal{I}} w_{\iota} \mathbf{D}_{\iota}. \tag{24}$$

With these definitions in hand, our goal in this section is to optimize the following jointly convex function over $(x, \mathbf{S}) \in \mathcal{Z}$, for an $(\ell, u)$-bounded submatrix $(\mathbf{D}, \mathcal{I})$ and parameters $C, \eta \in \mathbb{R}_{>0}$, named $F_{\text{prog-flat}}$ to denote its decomposition into progress and (approximate) flatness components:

$$F_{\text{prog-flat}}(x, \mathbf{S}) := - \langle \mathbb{1}_{\mathcal{I}}, x \rangle + C H^{*}_{\eta} \left( \text{slift} \left( \mathcal{D} \left( \frac{x}{v} \right) - \mathbf{S} \right) \right),$$
where $v_{\iota} = [\mathbf{D}]^2_{\iota}$ for all $\iota \in \mathcal{I}$, and $\frac{x}{v}$ denotes entrywise division. \tag{25}

Note that (25) is exactly the problem in (20), up to reparameterizing $x \leftarrow w \circ v$. It may be helpful to think of the parameter settings for (25) in the context (20), e.g., $C \approx \frac{\sqrt{r}}{\gamma}, \eta \approx \frac{1}{C}, u \approx \frac{\sqrt{r}}{d}, \ell \approx \frac{1}{d}, k \approx \frac{1}{\gamma}$, where $r$ and $\gamma$ are the rank and closeness parameter of a current iterate.

### C.1 Approach based on the Frank-Wolfe method

To efficiently minimize $f = F_{\text{prog-flat}}$ (defined in (25)) over $\mathcal{Z}$ (defined in (22)) we apply a variant of Frank-Wolfe or conditional gradient methods due to [35]. This method essentially reduces the

problem to approximately solving $\min_{z \in \mathcal{Z}} \langle \nabla f(q), z \rangle$ for different values of $q \in \mathcal{Z}$. The algorithm is given below as Algorithm 4.[10] Below we define the curvature constant, the key quantity which bounds the convergence of the method, as well as the main theorem from [35] which we use.

---

**Algorithm 4:** ApproxFrankWolfe$(f, \mathcal{D}, T, \delta)$ (Algorithm 2 of [35] restated)

---

1 **Input:** Differentiable, convex $f : \mathbb{R}^d \to \mathbb{R}$ and compact, convex $\mathcal{D} \subseteq \mathbb{R}^d$ such that the curvature constant of $f$ with respect to $\mathcal{D}$ is $C_f$, iteration count $K \geq 1$ and approximation tolerance $\delta \geq 0$
2 Compute arbitrary $z^{(0)} \in \mathcal{D}$
3 **for** $t = 0, \ldots, T - 1$ **do**
4      $\gamma^{(t)} \leftarrow \frac{2}{t+2}$
5      Compute $s^{(t)} \in \mathcal{D}$ such that $\left\langle \nabla f(z^{(t)}), s^{(t)} \right\rangle \leq \min_{s \in \mathcal{D}} \left\langle \nabla f(z^{(t)}), s \right\rangle + \frac{\delta \gamma^{(t)} C_{f,\mathcal{D}}}{2}$
6      $z^{(t+1)} \leftarrow (1 - \gamma^{(t)}) z^{(t)} + \gamma^{(t)} s^{(t)}$
7 **end**
8 **return** $z^{(T)}$

---

**Definition 6** (Curvature constant (restated from [35])). *The* curvature constant $C_{f,\mathcal{D}}$ *of convex and differentiable* $f : \mathbb{R}^d \to \mathbb{R}$ *with respect to compact domain* $\mathcal{D} \subseteq \mathbb{R}^d$ *is*

$$C_{f,\mathcal{D}} := \sup_{z_0, z_1 \in \mathcal{D}, \gamma \in [0,1], z_\gamma = z_0 + \gamma(z_1 - z_0)} \frac{2}{\gamma^2} \left( f(z_\gamma) - f(z_0) - \langle \nabla f(z_0), z_\gamma - z_0 \rangle \right).$$

**Proposition 4** (Approximate Frank-Wolfe, Theorem 1 of [35] restated and specialized). *Let* $f : \mathbb{R}^d \to \mathbb{R}$ *be convex and differentiable, let* $\mathcal{D} \subseteq \mathbb{R}^d$ *be compact and convex, and let* $z^\star \in \operatorname{argmin}_{z \in \mathcal{D}} f(z)$. *For any* $K \geq 1$ *and* $\delta \geq 0$, $z^{(T)} = $ ApproxFrankWolfe$(f, \mathcal{D}, T, \delta)$ *(Algorithm 4) satisfies*

$$f(z^{(T)}) - f(z^\star) \leq \frac{2(1 + \delta) C_{f,\mathcal{D}}}{T + 2}.$$

To facilitate applying Proposition 4 to efficiently minimize $F_{\text{prog-flat}}$ (25) over $\mathcal{Z}$ (22) we provide two general lemmas. The first is Lemma 15 below, which implies that to implement Line 5 of Algorithm 4 it suffices to compute approximate gradients and and then approximately solve constrained linear optimization problems for these approximate gradients.

**Lemma 15** (Gradient approximation). *For compact* $\mathcal{D} \subseteq \mathbb{R}^d$, $g, \hat{g} \in \mathbb{R}^d$, *and* $\Delta, M \geq 0$ *suppose that*

$$|\langle g - \hat{g}, s \rangle| \leq \Delta \text{ and } |\langle g, s \rangle| \leq M \text{ for all } s \in \mathcal{D}$$

*If* $\langle \hat{g}, \hat{s} \rangle \leq (1 - \alpha) \min_{x \in \mathcal{D}} \langle \hat{g}, s \rangle + \Delta$ *for* $\hat{s} \in \mathcal{D}$ *and* $\alpha \in [0, 1]$ *then*

$$\langle g, \hat{s} \rangle \leq \min_{x \in \mathcal{D}} \langle g, s \rangle + \alpha M + 3\Delta.$$

*Proof.* For $s^\star \in \operatorname{argmin}_{s \in \mathcal{D}} \langle g, s \rangle$ the assumptions imply that

$$\langle g, s \rangle = \langle \hat{g}, s \rangle + \langle \hat{g} - g, s \rangle \leq (1 - \alpha) \langle \hat{g}, s^\star \rangle + \Delta + \Delta.$$

The result then follows as

$$(1 - \alpha) \langle \hat{g}, s^\star \rangle = (1 - \alpha) \langle g, s^\star \rangle + (1 - \alpha) \langle g - \hat{g}, s^\star \rangle \leq \langle g, s^\star \rangle + \alpha M + \Delta.$$

$\square$

The second lemma is the following straightforward upper bound on the curvature constant (that is perhaps related to the relationship to smoothness given in Lemma 7 of [35]). It shows that to upper bound the curvature constant it suffices to upper bound $\nabla^2 f(z)[w, w]$ for all $w, z \in \mathcal{D}$.

**Lemma 16** (Curvature bound). *For convex, twice-differentiable* $f : \mathbb{R}^d \to \mathbb{R}$ *and compact* $\mathcal{D} \subseteq \mathbb{R}^d$ *if* $\nabla^2 f(z)[w, w] \leq M$ *for all* $w, z \in \mathcal{D}$ *then the curvature constant* $f$ *with respect to* $\mathcal{D}$ *is at most* $2M$.

---

[10]We perform minor changes to the names of the variables, choice of offsets, etc. to facilitate and simplify the use of [35] to obtain our results.

*Proof.* Let $z_0, z_1 \in \mathcal{Z}$ and $\gamma \in [0,1]$ and define $z_\gamma := z_0 + \gamma(z_1 - z_0)$ for all $\gamma \in [0,1]$. Note that

$$f(z_\gamma) - f(z_0) - \langle \nabla f(z_0), z_\gamma - z_0 \rangle = -\gamma \langle \nabla f(z_0), z_1 - z_0 \rangle + \int_0^\gamma \langle \nabla f(z_t), z_1 - z_0 \rangle \, \mathrm{d}t$$

$$= \int_0^\gamma \langle \nabla f(z_t) - \nabla f(z_0), z_1 - z_0 \rangle \, \mathrm{d}t.$$

Additionally,

$$\langle \nabla f(z_t) - \nabla f(z_0), z_1 - z_0 \rangle = \int_0^t \nabla^2 f(z_\alpha)[z_1 - z_0, z_1 - z_0] \mathrm{d}\alpha$$

$$\leq t \cdot \sup_{z \in \mathcal{D}} \nabla^2 f(z)[z_1 - z_0, z_1 - z_0].$$

Now, since $f$ is convex, $\nabla^2 f(x)$ is positive semidefinite for all $x \in D$ and so

$$\nabla^2 f(z)[z_1 - z_0, z_1 - z_0] \leq 2\nabla^2 f(z)[z_1, z_1] + 2\nabla^2 f(z)[z_0, z_0] \leq 2M.$$

Combining these bounds yields the desired bound of

$$f(z_\gamma) - f(z_0) - \langle \nabla f(z_0), z_\gamma - z_0 \rangle \leq \int_0^\gamma 2Mt\mathrm{d}t = M\gamma^2.$$

$\square$

In the following Appendix C.2 we bound the curvature constant of the particular $f = F_{\text{prog-flat}}$ defined in (24) over $\mathcal{Z}$ and provide additional regularity bounds on this $f$ over $\mathcal{Z}$. Then in Appendices C.3 and C.4, we discuss the implementation of a single step of Algorithm 5 (specifically, Line 5 by satisfying the conditions needed to apply Lemma 15). To fix some notation, we let $z_t := (x_t, \mathbf{S}_t)$ be the $t^{\text{th}}$ iterate of Algorithm 5 ($z^{(t)}$ in the code). We also let

$$\mathbf{Y}_t := \frac{\exp\left(\frac{1}{\eta}\text{slift}\left(\mathcal{D}\left(\frac{x_t}{v}\right) - \mathbf{S}_t\right)\right)}{\text{Tr}\exp\left(\frac{1}{\eta}\text{slift}\left(\mathcal{D}\left(\frac{x_t}{v}\right) - \mathbf{S}_t\right)\right)}$$

be the gradient of $H_\eta^*(\text{slift}(\mathcal{D}(\frac{x}{v}) - \mathbf{S}))$ before applying the chain rule, see Fact 1. Because of the block-separable structure of slift, we have that

$$\mathbf{Y}_t = \frac{1}{\text{Tr}(\mathbf{L}_t) + \text{Tr}(\mathbf{R}_t)} \begin{pmatrix} \mathbf{L}_t & \mathbf{0}_{d \times d} \\ \mathbf{0}_{d \times d} & \mathbf{R}_t \end{pmatrix},$$

$$\text{where } \mathbf{L}_t := \exp\left(\frac{1}{\eta}\left(\mathcal{D}\left(\frac{x_t}{v}\right) - \mathbf{S}_t\right)\right), \ \mathbf{R}_t := \exp\left(\frac{1}{\eta}\left(\mathbf{S}_t - \mathcal{D}\left(\frac{x_t}{v}\right)\right)\right). \tag{26}$$

For notational simplicity, we finally define

$$\mathbf{B}_t := \mathbf{L}_t - \mathbf{R}_t, \ T_t := \text{Tr}(\mathbf{L}_t) + \text{Tr}(\mathbf{R}_t).$$

We observe that by definition of the objective in (25), we have

$$[\nabla_x f(z_t)]_\iota = -1 + \frac{C}{T_t}\langle \mathbf{B}_t, \mathbf{E}_\iota \rangle \text{ for all } \iota \in \mathcal{I}, \ \nabla_{\mathbf{S}} f(z_t) = \frac{C}{T_t}\mathbf{B}_t.$$

Finally, $g_t := (h_t, \mathbf{H}_t)$ denotes the blocks of the approximate gradient computed in iteration $t$ and $v_t := (w_t, \mathbf{W}_t)$ denotes the blocks of the approximate linear objective optimizer in iteration $t$, as suffice for applying Lemma 15. Throughout Appendices C.3 and C.4, we fix a single iteration $t$ and drop $t$ from all subscripts, to decrease notational clutter.

## C.2 Curvature constants and regularity

Here we bound the curvature constant of $F_{\text{prog-flat}}$ as defined in (25) over $\mathcal{Z}$ defined in (22). To do so, we first provide a bound, Lemma 17, on the operator norm of $\mathcal{D}(x)$ defined in (24), for $x \in \mathcal{X}$. The proof follows from a previously established fact that the operator norm of any matrix (even if rectangular) is upper bounded by the maximum $\ell_1$ norm of any its rows or columns (see e.g., Lemma B.4, [19]); however, we directly give a self-contained proof for convenience.

**Lemma 17.** *For all $x \in \mathcal{X}$ (defined in (22)), we have $\|\mathcal{D}(x)\|_{\mathrm{op}} \le \frac{ku}{d}$ for $\mathcal{D}$ as defined in (24).*

*Proof.* For all $z \in \mathbb{R}^d$, the Cauchy-Schwarz inequality and the fact that $\mathcal{D}(x)$ is symmetric imply

$$
\left| z^\top \mathcal{D}(x) z \right| = \left| \sum_{i,j \in [d]} [\mathcal{D}(x)]_{ij}\, z_i z_j \right| \le \left( \sum_{i,j \in [d]} z_i^2 \left| [\mathcal{D}(x)]_{ij} \right| \right)^{\frac{1}{2}} \left( \sum_{i,j \in [d]} z_j^2 \left| [\mathcal{D}(x)]_{ij} \right| \right)^{\frac{1}{2}}
$$

$$
= \sum_{i \in [d]} z_i^2 \sum_{j \in [d]} \left| [\mathcal{D}(x)]_{ij} \right|.
$$

Additionally, for all $i \in [d]$ since $(\mathbf{D}, \mathcal{I})$ is an $(\ell, u)$-bounded submatrix and $x \in \mathcal{X}$,

$$
\sum_{j \in [d]} \left| [\mathcal{D}(x)]_{ij} \right| | = \sum_{j \in [d]\,|\,\iota=(i,j)^\sharp \in \mathcal{I}} \left| [\mathcal{D}(x)]_{ij} \right| \le \frac{ku}{d}.
$$

Combining shows that $|z^\top \mathcal{D}(x) z| \le \frac{ku}{d} \|z\|_2^2$ for all $z \in \mathbb{R}^d$ and yields the result as $\mathcal{D}(x)$ is symmetric. $\qquad \square$

Lemma 17 improves upon the trivial bound on $\|\mathcal{D}(x)\|_{\mathrm{op}}$ for $x$ with $\|x\|_1 \le k$. Note that if $x$ has $k$ on the index corresponding to the largest entry of $\mathbf{D}_\mathcal{I}$, then $\|\mathcal{D}(x)\|_{\mathrm{op}} \ge ku$, which is substantially larger than $\frac{ku}{d}$ bound given in Lemma 17. The improved bound for $x \in \mathcal{X}$ that we show in Lemma 17 is key to obtaining bounds on the curvature constant of $F_{\mathrm{prog\text{-}flat}}$ over $\mathcal{Z}$ which are sufficient for obtaining our desired runtimes (i.e., scaling polynomially only in $r^\star$, not $d$). We provide the curvature bound as well as different bounds on the regularity of $F_{\mathrm{prog\text{-}flat}}$ over $\mathcal{Z}$ below.

**Lemma 18.** *For all $z, z' \in \mathcal{Z}$ with $z = (x, \mathbf{S})$ for $x \in \mathcal{X}$ and $\mathbf{S} \in \mathcal{S}$ as defined in (22) and for $f = F_{\mathrm{prog\text{-}flat}}$ defined in (25),*

$$
|\langle \nabla_\mathcal{X} f(z'), x \rangle| \le C \frac{ku}{\ell^2 d} + k,\ |\langle \nabla_\mathcal{S} f(z'), \mathbf{S} \rangle| \le 2C,\ \text{and}\ \nabla^2 f(z')[z, z] \le \frac{C}{\eta} \left( 2 + \frac{ku}{\ell^2 d} \right)^2. \quad (27)
$$

*Consequently, $C_{F_{\mathrm{prog\text{-}flat}}, \mathcal{Z}} \le \frac{2C}{\eta} \left( 2 + \frac{ku}{\ell^2 d} \right)^2.$*

*Proof.* Let $z, z' \in \mathcal{Z}$ be arbitrary. Let $\mathbf{A}$ be the linear operator such that $\mathbf{A}(x, \mathbf{S}) = \mathrm{slift}\left( \mathcal{D}\left( \frac{x}{v} \right) - \mathbf{S} \right)$. Let $\mathbf{G}$ be the gradient of $H_\eta^*$ at $\mathbf{A}z'$ and let $\mathbf{H}$ be the Hessian of $H_\eta^*$ at $\mathbf{A}z'$. Note that the chain rule, Fact 1, and that the trace and operator norm are dual imply that, for $z = (x, \mathbf{S})$,

$$
|\langle \nabla f(z'), z \rangle| = |- \langle \mathbb{1}_\mathcal{I}, x \rangle + C \langle \mathbf{G}, \mathbf{A}z \rangle| \le \|x\|_1 + C \|\mathbf{G}\|_{\mathrm{tr}} \|\mathbf{A}z\|_{\mathrm{op}} \le k + C \|\mathbf{A}z\|_{\mathrm{op}},
$$

$$
\nabla^2 f(z')[z, z] = C \mathbf{H}[\mathbf{A}z, \mathbf{A}z] \le \frac{C}{\eta} \|\mathbf{A}z\|_{\mathrm{op}}^2.
$$

However, for any $(x, \mathbf{S}) \in \mathcal{Z} = \mathcal{X} \times \mathcal{S}$,

$$
\left\| \mathrm{slift}\left( \mathcal{D}\left( \frac{x}{v} \right) - \mathbf{S} \right) \right\|_{\mathrm{op}} \le \left\| \mathcal{D}\left( \frac{x}{v} \right) \right\|_{\mathrm{op}} + 1 + \epsilon \le \frac{ku}{\ell^2 d} + 2, \quad (28)
$$

where the first inequality used the triangle inequality and that the operator norm is bounded by the Frobenius norm, and the second used the range of $v$, that $\epsilon \in (0, 1)$, and Lemma 17. Combining these inequalities yields $\nabla^2 f(z')[z, z] \le \frac{C}{\eta} \left( 2 + \frac{ku}{\ell^2 d} \right)^2$. Applying the same argument with $\mathbf{S} = \mathbf{0}$ and $x = 0$ yields $|\langle \nabla_\mathcal{X} f(z'), x \rangle| \le C \frac{ku}{\ell^2 d} + k$ and $|\langle \nabla_\mathcal{S} f(z'), \mathbf{S} \rangle| \le 2C$. Finally, $C_{F_{\mathrm{prog\text{-}flat}}, \mathcal{Z}} \le \frac{2C}{\eta} \left( 2 + \frac{ku}{\ell^2 d} \right)^2$ follows from Lemma 16. $\qquad \square$

## C.3 Matrix step implementation

In this section, we discuss how to approximate the $\mathcal{S}$ block of $\nabla F_{\text{prog-flat}}(z) := \nabla F_{\text{prog-flat}}(x, \mathbf{S})$ within the error allowed by Line 5 and Lemma 15, where (as discussed earlier) we denote our approximation by

$$\mathbf{H} \approx \nabla_{\mathbf{S}} f(x, \mathbf{S}) = \frac{C}{T} \mathbf{B}, \text{ for } T = \text{Tr}(\mathbf{L}) + \text{Tr}(\mathbf{R}), \ \mathbf{B} = \mathbf{L} - \mathbf{R},$$

following notation in (26). We then give implementation details for computing

$$\mathbf{W} \approx \min_{\mathbf{M} \in \mathcal{S}} \langle \mathbf{H}, \mathbf{M} \rangle,$$

within the error threshold required by Line 5. Our approximately-optimal step matrix $\mathbf{W}$ will have the convenient property of coming with an explicit low-rank representation, facilitating faster implementation of future iterations of our subproblem solver. We will give approximation guarantees in a mixed multiplicative-additive format, whose conversion to additive error is facilitated by Lemma 15.

The main result of this section is the following.

**Lemma 19** ($\mathcal{S}$ step)**.** *There is an algorithm which takes as input $x \in \mathcal{X}$, $\mathbf{S} \in \mathcal{S}$ given as a rank-$r$ factorization, $R \geq \|\frac{1}{\eta}(\mathcal{D}(\frac{x}{v}) - \mathbf{S})\|_{\text{op}}$, $\Delta \in [0, C]$, and $\alpha, \delta \in (0, 1)$, and computes $\mathbf{W} \in \mathcal{S}$ as a rank-$q$ factorization satisfying*

$$\langle \mathbf{W}, \mathbf{H} \rangle \leq (1 - \alpha) \text{argmin}_{\mathbf{W}^{\star} \in \mathcal{S}} \langle \mathbf{W}^{\star}, \mathbf{H} \rangle + \Delta, \tag{29}$$

$$\text{for } \mathbf{H} \in \mathbb{S}^{d \times d} \text{ with } |\langle \mathbf{H} - \nabla_{\mathbf{S}} F_{\text{prog-flat}}(x, \mathbf{S}), \mathbf{S}' \rangle| \leq \Delta \text{ for all } \mathbf{S}' \in \mathcal{S}, \tag{30}$$

*with probability $\geq 1 - \delta$. Moreover, $q = O(\frac{C^2}{\Delta^2})$, and the algorithm runs in time*

$$O\left( (|\mathcal{I}| + dr) \cdot R\sqrt{\log \frac{C}{\Delta}} \cdot \left( \frac{1}{\alpha^2} + \frac{C^2}{\Delta^2} \right) \log \left( \frac{d}{\delta} \right) \right).$$

**Approximating the gradient.** We begin by bounding the complexity of matrix-vector access to a matrix $\widetilde{\mathbf{B}} \approx \mathbf{B} - \mathbf{L} - \mathbf{R}$, with a given tolerance for approximation error. Ultimately we will use the approximate gradient

$$\mathbf{H} \leftarrow \frac{C}{T} \widetilde{\mathbf{B}} \tag{31}$$

to implement the update to $\mathbf{S}$ in our algorithm. Our access is based on a standard polynomial approximation to the exponential which is well-known in the literature.

**Lemma 20** ([2])**.** *For $\alpha \geq 0$ and $R \geq 0$, there is a polynomial $p$ of degree $O(R\sqrt{\log \frac{1}{\alpha}})$ such that $|\exp(x) - p(x)| \leq \alpha \exp(x)$ for all $x \in [-R, R]$.*

Applying Lemma 20 twice directly yields our approximation $\widetilde{\mathbf{B}}$.

**Corollary 2.** *Let $x \in \mathcal{X}$, $\mathbf{S} \in \mathcal{S}$ satisfy $\|\frac{1}{\eta}(\mathcal{D}(\frac{x}{v}) - \mathbf{S})\|_{\text{op}} \leq R$, and let $\Delta \geq 0$. There is a matrix $\widetilde{\mathbf{B}}$ such that, for $\mathbf{L} := \exp(\frac{1}{\eta}(\mathcal{D}(\frac{x}{v}) - \mathbf{S}))$ and $\mathbf{R} := \exp(\frac{1}{\eta}(\mathbf{S} - \mathcal{D}(\frac{x}{v})))$, and defining*

$$T := \text{Tr}(\mathbf{L}) + \text{Tr}(\mathbf{R}), \ \mathbf{B} := \mathbf{L} - \mathbf{R},$$

*we have $\|\widetilde{\mathbf{B}}\|_{\text{tr}} \leq (1 + \frac{\Delta}{8C})T$, as well as*

$$\frac{C}{T} \left\langle \widetilde{\mathbf{B}} - \mathbf{B}, \mathbf{S}' \right\rangle \leq \Delta \text{ for all } \mathbf{S}' \in \mathcal{S}, \ \mathcal{T}_{\text{mv}}(\widetilde{\mathbf{B}}) = O\left( (|\mathcal{I}| + \mathcal{T}_{\text{mv}}(\mathbf{S})) \cdot R\sqrt{\log \frac{C}{\Delta}} \right).$$

*Proof.* It suffices to provide matrices $\widetilde{\mathbf{L}}$, $\widetilde{\mathbf{R}}$ such that

$$\mathcal{T}_{\text{mv}}(\widetilde{\mathbf{L}}) + \mathcal{T}_{\text{mv}}(\widetilde{\mathbf{R}}) = O\left( (|\mathcal{I}| + \mathcal{T}_{\text{mv}}(\mathbf{S})) \cdot R\sqrt{\log \frac{C}{\Delta}} \right),$$

$$\left\| \widetilde{\mathbf{L}} - \mathbf{L} \right\|_{\text{tr}} \leq \frac{\Delta}{8C} \text{Tr}(\mathbf{L}), \ \left\| \widetilde{\mathbf{R}} - \mathbf{R} \right\|_{\text{tr}} \leq \frac{\Delta}{8C} \text{Tr}(\mathbf{R}).$$

Indeed, letting $\widetilde{\mathbf{B}} := \widetilde{\mathbf{L}} - \widetilde{\mathbf{R}}$, the bound on $\mathcal{T}_{\mathrm{mv}}(\widetilde{\mathbf{B}})$ is then immediate, and by Hölder's inequality (with $\|\mathbf{S}'\|_{\mathrm{op}} \leq \|\mathbf{S}'\|_F \leq 2$) and the triangle inequality, we also have the desired

$$\frac{C}{T}\left\langle \widetilde{\mathbf{B}} - \mathbf{B}, \mathbf{S}' \right\rangle \leq \frac{2C}{T}\left\|\widetilde{\mathbf{B}} - \mathbf{B}\right\|_{\mathrm{tr}} \leq \frac{2C}{T}\left(\left\|\widetilde{\mathbf{L}} - \mathbf{L}\right\|_{\mathrm{tr}} + \left\|\widetilde{\mathbf{R}} - \mathbf{R}\right\|_{\mathrm{tr}}\right) \leq \Delta,$$

by the definition of $T = \mathrm{Tr}(\mathbf{L}) + \mathrm{Tr}(\mathbf{R})$. We only discuss $\widetilde{\mathbf{L}}$, as the case of $\widetilde{\mathbf{R}}$ is symmetric. We let $\widetilde{\mathbf{L}} = p(\frac{1}{\eta}(\mathcal{D}(\frac{x}{v}) - \mathbf{S}))$, where $p$ is the polynomial from Lemma 20 with $\alpha \leftarrow \frac{\Delta}{8C}$. The bound on $\mathcal{T}_{\mathrm{mv}}(\widetilde{\mathbf{L}})$ follows from the degree of $p$, and the fact that $\mathcal{T}_{\mathrm{mv}}(\mathcal{D}(\frac{x}{v})) = O(|\mathcal{I}|)$. Moreover, letting $\{\lambda_i\}_{i \in [d]} \subseteq [-R, R]$ be the eigenvalues of $\frac{1}{\eta}(\mathcal{D}(\frac{x}{v}) - \mathbf{S})$, we indeed have

$$\left\|\widetilde{\mathbf{L}} - \mathbf{L}\right\|_{\mathrm{tr}} = \sum_{i \in [d]} |p(\lambda_i) - \exp(\lambda_i)| \leq \frac{\Delta}{8C}\sum_{i \in [d]} \exp(\lambda_i) = \frac{\Delta}{8C}\mathrm{Tr}(\mathbf{L}).$$

Finally, the claim $\|\widetilde{\mathbf{B}}\|_{\mathrm{tr}} \leq (1 + \frac{\Delta}{8C})T$ follows from the triangle inequality. $\qquad\square$

**Approximating the S step.** We now build upon the access afforded by Corollary 2 to compute $\mathbf{W}$, our approximate solution to $\min_{\mathbf{M} \in \mathcal{S}}\langle \mathbf{H}, \mathbf{M}\rangle$, where $\mathbf{H} := \frac{C}{T}\widetilde{\mathbf{B}}$ is our gradient approximation. Specifically, we use the following standard tool with Lemma 4 to provide our approximate linear optimization implementation over $\mathcal{S}$ for any matrix supporting vector multiplication access.

**Lemma 21.** *Let $\alpha, \delta \in (0, 1)$ and $\mathbf{M} \in \mathbb{S}^{d \times d}$. There is an algorithm which returns $F \in [(1 - \alpha)\|\mathbf{M}\|_F, (1 + \alpha)\|\mathbf{M}\|_F]$ with probability $\geq 1 - \delta$ in time $O(\mathcal{T}_{\mathrm{mv}}(\mathbf{M})\alpha^{-2}\log\frac{d}{\delta})$.*

*Proof.* Let $\mathbf{Q} \in \mathbb{R}^{k \times d}$ with independently uniform random unit vector rows in $\mathbb{R}^d$ scaled down by $\frac{1}{\sqrt{k}}$ for appropriately large $k = \Theta(\alpha^{-2}\log\frac{d}{\delta})$. By the Johnson-Lindenstrauss lemma of [20], with probability $\geq 1 - \delta$, we have for all $i \in [d]$ that

$$\left\|[\mathbf{MQ}^\top]_{i:}\right\|_2^2 \in \left[(1 - \alpha)\|\mathbf{M}_{i:}\|_2^2, (1 + \alpha)\|\mathbf{M}_{i:}\|_2^2\right].$$

Therefore, summing for all $i \in [d]$ proves that $\mathrm{Tr}(\mathbf{QM}^2\mathbf{Q}^\top) = F$ satisfies the desired bound. We can compute $\mathbf{MQ}^\top$ explicitly in time $k \cdot \mathcal{T}_{\mathrm{mv}}(\mathbf{M})$, and computing the Frobenius norm of $\mathbf{MQ}^\top \in \mathbb{R}^{d \times k}$ does not dominate the stated running time. $\qquad\square$

**Lemma 22.** *Let $\alpha, \delta \in (0, 1)$, $\Delta \geq 0$, let $\mathbf{H} \in \mathbb{S}^{d \times d}$ have $\|\mathbf{H}\|_{\mathrm{tr}} \leq \tau$, and let $\mathbf{W}^\star := \mathrm{argmin}_{\mathbf{W} \in \mathcal{S}}\langle \mathbf{W}, \mathbf{H}\rangle$. There is an algorithm which with probability $\geq 1 - \delta$ returns $\mathbf{W} = \mathbf{ZRZ}^\top$ satisfying $\mathbf{R} \in \mathbb{S}^{q \times q}$ and $\mathbf{Z} \in \mathbb{S}^{d \times q}$, with*

$$\langle \mathbf{W}, \mathbf{H}\rangle \leq (1 - \alpha)\langle \mathbf{W}^\star, \mathbf{H}\rangle + \Delta, \quad q = O\left(\frac{\tau^2}{\Delta^2}\right), \tag{32}$$

*in time*

$$O\left(\mathcal{T}_{\mathrm{mv}}(\mathbf{H}) \cdot \left(\frac{1}{\alpha^2} + \frac{\tau^2}{\Delta^2}\right)\log\left(\frac{d}{\delta}\right)\right).$$

*Proof.* Note that in closed form, we have $\mathbf{W}^\star = -(1 + \epsilon)\frac{\mathbf{H}}{\|\mathbf{H}\|_F}$, and the optimal objective value is $\langle \mathbf{W}^\star, \mathbf{H}\rangle = -\|\mathbf{H}\|_F$. We first compute $F$ using Lemma 21 such that

$$\|\mathbf{H}\|_F \leq F \leq \left(1 + \frac{\alpha}{8}\right)\|\mathbf{H}\|_F,$$

within the allotted time and failure probability $\frac{\delta}{2}$. Then, letting $\mathbf{W}' \leftarrow -(1 + \epsilon)\frac{\mathbf{H}}{F} \in \mathcal{S}$, we have

$$\langle \mathbf{W}', \mathbf{H}\rangle - \langle \mathbf{W}^\star, \mathbf{H}\rangle = -\frac{\|\mathbf{H}\|_F^2}{F} + (1 + \epsilon)\|\mathbf{H}\|_F \leq \frac{\alpha}{2}\|\mathbf{H}\|_F \leq -\alpha\langle \mathbf{W}^\star, \mathbf{H}\rangle. \tag{33}$$

Next, let $\mathbf{W} \leftarrow \mathbf{ZZ}^\top\mathbf{W}'\mathbf{ZZ}^\top$ where $\mathbf{Z}$ is the result of Lemma 4 with $q \leftarrow \lceil\frac{9\tau^2}{2\Delta^2}\rceil$, $\epsilon \leftarrow \frac{1}{2}$, and failure probability $\frac{\delta}{2}$. We output $\mathbf{R} := \mathbf{Z}^\top\mathbf{W}'\mathbf{Z}$ and $\mathbf{Z}$, which can be computed within the allotted time. The result follows by combining (33) with

$$\langle \mathbf{W}, \mathbf{H}\rangle - \langle \mathbf{W}', \mathbf{H}\rangle \leq \|\mathbf{W}' - \mathbf{W}\|_{\mathrm{op}}\|\mathbf{H}\|_{\mathrm{tr}} \leq \frac{3\tau}{2}\sigma_{q+1}(\mathbf{W}') \leq \Delta.$$

Here, we used that $\|\mathbf{W}'\|_F \leq 2$ implies $\leq \frac{9\tau^2}{2\Delta^2}$ singular values of $\mathbf{W}'$ are larger than $\frac{2\tau}{3\Delta}$. $\qquad\square$

We make one slight extension to Lemma 22 under implicit access to $\mathbf{H}$.

**Corollary 3.** *In the setting of Lemma 22, suppose $\mathbf{H} = \frac{1}{A}\widetilde{\mathbf{B}}$ for an unknown constant $A$. There is an algorithm which returns $\mathbf{W} = \mathbf{Z}\mathbf{R}\mathbf{Z}^\top$ with $\mathbf{R} \in \mathbb{S}^{q \times q}$ and $\mathbf{Z} \in \mathbb{S}^{d \times q}$ satisfying the guarantee in* (32) *with probability $\geq 1 - \delta$ in time*

$$O\left( \mathcal{T}_{\mathrm{mv}}(\widetilde{\mathbf{B}}) \cdot \left( \frac{1}{\alpha^2} + \frac{\tau^2}{\Delta^2} \right) \log\left( \frac{d}{\delta} \right) \right).$$

*Proof.* We follow the steps of Lemma 22 exactly, except we instead let $\mathbf{W}' \leftarrow -(1+\epsilon)\frac{\widetilde{\mathbf{B}}}{F'}$ where $F'$ satisfies $\|\widetilde{\mathbf{B}}\|_{\mathrm{F}} \leq F' \leq (1 + \frac{\alpha}{8})\|\widetilde{\mathbf{B}}\|_{\mathrm{F}}$, again with probability $\geq 1 - \frac{\delta}{2}$, using Lemma 21. Because the computation in Lemma 21 is scale-invariant, the distribution of $\widetilde{\mathbf{W}}'$ remains unchanged from Lemma 22, and all remaining steps are the same as before. $\square$

We combine the developments of this section to prove Lemma 19.

*Proof of Lemma 19.* We define $\mathbf{H} = \frac{C}{T}\widetilde{\mathbf{B}}$ as in (31) (using $\widetilde{\mathbf{B}}$ defined in Corollary 2), and take the step given by Lemma 22 and Corollary 3. Recall from Corollary 2 that $\|\mathbf{H}\|_{\mathrm{op}} \leq (1 + \frac{\Delta}{8C})C \leq 2C$, so we can set $\tau = 2C$ in Lemma 22. Moreover, in the context of Corollary 3,

$$\mathcal{T}_{\mathrm{mv}}(\widetilde{\mathbf{B}}) = O\left( (|\mathcal{I}| + dr) \cdot R\sqrt{\log \frac{C}{\Delta}} \right)$$

The runtime bound then follows from Corollary 3. Finally, the rank bound $q$ follows from (32). $\square$

### C.4 Vector step implementation

In this section, analogously to Appendix C.3, we discuss how to approximate the $\mathcal{X}$ block of $\nabla f(z)$ with a vector $h$, and give implementation details for computing

$$w \approx \min_{v \in \mathcal{X}} \langle h, v \rangle.$$

We crucially use the characterization in Fact 3, which implies this is a weighted matching problem.

The main result of this section is the following.

**Lemma 23** ($\mathcal{X}$ step). *There is an algorithm which takes as input $x \in \mathcal{X}$, $\mathbf{S} \in \mathcal{S}$ given as a rank-$r$ factorization, $R \geq \|\frac{1}{\eta}(\mathcal{D}(\frac{x}{v}) - \mathbf{S})\|_{\mathrm{op}}$, $\Delta \in [0, C]$, and $\alpha, \delta \in (0, 1)$, and computes $y \in \mathcal{X}$ satisfying*

$$\langle y, h \rangle \leq (1 - \alpha)\mathrm{argmin}_{y^\star \in \mathcal{X}} \langle y^\star, h \rangle, \tag{34}$$

$$\text{for } h \in \mathbb{R}^{\mathcal{I}} \text{ with } |\langle h - \nabla_x F_{\mathrm{prog\text{-}flat}}(x, \mathbf{S}), x' \rangle| \leq \Delta \text{ for all } x' \in \mathcal{X}, \tag{35}$$

*with probability $\geq 1 - \delta$. The algorithm runs in time*

$$O\left( (|\mathcal{I}| + dr) \cdot \frac{\sqrt{R}k^2 C^2 \log^{1.5}(\frac{RdkC}{\delta\Delta})}{\Delta^2} + \frac{|\mathcal{I}|}{\alpha}\log\left( \frac{1}{\alpha} \right) \right).$$

**Approximating the gradient.** Recall that $[\nabla_x f(x, \mathbf{S})]_\iota = -1 + \frac{C}{T}\langle \mathbf{B}, \mathbf{E}_\iota \rangle$ for all $\iota \in \mathcal{I}$, where again $\mathbf{B} = \mathbf{L} - \mathbf{R}$ and $T = \mathrm{Tr}(\mathbf{L}) + \mathrm{Tr}(\mathbf{R})$ as in Appendix C.3. To produce our approximation $h \approx \nabla_x f(x, \mathbf{S})$, we use the following helper guarantee.

**Lemma 24** (Lemma 20, [38]). *Let $\epsilon, \epsilon', \delta \in (0, 1)$, $\{\mathbf{A}_i\}_{i \in [n]} \in \mathbb{S}^{d \times d}$, $R \geq \log \frac{1}{\epsilon'}$, and let $\mathbf{M} \in \mathbb{S}^{d \times d}$ satisfy $\|\mathbf{M}\|_{\mathrm{op}} \leq R$. There is an algorithm which computes $\{V_i\}_{i \in [n]} \subseteq \mathbb{R}$ such that for all $i \in [n]$,*

$$|V_i - \langle \mathbf{A}_i, \exp(\mathbf{M}) \rangle| \leq \epsilon \langle |\mathbf{A}_i|, \exp(\mathbf{M}) \rangle + \epsilon'\mathrm{Tr}(|\mathbf{A}_i|)\mathrm{Tr}\exp(\mathbf{M})$$

*with probability $\geq 1 - \delta$ in time*

$$O\left( \mathcal{T}_{\mathrm{mv}}(\mathbf{M}) \cdot \frac{\sqrt{R}\log^{1.5}(\frac{Rnd}{\delta\epsilon\epsilon'})}{\epsilon^2} + \left( \sum_{i \in [n]} \mathcal{T}_{\mathrm{mv}}(\mathbf{A}_i) \right) \cdot \frac{\log \frac{nd}{\delta}}{\epsilon^2} \right).$$

With Lemma 24 established, our gradient approximation follows straightforwardly,

**Corollary 4.** *Let $x \in \mathcal{X}$, $\mathbf{S} \in \mathcal{S}$ satisfy $\|\frac{1}{\eta}(\mathcal{D}(\frac{x}{v}) - \mathbf{S})\|_{\text{op}} \leq R$, and let $\Delta \geq 0$ and $\delta \in (0, 1)$. We can compute $h \in \mathbb{R}^{\mathcal{I}}$ in time*

$$O\left((|\mathcal{I}| + \mathcal{T}_{\text{mv}}(\mathbf{S})) \cdot \frac{\sqrt{R}k^2 C^2 \log^{1.5}(\frac{RdkC}{\delta\Delta})}{\Delta^2}\right),$$

*such that with probability $\geq 1 - \delta$,*

$$|\langle h - \nabla_x f(x, \mathbf{S}), x' \rangle| \leq \Delta \text{ for all } x' \in \mathcal{X}.$$

*Proof.* Define $T$ as in Corollary 2 throughout the proof. We let $h \in \mathbb{R}^{\mathcal{I}}$ satisfy $h_{\mathcal{I}} = -1 + \frac{C(V_\iota - U_\iota)}{T}$ for all $\iota \in \mathcal{I}$, for $V, U \in \mathbb{R}^{\mathcal{I}}$ to be defined, which we can compute within the allotted time. Since $\|x'\|_1 \leq k$ for all $x' \in \mathcal{X}$, it suffices to show that $\|h - \nabla_x f(x, \mathbf{S})\|_\infty \leq \frac{\Delta}{2k}$.

To this end, throughout this proof let $\mathbf{M} := \frac{1}{\eta}(\mathcal{D}(\frac{x}{v}) - \mathbf{S})$, and define

$$V_\iota^\star := \langle \exp(\mathbf{M}), \mathbf{E}_\iota \rangle, \ U_\iota^\star := \langle \exp(-\mathbf{M}), \mathbf{E}_\iota \rangle, \text{ for all } \iota \in \mathcal{I}.$$

By our choice of $h_\iota$, we have

$$|[h - \nabla_x f(x, \mathbf{S})]_\iota| = \frac{C}{T}(|V_\iota - V_\iota^\star| + |U_\iota - U_\iota^\star|).$$

We take $\{V_\iota\}_{\iota \in \mathcal{I}}$ to be the approximations given by Lemma 24 with the set of matrices $\{\mathbf{E}_\iota\}_{\iota \in \mathcal{I}}$ and $\mathbf{M}$, $\delta \leftarrow \frac{\delta}{2}$, and $\epsilon, \epsilon' \leftarrow \frac{\Delta}{4kC}$; we define $\{U_\iota\}_{\iota \in \mathcal{I}}$ identically, except we apply Lemma 24 with $-\mathbf{M}$ instead of $\mathbf{M}$. Note that these approximations yield

$$\frac{C}{T}(|V_\iota - V_\iota^\star| + |U_\iota - U_\iota^\star|) \leq \frac{C}{T}(\epsilon + \epsilon')\text{Tr}(|\mathbf{E}_\iota|)(\text{Tr}\exp(\mathbf{M}) + \text{Tr}\exp(-\mathbf{M}))$$

$$\leq 2C(\epsilon + \epsilon') \leq \frac{\Delta}{k},$$

as desired. The first inequality used the guarantee in Lemma 24 with Hölder's inequality, and the second used the definition of $T = \text{Tr}\exp(\mathbf{M}) + \text{Tr}\exp(-\mathbf{M})$ and $\text{Tr}(|\mathbf{E}_\iota|) = 2$. Finally, the runtime follows directly from the bound in Lemma 24. $\square$

**Approximating the $x$ step.** Given our estimate $h$ from Corollary 4, our approximate linear optimization oracle follows from a known result on approximating maximum weighted matchings.

**Proposition 5** (Theorem 3.12, [22]). *Let $G = (L \cup R, E)$ be a bipartite graph, let $w \in \mathbb{R}^E$ be a set of weights, and let $m := |E|$. There is an algorithm which computes $f \in \mathcal{M}$, defined in (23), such that $\langle w, f \rangle \leq (1 - \alpha)\min_{f^\star \in \mathcal{M}} \langle w, f^\star \rangle$, in time $O(\frac{m}{\alpha}\log\frac{1}{\alpha})$.*

We remark that Proposition 5 is stated in [3] for maximization of linear objectives in nonnegative weight vectors $w$, i.e. maximum weight matchings. However, it is straightforward to see that this implies the same result for minimization of linear objectives in arbitrary cost vectors $w \in \mathbb{R}^E$, by first dropping any positive coordinates (which can only increase the objective $\langle w, x \rangle$ when $x$ is coordinatewise nonnegative), and then negating the result of [22]. Because our access to our cost vector $w$ of interest is implicit, we make a simple observation which extends Proposition 5.

**Corollary 5.** *In the setting of Proposition 5, let $C \in \mathbb{R}_{\geq 0}$, and suppose $w = \frac{1}{A}v$ for an unknown constant $A > 0$, and an explicit vector $v \in \mathbb{R}^n$. The guarantees of Proposition 5 hold true for inputs $(w, G, \mathcal{M})$ if run on inputs $(v, G, C \cdot \mathcal{M})$ instead.*

*Proof.* This follows directly from scale-invariance of the guarantees in Proposition 5. $\square$

We now use these developments to give a short proof of Lemma 23.

*Proof of Lemma 23.* We define $h_\iota = -1 + \frac{C(V_\iota - U_\iota)}{T}$ entrywise (using notation in Corollary 4), and take the step given by Proposition 5 and Corollary 5. The runtime follows immediately from these results. $\square$

## C.5 Runtime analysis for subproblem solver

In this section, we put together the pieces and provide a runtime bound for approximately solving (25). We first provide Lemma 25 which provides the bounds for implementing Line 5 for $\mathcal{Z}$ and then use this to provide the main result for approximately solving, (25), Proposition 6.

**Lemma 25.** *There is an algorithm that for input $z = (x, \mathbf{S}) \in \mathcal{X} \times \mathcal{S}$ where $\mathbf{S}$ is given as a rank-$r$ factorization, $R \geq \|\frac{1}{\eta}(\mathcal{D}(\frac{x}{v}) - \mathbf{S})\|_{\mathrm{op}}$, $\Delta \in [0, C]$, and $\delta \in (0, 1)$ outputs $\hat{z} \in \mathcal{Z}$ with $\hat{z} = (y, \mathbf{W}) \in \mathcal{X} \times \mathcal{S}$ where $\mathbf{W}$ is output as a rank $O(\frac{C^2}{\Delta^2})$ factorization such that with probability $\geq 1 - \delta$,*

$$\langle \nabla F_{\mathrm{prog\text{-}flat}}(z), \hat{z} \rangle \leq \min_{z' \in \mathcal{Z}} \langle \nabla_{\mathcal{S}} F_{\mathrm{prog\text{-}flat}}(x, \mathbf{S}), z' \rangle + \Delta$$

*in time*

$$O\left( (|\mathcal{I}| + dr) \cdot \left( R \frac{C^2}{\Delta^2} \sqrt{\log \frac{C}{\Delta}} \log\left(\frac{d}{\delta}\right) + \frac{\sqrt{R} k^2 C^2 \log^{1.5}(\frac{RdkC}{\delta\Delta})}{\Delta^2} \right) + |\mathcal{I}|\left(k + \frac{Cku}{\Delta\ell^2 d}\right) \log\left(\frac{Cku}{\Delta\ell^2 d}\right) \right).$$

*Proof.* Applying Lemma 19 and Lemma 23 yields that for any $\alpha_{\mathcal{S}}, \alpha_{\mathcal{X}} \in (0, 1]$ and $\bar{\Delta} \in [0, C]$ (which we set later) we can obtain with probability $\geq 1 - \delta$ (by a union bound) $\mathbf{W} \in \mathcal{S}$ as a rank-$q$ factorization for $q = O(\frac{C^2}{\bar{\Delta}^2})$ and $\mathbf{H} \in \mathbb{S}^{d \times d}$ satisfying (Lemma 19)

$$\langle \mathbf{H}, \mathbf{W} \rangle \leq (1 - \alpha_{\mathcal{S}}) \min_{\mathbf{S}' \in \mathcal{S}} \langle \mathbf{H}, \mathbf{S}' \rangle + \bar{\Delta} \text{ and } |\langle \mathbf{H} - \nabla_{\mathcal{S}} F_{\mathrm{prog\text{-}flat}}(x, \mathbf{S}), z' \rangle| \leq \bar{\Delta} \text{ for all } \mathbf{S}' \in \mathcal{S},$$

and $y \in \mathcal{X}$ and $h \in \mathbb{R}^{\mathcal{I}}$ satisfying (Lemma 23)

$$\langle h, y \rangle \leq (1 - \alpha_{\mathcal{X}}) \min_{x' \in \mathcal{X}} \langle h, x' \rangle \text{ and } |\langle h - \nabla_{\mathcal{X}} F_{\mathrm{prog\text{-}flat}}(x, \mathbf{S}), x' \rangle| \leq \bar{\Delta} \text{ for all } x' \in \mathcal{X},$$

in time

$$O\left( (|\mathcal{I}| + dr)\left( R\sqrt{\log \frac{C}{\bar{\Delta}}} \cdot \left(\frac{1}{\alpha_{\mathcal{S}}^2} + \frac{C^2}{\bar{\Delta}^2}\right) \log\left(\frac{d}{\delta}\right) + \frac{\sqrt{R} k^2 C^2 \log^{1.5}(\frac{RdkC}{\delta\bar{\Delta}})}{\bar{\Delta}^2} \right) + \frac{|\mathcal{I}|}{\alpha_{\mathcal{X}}} \log\left(\frac{1}{\alpha_{\mathcal{X}}}\right) \right).$$

Conditioning on these events and applying Lemma 15 and Lemma 18 yields that

$$\langle \nabla_{\mathcal{S}} F_{\mathrm{prog\text{-}flat}}(x, \mathbf{S}), \mathbf{W} \rangle \leq \min_{x' \in \mathcal{X}} \langle \nabla_{\mathcal{S}} F_{\mathrm{prog\text{-}flat}}(x, \mathbf{S}), x' \rangle + \alpha_{\mathcal{S}} \cdot 2C + 3\bar{\Delta}.$$

and

$$\langle \nabla_{\mathcal{X}} F_{\mathrm{prog\text{-}flat}}(x, \mathbf{S}), y \rangle \leq \min_{\mathbf{S}' \in \mathcal{S}} \langle \nabla_{\mathcal{X}} F_{\mathrm{prog\text{-}flat}}(x, \mathbf{S}), \mathbf{S}' \rangle + \alpha_{\mathcal{X}} \cdot \left(C\frac{ku}{\ell^2 d} + k\right) + 3\bar{\Delta}.$$

Picking $\alpha_{\mathcal{S}} = \frac{\Delta}{8C}$, $\alpha_{\mathcal{X}} = \min\{\frac{\Delta}{8} \cdot (k + \frac{Cku}{\ell^2 d})^{-1}, 1\}$ and $\bar{\Delta} = \frac{\Delta}{24}$ then yields the result. $\square$

We now prove a more quantitative variant of Proposition 2.

**Proposition 6.** *Let $\epsilon \in [0, C]$ and $\delta \in (0, 1)$, and suppose $|[\mathbf{D}]_\iota| \in [\ell, u]$ for all $\iota \in \mathcal{I}$. There is an algorithm computing $(x, \mathbf{S})$, an $\epsilon$-approximate minimizer to (5) with probability $\geq 1 - \delta$, in time*

$$O\left( T\left(|\mathcal{I}| + \frac{C^2 T}{\epsilon^2} \cdot d\right) \cdot \left( R_\eta \frac{C^2}{\epsilon^2} \sqrt{\log \frac{C}{\epsilon}} \log\left(\frac{dT}{\delta}\right) + \frac{\sqrt{R_\eta} k^2 C^2 \log^{1.5}(\frac{R_\eta dkC}{\delta\epsilon})}{\epsilon^2} \right) \right)$$

$$+ O\left( T|\mathcal{I}| \cdot \frac{C\eta R_\eta}{\epsilon} \log\left(\frac{C\eta R_\eta}{\epsilon}\right) \right),$$

*for $R_\eta := \eta^{-1}(2k + \frac{ku}{\ell^2 d})$ and $T = \lceil \frac{8C\eta R_\eta^2}{\epsilon} \rceil$, and $\mathbf{S}$ is given as a rank-$r = O(\frac{C^2}{\epsilon^2} \cdot T)$ factorization.*

*Proof.* Throughout the proof let $f := F_{\mathrm{prog\text{-}flat}}$ for simplicity. Our algorithm computes $z^{(T)} = \mathsf{ApproxFrankWolfe}(f, \mathcal{Z}, T, \frac{2C\eta R_\eta^2}{C_{f,\mathcal{Z}}})$ (Algorithm 4), where we note that $C_{f,\mathcal{Z}}$ need not necessarily be computed. By Lemma 18 we know that $C_{f,\mathcal{Z}} \leq 2C\eta R_\eta^2$ and consequently Proposition 4 implies that so long as each iteration of Algorithm 4 is implemented to the desired accuracy,

$$f(z^{(T)}) - f(z^\star) \leq \frac{2C_{f,\mathcal{Z}}}{T+2} + \frac{4C\eta R_\eta^2}{T+2} \leq \frac{8C\eta R_\eta^2}{T+2} \leq \epsilon.$$

To implement the method we apply Lemma 25 with $(\Delta, \delta)$ in the lemma set to $(\frac{C\eta R_\eta^2}{T+2}, \frac{\delta}{T})$. This suffices to have the desired output with the desired probability by union bound.

It remains to bound the computational cost of this algorithm and the rank of the output. Note that $\frac{C\eta R_\eta^2}{T} = \Omega(\epsilon^{-1})$ and $R_\eta \geq \|\frac{1}{\eta}(\mathcal{D}(\frac{x'}{v}) - \mathbf{S}')\|_{\mathrm{op}}$ for all $(x', \mathbf{S}') \in \mathcal{X} \times \mathcal{S}$ (by Lemma 17, see e.g., (28)). Additionally, if we initialize the method with $(\mathbb{0}_d, \mathbf{0}_{d\times d})$ then we see that the rank of the $\mathcal{S}$ component of each computed $s^{(t)}$ is $O(\frac{C^2}{\epsilon^2})$ and that each $\mathcal{S}$ component of each $z^{(t)}$ can be maintained as a rank $O(t \cdot \frac{C^2}{\epsilon^2})$ factorization. The computational cost is then dominated by the cost of each iteration which, by Lemma 25 is bounded as claimed after adjusting by $T$. $\qquad\square$

# D  Postprocessing

In this section, we give the second main component of our overall matrix completion algorithm in Algorithm 7. This piece takes as input two matrices $\mathbf{M}, \mathbf{M}^\star \in \mathbb{S}^{d\times d}$ which are close on a submatrix (where we only have noisy semi-random observations from $\mathbf{M}^\star$). It then performs a sequence of postprocessing steps to produce a matrix $\widetilde{\mathbf{M}}$ such that $\widetilde{\mathbf{M}}$ is close to $\mathbf{M}^\star$ on the full matrix, as measured in the $\ell_\infty$ norm. For the first step of Algorithm 7, we simply reuse our earlier subroutine Sparsify (Algorithm 1), whose guarantees we recall here for convenience.

**Lemma 13.** *Let $\mathbf{D} \in \mathbb{S}^{d\times d}$, such that there is $A \subseteq U \subseteq [d]$ with $|U\backslash A| \leq \gamma d$, $\|\mathbf{D}_{A\times A}\|_\infty \leq \tau$. Let $s \in \mathbb{N}$ and $p \geq \frac{800}{s}\log(\frac{d}{\delta})$. Then with probability $\geq 1 - \delta$, Algorithm 1 returns $U' \subseteq U$ with $|U \setminus U'| \leq 4\gamma d \log d$, and $\mathbf{D}_{U'\times U'}$ has at most $s$ entries with magnitudes $\geq \tau$ per row or column.*

In Appendix D.1, we prove a structural fact about bounded spanning subsets of low-rank matrices. We use this to analyze our fixing step in Appendices D.2 and D.3, where we use regression problems on a subset of columns passing an appropriate test to complete the matrix.

## D.1  Localizing errors via spanning subsets

Before giving our next postprocessing steps, we prove a structural property, which can be stated concisely in the language of *volumetric spanners* [5]. Following definitions in [7], Lemma 26 shows any $n$-sized set in $\mathbb{R}^d$ admits a $2d$-approximate $\ell_1$-volumetric spanner of size $d + 1$.

**Lemma 26.** *Given $\{v_i\}_{i\in[n]} \subset \mathbb{R}^d$, there exists $S \subseteq [n]$ with $|S| \leq d + 1$ such that for all $i \in [n]$,*

$$v_i = \sum_{j\in S} a_j v_j$$

*with coefficients satisfying $\sum_{j\in S} |a_j| \leq 2d$.*

*Proof.* Consider the largest volume simplex among the $\{v_i\}_{i\in[d]}$, say given by $\{v_i\}_{i\in[d+1]}$ without loss of generality. We claim all the other vectors are contained in the simplex with $d + 1$ vertices,

$$\sum_{j\in[d+1]} v_j - dv_i, \text{ for all } i \in [d + 1].$$

To see this, fix $i \in [d + 1]$, and let the $(d - 1)$-dimensional plane spanned by the other $d$ points be $P$. If any other $v_j$ lies outside the two planes parallel to $P$ at equal distances, such that one contains $v_i$, then including $v_j$ would produce a larger-volume simplex. Taking the intersections of the halfspaces defined by these planes produces the simplex described above. From this, we have the desired property by convexity of the $\ell_1$ norm, applied to the coefficients defining this simplex. $\qquad\square$

Using Lemma 26, we obtain the following corollary on boundedness of large index subsets.

**Lemma 27.** *Let $\mathbf{D} \in \mathbb{S}^{d\times d}$ be rank-$r$, and assume each row of $\mathbf{D}$ has at most $s$ entries with magnitude $\geq \tau$. There exists $S \subseteq [d]$ with $|[d] \setminus S| \leq (r + 1)s$ such that $\|[\mathbf{D}]_{S:}\|_\infty, \|[\mathbf{D}]_{:S}\|_\infty \leq 2r\tau$.*

*Proof.* Let $\mathbf{U}\Sigma\mathbf{U}^\top$ be an SVD of $\mathbf{D}$, and let $\{u_i\}_{i\in[d]} \in \mathbb{R}^r$ be the rows of $\mathbf{U}$. By Lemma 26, we can find a subset $T$ such that all rows $i \in [d]$ can be written as

$$u_i = \sum_{j\in T} a_j u_j, \text{ with } \sum_{j\in T} |a_j| \leq 2r.$$

This also implies every row of $\mathbf{D}$ can be written as a linear combination of the row vectors $\{\mathbf{D}_{j:}\}_{j \in T}$, such that the $\ell_1$ norm of the linear combination coefficients is at most $2r$. Now, let $S = [d] \setminus R$, where $R$ consists of all indices $i \in [d]$ such that $|[\mathbf{D}]_{ji}| \geq \tau$ for some $j \in T$; by assumption, $|R| \leq (r+1)s$, giving the size bound. Take some row $\mathbf{D}_{i:}$ for $i \in [d]$, and let $\mathbf{D}_{i,S}$ be the restriction of this row to columns in $S$. By applying the given linear combination to the rows $\{\mathbf{D}_{j,S}\}_{j \in T}$, all entrywise at most $\tau$ in magnitude, the triangle inequality shows $\|\mathbf{D}_{i,S}\|_\infty \leq 2r\tau$ as claimed. $\qquad \square$

### D.2 Testing columns

In this section, we present the second step of our postprocessing algorithm, where we design a test to identify columns with no large errors. This step uses our structural claim in Lemma 27.

---

**Algorithm 5:** $\mathsf{Test}(\mathbf{M}, j, \phi, q, \delta)$

---

1 **Input:** $\mathbf{M} \in \mathbb{R}^{d \times d}$, $j \in [d]$, $\phi \geq 0$, $q, \delta \in (0,1)$
2 $t_{\max} \leftarrow \lceil 10 \log \frac{1}{\delta} \rceil$
3 $c \leftarrow 0$
4 **for** $t \in [t_{\max}]$ **do**
5 $\quad$ Sample $T \subseteq [d]$ by including each element independently with probability $q$
6 $\quad$ **if** $\min_{v \in \mathbb{R}^{|T|}} \|\mathbf{M}_{:T} v - \mathbf{M}_{:j}\|_\infty + \phi \|v\|_\infty \leq 2\phi$ **then** $c \leftarrow c+1$
7 **end**
8 **if** $c \geq \frac{1}{2} t_{\max}$ **then Return: true**
9 **else Return: false**
10

---

To analyze Algorithm 5, we need the following basic claims which are standard in the literature.

**Lemma 28.** *Let rank-$r$ $\mathbf{M} \in \mathbb{R}^{d \times k}$ be $\mu$-incoherent with $d \geq k$, and let $x \in \mathrm{span}(\mathbf{M})$. Then for all $i \in [d]$, $|x_i| \leq (\frac{\mu r}{d})^{1/2} \|x\|_2$.*

*Proof.* Let $\mathbf{U} \in \mathbb{R}^{d \times r}$ be an orthonormal basis for $\mathrm{span}(\mathbf{M})$ with rows $\{u_i\}_{i \in [d]}$, so that $x = \mathbf{U}v$ with $\|x\|_2 = \|v\|_2$. The conclusion follows from

$$|x_i| = |\langle u_i, v \rangle| \leq \|u_i\|_2 \|x\|_2 \leq \sqrt{\frac{\mu r}{d}} \|x\|_2.$$

$\qquad \square$

**Lemma 29.** *Let $V \subseteq \mathbb{R}^d$ be a $\mu$-incoherent subspace of dimension $r$, and let $\mathbf{B}_V \in \mathbb{R}^{d \times r}$ be an orthonormal basis for $V$. Let $S \subseteq [d]$ have $|S| \geq (1 - \frac{1}{3\mu r})d$. Then the following properties hold.*

- *For any $v \in V$, $\|v_S\|_2 \geq \sqrt{\frac{2}{3}} \|v\|_2$.*

- *$\mathrm{span}([\mathbf{B}_V]_{S:})$, as a subspace of $\mathbb{R}^{|S|}$, is $3\mu$-incoherent.*

*Proof.* Let $\mathbf{A} := [\mathbf{B}_V]_{S:}^\top [\mathbf{B}_V]_{S:} = \sum_{i \in S} b_i b_i^\top$, where $\{b_i\}_{i \in [d]}$ are rows of $\mathbf{B}_V$. By the definition of incoherence, we have that $\|\mathbf{I}_r - \mathbf{A}\|_{\mathrm{op}} \leq \mathrm{Tr}(\mathbf{I}_r - \mathbf{A}) = \sum_{i \in [d] \setminus S} \mathrm{Tr}(b_i b_i^\top) \leq \frac{1}{3}$, so that

$$\mathbf{I}_r \succeq \mathbf{A} \succeq \frac{2}{3} \mathbf{I}_r.$$

The lower bound then yields the first statement. Now, the matrix $\mathbf{M} := [\mathbf{B}_V]_{S:} \mathbf{A}^{-\frac{1}{2}}$ has orthonormal columns, and the same span as $[\mathbf{B}_V]_{S:}$. Furthermore, by incoherence and the fact that $\mathbf{A} \succeq \frac{2}{3} \mathbf{I}_r$, all of the rows $i \in S$ of $\mathbf{M}$ have

$$\|\mathbf{M}_{i:}\|_2 \leq \sqrt{\frac{3}{2}} \|[\mathbf{B}_V]_{i:}\|_2 \leq \sqrt{\frac{3\mu r}{2d}}.$$

Finally, $|S| \geq \frac{2d}{3}$ (since $\mu \geq 1$). Combining these claims proves the second statement. $\qquad \square$

We also include the following application of the matrix Bernstein inequality from [43].

**Lemma 30** (Lemma 12, [43]). *Let $\delta \in (0,1)$ and let $V \subseteq \mathbb{R}^d$ be a $\mu$-incoherent subspace of dimension $r$. Let $\mathbf{B}_V$ be an orthononormal basis for $V$, with rows $\{b_i\}_{i \in [d]} \subset \mathbb{R}^r$. Let $S \subseteq [d]$ have $|S| \geq (1 - \frac{1}{3\mu r})d$. Let $T \subseteq S$ include each element independently with probability $p \geq \frac{100\mu r}{d} \log \left( \frac{2r}{\delta} \right)$. Then with probability $\geq 1 - \delta$,*

$$\frac{p}{2}\mathbf{I}_r \preceq \sum_{i \in T} b_i b_i^\top \preceq 2p\mathbf{I}_r.$$

Now, we are ready to analyze our testing algorithm and prove that it has the desired behavior.

**Lemma 31.** *Let $\mathbf{M}^\star \in \mathbb{S}^{d \times d}$ be rank-$r^\star$ and $\mu$-incoherent, and let $\mathbf{M} \in \mathbb{S}^{d \times d}$ be rank-$r$ for $r \geq r^\star$. For $s, \tau > 0$, assume $\mathbf{M} - \mathbf{M}^\star$ has at most $s$ entries with magnitudes $\geq \tau$ in each row. Then with probability $1 - \delta$, if we run Algorithm 5 with parameters $q, \phi$ and $j \in [d]$ such that*

$$q \geq \frac{1000\mu r^\star}{d} \log \left( \frac{d}{\delta} \right), \ q \leq \frac{1}{20\mu s r^\star}, \ \phi = \sqrt{q d} r^\star \tau,$$

*Algorithm 5 has the following behavior.*

- *If $\|\mathbf{M}_{:j}^\star - \mathbf{M}_{:j}\|_\infty \leq 4r^\star \tau \sqrt{\mu r^\star}$, Algorithm 5 returns* **true**.
- *If $\|\mathbf{M}_{:j}^\star - \mathbf{M}_{:j}\|_\infty \geq 10\tau q d \mu r^\star$, Algorithm 5 returns* **false**.

*Proof.* Let the SVD of $\mathbf{M}^\star$ be $\mathbf{M}^\star = \mathbf{U}^\star \boldsymbol{\Sigma}^\star (\mathbf{U}^\star)^\top$. We consider the two claims separately.

*True case.* Suppose that $\|\mathbf{M}_{:j}^\star - \mathbf{M}_{:j}\|_\infty \leq 4r^\star \tau \sqrt{\mu r^\star}$. We will prove that each run of the inner loop of the algorithm succeeds with probability $\geq \frac{4}{5}$. By using Lemma 27 with $\mathbf{D} \leftarrow \mathbf{M}^\star - \mathbf{M}$, and applying our upper and lower bounds on $q$, there exists $S \subseteq [d]$ with $|S| \geq (1 - \frac{1}{3\mu r^\star})d$ such that for all $i \in S$, $\|\mathbf{M}_{:i}^\star - \mathbf{M}_{:i}\|_\infty \leq 4r^\star \tau$. Next, by Lemma 30 and incoherence of $\mathbf{M}^\star$, with probability $\frac{9}{10}$,

$$(\mathbf{V}_{:T \cap S}^\star)^\top \mathbf{V}_{:T \cap S}^\star \succeq \frac{2}{3}q\mathbf{I}_r.$$

This means there exists $v \in \mathbb{R}^{T \cap S}$ with $\|v\|_2 \leq \sqrt{\frac{3\mu r^\star}{2qd}}$, such that $(\mathbf{V}_{:T \cap S}^\star)^\top v = (\mathbf{V}_{:j}^\star)^\top$, which also means $\mathbf{M}_{:T \cap S}^\star v = \mathbf{M}_{:j}^\star$. We thus have for this vector $v$,

$$
\begin{aligned}
\|\mathbf{M}_{:T \cap S} v - \mathbf{M}_{:j}\|_\infty + \phi \|v\|_\infty &\leq \|\mathbf{M}_{:T \cap S}^\star v - \mathbf{M}_{:T \cap S} v\|_\infty + \|\mathbf{M}_{:j}^\star - \mathbf{M}_{:j}\|_\infty + \phi \|v\|_\infty \\
&\leq 4r^\star \tau \|v\|_1 + 4r^\star \tau \sqrt{\mu r} + \phi \|v\|_\infty \quad\quad (36) \\
&\leq 4r^\star \tau \|v\|_2 \sqrt{|T|} + 6r^\star \tau \sqrt{\mu r^\star} \leq 20 r \tau \sqrt{\mu r^\star} \leq 2\phi,
\end{aligned}
$$

where we used that with probability $\frac{9}{10}$, $|T| \leq 2qd$. By a union bound, each independent test thus increments $c$ with probability $\frac{4}{5}$ over the randomness of $T$, so a Chernoff bound gives the claim.

*False case.* Next, suppose that $\|\mathbf{M}_{:j}^\star - \mathbf{M}_{:j}\|_\infty \geq 10\tau q d \mu r^\star$, and again consider a single run of the algorithm. Let $i_0 \in [d]$ be the entry such that $|\mathbf{M}_{i_0 j}^\star - \mathbf{M}_{i_0 j}| \geq 10\tau q d \mu r^\star$. First, with probability at least $\frac{9}{10}$ over the randomness of the inner loop, for all $j' \in T$,

$$\left| \mathbf{M}_{i_0 j'}^\star - \mathbf{M}_{i_0 j'} \right| \leq \tau, \quad\quad (37)$$

because by assumption, there are at most $s$ indices $j'$ where $|\mathbf{M}_{i_0, j'}^\star - \mathbf{M}_{i_0, j'}| \geq \tau$. Since $qs \leq \frac{1}{20}$, there is at least a $\frac{9}{10}$ probability that none of these indices is selected in $T$ by a Chernoff bound. As before, with probability $\frac{9}{10}$, there is a vector $v$ with $\|v\|_2 \leq \sqrt{\frac{3\mu r^\star}{2qd}}$ such that $\mathbf{M}_{:T}^\star v = \mathbf{M}_{:j}^\star$. Also, again, with probability $\frac{9}{10}$, we have $|T| < 2qd$. Now let $R \subseteq [d]$ be the set of rows $i$ such that

$$\left\| [\mathbf{M}^\star]_{i, T \cup \{j\}} - [\mathbf{M}]_{i, T \cup \{j\}} \right\|_\infty \leq \tau.$$

By assumption, each row in $T \cup \{j\}$ can only remove $s$ indices from $R$, so

$$|R| \geq d - s(|T| + 1) \geq d - 2qsd \geq \left(1 - \frac{1}{10\mu r^\star}\right)d.$$

Consider any solution $v'$ to the problem in Line 6 which increments $c$. For $v'$ to be feasible, we must have $\|v'\|_\infty \leq 2$, so $\|v'\|_1 \leq 2|T|$. Further,

$$\|\mathbf{M}_{:T}v' - \mathbf{M}_{:j}\|_\infty \leq 2r^\star\tau\sqrt{qd}. \tag{38}$$

Hence, for all $i \in R$,

$$\begin{aligned}
\left|[\mathbf{M}^\star]_{i,T}(v - v')\right| &\leq \left|\mathbf{M}^\star_{ij} - \mathbf{M}_{ij}\right| + \left|([\mathbf{M}]_{i,T} - [\mathbf{M}^\star]_{i,T})v'\right| + \left|[\mathbf{M}]_{i,T}v' - \mathbf{M}_{ij}\right| \\
&\leq \tau + \|v'\|_1\tau + 2r^\star\tau\sqrt{qd} \leq 3\tau qd\sqrt{r^\star}.
\end{aligned} \tag{39}$$

In the first line, we used that $\mathbf{M}^\star_{:T}v = \mathbf{M}^\star_{:j}$ by definition. Also, note that $\mathbf{M}^\star_{:T}(v - v')$ is in the span of $\mathbf{M}^\star$, so by Lemma 29, the fact that $|R| \geq (1 - \frac{1}{10\mu r^\star})d$, and (39),

$$\|\mathbf{M}^\star_{:T}(v - v')\|_2 \leq \sqrt{\frac{3}{2}}\|\mathbf{M}^\star_{R \times T}(v - v')\|_2 \leq 4\tau qd\sqrt{r^\star d}.$$

Therefore, by incoherence of $\mathbf{M}^\star$, Lemma 28 implies that for all $i \in [d]$,

$$\left|[\mathbf{M}^\star]_{i,T}(v - v')\right| \leq 4\tau qd\mu r^\star. \tag{40}$$

However, consider the index $i_0$ defined at the beginning of this case. Similarly to (39),

$$\begin{aligned}
\left|\mathbf{M}^\star_{i_0 j} - \mathbf{M}_{i_0 j}\right| &\leq \left|[\mathbf{M}]_{i_0,T}v' - \mathbf{M}_{i_0 j}\right| + \left|\left([\mathbf{M}]_{i_0,T} - [\mathbf{M}^\star]_{i_0,T}\right)v'\right| + \left|[\mathbf{M}^\star]_{i_0,T}(v - v')\right| \\
&\leq 2\sqrt{qd}r^\star\tau + \tau \cdot 2|T| + 4\tau qd\mu r^\star < 10\tau qd\mu r^\star,
\end{aligned}$$

where we used (37), (38), (40), and $|T| \leq 2qd$. This contradicts the assumption that

$$\left|\mathbf{M}^\star_{i_0 j} - \mathbf{M}_{i_0 j}\right| \geq 10\tau qd\mu r^\star.$$

We conclude that the problem on Line 6 fails to increment $c$ with probability $\geq \frac{2}{3}$, by a union bound. Since we repeat the loop $10\log(\frac{1}{\delta})$ times, with $1 - \delta$ probability the test will reject, as claimed. $\square$

**Remark 2.** *From observation, the proof of Lemma 31 continues to hold if the problem on Line 6 is solved up to accuracy $0.1\phi$. That is, the true case actually proves a stronger bound of $1.9\phi$ in (36), so it would continue to pass for any error $\leq 0.1\phi$. Similarly, even if we assumed the vector $v$ in the false case achieved a value of $\leq 2.1\phi$ (instead of $2\phi$), we would still have a contradiction.*

Finally, we include a claim from prior work which bounds the complexity of implementing Line 6.

**Lemma 32.** *Following notation in Lemma 31, for any $\delta \in (0, 1)$, assuming $q = O(\frac{\mu r^\star}{d}\log(\frac{d}{\delta}))$, the optimization problem on Line 6 can be solved to accuracy $0.1\phi$ with probability $\geq 1 - \delta$, in time*

$$O\left(d \cdot \text{poly}\left(\mu r^\star \log\left(\frac{d}{\delta}\right)\right)\right).$$

*Proof.* For shorthand let $\mathbf{M} := \mathbf{M}_{:T}$, $k := |T|$, and $u := \mathbf{M}_{:j}$ in this proof, and assume that $k \leq 2qd$, which occurs with probability $\geq 1 - \frac{\delta}{2}$. We create a linear program,

$$\min_{x \in \mathbb{R}^{k+2}, \mathbf{A}x \leq b} c^\top x,$$

where $x^\top = \begin{pmatrix} v^\top & \alpha & \beta \end{pmatrix}$ for $\alpha, \beta \in \mathbb{R}$ and $v \in \mathbb{R}^k$. We encode the constraints $\|\mathbf{M}v - u\|_\infty \leq \alpha$, $\|v\|_\infty \leq \beta$ using $2d + 2k$ linear constraints on $x$. Finally, we make the objective $c^\top x = \alpha + \phi\beta$. The conclusion follows by applying the solver of [69], Theorem 1, with failure probability $\frac{\delta}{2}$, since $k = O(\mu r^\star \log(\frac{d}{\delta}))$ and we have a $O(d + k) \times O(k)$ constraint matrix. $\square$

**Algorithm 6:** $\mathsf{Fix}(\mathcal{O}_p^{\mathrm{sr}}(\widehat{\mathbf{M}}), \mathbf{M}, T, \phi, q, \delta)$

---

**1** **Input:** $\mathcal{O}_p^{\mathrm{sr}}(\widehat{\mathbf{M}}), \mathbf{M} \in \mathbb{S}^{d \times d}, T \subseteq [d], \phi \geq 0, q, \delta \in (0, 1)$
**2** Sample $B \subseteq T$ by including each element independently with probability $q$
**3** $G \leftarrow \emptyset$
**4** **for** $b \in B$ **do**
**5** $\quad$ **if** $\mathsf{Test}(\mathbf{M}_{:T}, b, \phi, q, \frac{\delta}{10d^2})$ **then** $G \leftarrow G \cup \{b\}$
**6** **end**
**7** **for** $j \in [d]$ **do**
**8** $\quad S_j \leftarrow$ indices in $[d]$ of observed entries in our call to $\mathcal{O}_p^{\mathrm{sr}}(\widehat{\mathbf{M}})$
**9** $\quad v_j \leftarrow \arg\min_{v \in \mathbb{R}^{|G|}} \phi\sqrt{qd}\, \|v\|_\infty + \|\mathbf{M}_{S_j \times G} v - \widehat{\mathbf{M}}_{S_j, j}\|_\infty$
**10** **end**
**11** $\widetilde{\mathbf{V}} \leftarrow$ vertical concatenation of $\{v_j^\top\}_{j \in [d]}$
**12** **return** $\widetilde{\mathbf{M}} = \mathbf{M}_{:G} \widetilde{\mathbf{V}}^\top$

---

### D.3 Fixing columns via regression

We now complete our postprocessing algorithm. Given an estimate $\mathbf{M}$ that is entrywise close to $\mathbf{M}^\star$, except for a small number of potentially large errors in each row and column, we first run Algorithm 5 to identify a set of columns containing no large errors. We then set up regression problems to fix the rest of the matrix (which may initially contain large errors on some entries).

**Lemma 33.** *Let $\mathbf{M}^\star \in \mathbb{S}^{d \times d}$ be rank-$r^\star$ and $\mu$-incoherent, and let $\widehat{\mathbf{M}} = \mathbf{M}^\star + \mathbf{N}$ for $\mathbf{N} \in \mathbb{S}^{d \times d}$ with $\|\mathbf{N}\|_\infty \leq \tau$. Let $\mathbf{M} \in \mathbb{S}^{d \times d}$ be rank-$r$ for $r \geq r^\star$. For $s, \tau > 0$, assume $[\mathbf{M} - \mathbf{M}^\star]_{:T}$ has at most $s$ entries larger than $\tau$ in each row and column. Assume*

$$s \leq \frac{d}{20\mu r^2}, \ |T| \geq \left(1 - \frac{1}{10\mu r}\right) d, \ p \geq \frac{100\mu r}{d} \log\left(\frac{d}{\delta}\right).$$

*Then with probability $1 - \delta$, if we run Algorithm 6 with parameters $q, \phi$ such that*

$$q \geq \frac{1000\mu r}{d} \log\left(\frac{d}{\delta}\right), \ q \leq \frac{1}{20\mu s r}, \ \phi = \sqrt{qd}r\tau,$$

*then the output satisfies*

$$\left\|\widetilde{\mathbf{M}} - \mathbf{M}^\star\right\|_\infty \leq 10^4 q^2 d^2 (\mu r)^3 \tau.$$

*Proof.* First, by Lemma 27 and applying our upper and lower bounds on $q$, there is a subset $T' \subseteq T$ with $|T'| \geq (1 - \frac{1}{3\mu r})d$ such that for all $j \in T'$, $\|\mathbf{M}_{:j} - \mathbf{M}_{:j}^\star\|_\infty \leq 4r\tau$. Next, by Lemma 31, with probability $1 - \frac{\delta}{10}$, we have $B \cap T' \subseteq G$. Thus, by applying Lemma 30 with $S \leftarrow T'$ and $p \leftarrow q$, we have that with $1 - \frac{\delta}{10}$ probability, for any $j \in [d]$, there is a vector $u_j$ with $\|u_j\|_2 \leq \sqrt{\frac{3\mu r}{2qd}}$ such that $\mathbf{M}_{:G}^\star u_j = \mathbf{M}_{:j}^\star$. Also, by Lemma 31, we have with probability $1 - \frac{\delta}{10}$ that

$$\|\mathbf{M}_{:G}^\star - \mathbf{M}_{:G}\|_\infty \leq 10\tau q d \mu r.$$

Now fix a $j \in [d]$ and consider the execution of the loop in Lines 7 to 10 for this index $j$. Consider $u_j$ as a candidate solution to the optimization problem. We have

$$\phi\sqrt{qd}\, \|u_j\|_\infty + \left\|\mathbf{M}_{S_j \times G} u_j - \widehat{\mathbf{M}}_{S_j, j}\right\|_\infty \leq 2\sqrt{qd}\mu r^{1.5}\tau + \left\|\mathbf{M}_{S_j \times G} u_j - \mathbf{M}_{S_j \times G}^\star u_j\right\|_\infty$$
$$+ \left\|\mathbf{M}_{S_j, j}^\star - \widehat{\mathbf{M}}_{S_j, j}\right\|_\infty$$
$$\leq 2\sqrt{qd}\mu r^{1.5}\tau + \sqrt{|G|}\, \|u_j\|_2\, 10\tau q d \mu r + \tau$$
$$\leq 100\tau q d (\mu r)^{1.5},$$

where we use that with $1 - \frac{\delta}{10}$ probability, $|G| \leq 2qd$. Thus, the solution $v_j$ in Line 9 must have

$$\left\|\mathbf{M}_{S_j \times G} v_j - \widehat{\mathbf{M}}_{S_j, j}\right\|_\infty \leq 100\tau q d (\mu r)^{1.5}, \ \|v_j\|_\infty \leq 100\mu^{1.5}\sqrt{r}. \tag{41}$$

This implies that

$$\left\|\mathbf{M}^\star_{S_j\times G}v_j - \mathbf{M}^\star_{S_j,j}\right\|_\infty \leq \left\|\mathbf{M}^\star_{S_j\times G}v_j - \mathbf{M}_{S_j\times G}v_j\right\|_\infty + \left\|\mathbf{M}_{S_j\times G}v_j - \widehat{\mathbf{M}}_{S_j,j}\right\|_\infty$$
$$+ \left\|\widehat{\mathbf{M}}_{S_j,j} - \mathbf{M}^\star_{S_j,j}\right\|_\infty$$
$$\leq |G|\,\|v_j\|_\infty\, 10\tau q d\mu r + 100\tau q d(\mu r)^{1.5} + \tau$$
$$\leq 2500 q^2 d^2 (\mu r)^{2.5}\tau.$$

Next, recall that $\mathrm{span}(\mathbf{M}^\star)$ is $\mu$-incoherent. Let $\mathbf{B} \in \mathbb{R}^{d\times r}$ have orthonormal columns with $\mathrm{span}(\mathbf{B}) = \mathrm{span}(\mathbf{M}^\star)$, with rows $\{b_i\}_{i\in[d]}$. By Lemma 30, with probability $1 - \frac{\delta}{10d^2}$, $S_j$ contains a subset $S'_j$ such that $|S'_j| \leq 2pd$ and

$$\sum_{i\in S'_j} b_i b_i^\top \succeq \frac{p}{2}\mathbf{I}_r \succeq \frac{|S'_j|}{4d}\mathbf{I}_r.$$

Thus, for all vectors $z \in \mathrm{span}(\mathbf{M}^\star)$, $\|z\|_2 \leq \sqrt{\frac{4d}{|S'_j|}}\|z_{S'_j:}\|_2$. Using $\mu$-incoherence of $\mathrm{span}(\mathbf{M}^\star)$ again,

$$\left\|\mathbf{M}^\star_{:G}v_j - \mathbf{M}^\star_{:j}\right\|_\infty \leq \left\|\mathbf{M}^\star_{:G}v_j - \mathbf{M}^\star_{:j}\right\|_2 \sqrt{\frac{\mu r}{d}}$$
$$\leq \left\|\mathbf{M}^\star_{S'_j:G}v_j - \mathbf{M}^\star_{S'_j:j}\right\|_2 \sqrt{\frac{4\mu r}{|S'_j|}}$$
$$\leq \left\|\mathbf{M}^\star_{S'_j:G}v_j - \mathbf{M}^\star_{S'_j:j}\right\|_\infty \sqrt{4\mu r}$$
$$\leq 5000 q^2 d^2 (\mu r)^3 \tau,$$

where the first line used Lemma 28. Finally, recall from (41) that for $v_j$ to be an optimal solution, we must have $\|v_j\|_\infty \leq 100\mu^{1.5}\sqrt{r}$. Since $|G| \leq 2qd$, we have the desired bound for column $j \in [d]$:

$$\left\|\mathbf{M}_{:G}v_j - \mathbf{M}^\star_{:j}\right\|_\infty \leq \left\|\mathbf{M}_{:G}v_j - \mathbf{M}^\star_{:G}v_j\right\|_\infty + \left\|\mathbf{M}^\star_{:G}v_j - \mathbf{M}^\star_{:j}\right\|_\infty$$
$$\leq |G|\,\|v_j\|_\infty\,\|\mathbf{M}_{:G} - \mathbf{M}^\star_{:G}\|_\infty + 5000 q^2 d^2 (\mu r)^3 \tau$$
$$\leq 10^4 q^2 d^2 (\mu r)^3 \tau.$$

Union bounding all of the failure probabilities then completes the proof. $\qquad\square$

**Remark 3.** *Similarly to Remark 2, we note that the proof of Lemma 33 is tolerant to an error of $\tau q d(\mu r)^{1.5}$ for the problem in Line 9, because this implies (41) holds up to a negligible constant factor, which does not affect correctness of the rest of the proof. Line 9 can again be solved to this accuracy in time $O(d \cdot \mathrm{poly}(\mu r \log(\frac{d}{\delta})))$, with failure probability $\delta$, as described in Lemma 32.*

Finally, we put everything together and give our full postprocessing procedure.

**Proposition 3.** *Let $\mathbf{M}^\star \in \mathbb{S}^{d\times d}$ be rank-$r^\star$ and $\mu$-incoherent, and let $\widehat{\mathbf{M}} = \mathbf{M}^\star + \mathbf{N}$ for $\mathbf{N} \in \mathbb{S}^{d\times d}$ with $\|\mathbf{N}\|_\infty \leq \Delta$ for some $\Delta \geq 0$. Let $\mathbf{M} \in \mathbb{S}^{d\times d}$ be given as a rank-$r$ decomposition, with $r \geq r^\star$. Further, for $\gamma \in (0,1)$ suppose $\mathbf{M}$ and $\mathbf{M}^\star$ are $d\Delta$-close on a $\gamma$-submatrix. Finally, assume*

$$\gamma \leq \frac{1}{10^4 \mu r \log(d)}.$$

*Then for any $\delta \in (0,1)$ if $p \geq \frac{1}{d} \cdot \mathrm{poly}(\mu r \log(\frac{d}{\delta}))$ for an appropriate polynomial, Algorithm 7 uses one call to $\mathcal{O}_p(\widehat{\mathbf{M}})$ and with probability $\geq 1 - \delta$, outputs $\widetilde{\mathbf{M}} \in \mathbb{S}^{d\times d}$ such that*

$$\left\|\widetilde{\mathbf{M}} - \mathbf{M}^\star\right\|_\infty \leq \mathrm{poly}\left(\mu r \log\left(\frac{d}{\delta}\right)\right)\Delta. \tag{6}$$

*Also, $\widetilde{\mathbf{M}}$ is given as a rank-$\mathrm{poly}(\mu r \log(\frac{d}{\delta}))$ factorization. The algorithm runs in $m \cdot \mathrm{poly}(\mu r \log(\frac{d}{\delta}))$ time, where $m$ is the number of observed entries upon calling $\mathcal{O}_p^{\mathrm{sr}}(\widehat{\mathbf{M}})$.*

*Proof.* Let $\tau = 1000\Delta\mu r^{1.5}\log d$. By applying Lemma 6, [43], and the assumed bound on $\gamma$, there is $A' \subseteq [d]$ with $|[d] \setminus A'| \le \frac{d}{300\mu r\log d}$ such that

$$\left\|[\mathbf{M} - \widehat{\mathbf{M}}]_{A' \times A'}\right\|_\infty \le \|[\mathbf{M} - \mathbf{M}^\star]_{A' \times A'}\|_\infty + \Delta \le \tau.$$

Now we run Algorithm 1 with

$$p \leftarrow \frac{10^8\mu^2 r^2 \log^2(\frac{d}{\delta})}{d}, \ s \leftarrow \frac{d}{10^5\mu^2 r^2 \log d}.$$

These satisfy the conditions in Lemma 13, so with probability $1 - \frac{\delta}{10}$, we obtain $S \subseteq [d]$ with $|S| \ge d(1 - \frac{1}{300\mu r})$, and $[\mathbf{M} - \widehat{\mathbf{M}}]_{S \times S}$ has $\le s$ entries with magnitude larger than $\tau$ in each row or column, implying $[\mathbf{M} - \mathbf{M}^\star]_{S \times S}$ has $\le s$ entries with magnitude larger than $2\tau$ per row or column.

Next, we run Algorithm 6 on the submatrix with rows indexed by $S$, i.e. we use the observations from $\widehat{\mathbf{M}}_{S:}$ and input $\mathbf{M}_{S:}$. The subset that is input to Algorithm 6 will be $T \leftarrow S$, and we use $q = \frac{3000\mu r}{d}\log(\frac{d}{\delta})$. Note that we have $q \le \frac{1}{20\mu s r}$. Also by construction, $|S| \ge d(1 - \frac{1}{300\mu r})$ and by Lemma 29, $\mathbf{M}_{S:}^\star$ is $3\mu$-incoherent. Thus, Lemma 33 yields, with probability $1 - \frac{\delta}{10}$, $\widetilde{\mathbf{M}}$ such that

$$\left\|[\widetilde{\mathbf{M}} - \mathbf{M}^\star]_{S:}\right\|_\infty \le 10^4 q^2 d^2 (\mu r)^3\tau \le 10^{14}(\mu r)^{6.5}\log^2\left(\frac{d}{\delta}\right)\Delta.$$

Finally, we transpose the matrix and then run Algorithm 6 again to complete the whole matrix, i.e. with inputs $\mathbf{M} \leftarrow \widetilde{\mathbf{M}}^\top$ and $T \leftarrow S$. Overall, we compute rank-poly$(\mu r \log(\frac{d}{\delta}))$ $\widetilde{\mathbf{M}}$ such that

$$\left\|\widetilde{\mathbf{M}} - \mathbf{M}^\star\right\|_\infty \le \text{poly}\left(\mu r \log\left(\frac{d}{\delta}\right)\right)\Delta.$$

Finally, by convexity of norms, we can symmetrize $\widetilde{\mathbf{M}} \leftarrow \frac{1}{2}(\widetilde{\mathbf{M}} + \widetilde{\mathbf{M}}^\top)$ and this will only improve the bound. Combining over all of the subroutines, the sample complexity and runtime are as claimed, by appropriately applying the linear program solvers as described in Lemma 32 and Remark 3. $\quad\square$

We provide pseudocode for the procedure in Proposition 3 in Algorithm 7.

---

**Algorithm 7:** Postprocess$(\mathcal{O}_{[0,1]}^{\text{sr}}(\widehat{\mathbf{M}}), \mathbf{M}, \gamma, \Delta, r, r^\star, \mu, \delta)$

---

1 **Input:** $\mathcal{O}_{[0,1]}^{\text{sr}}(\widehat{\mathbf{M}})$ where $\widehat{\mathbf{M}} = \mathbf{M}^\star + \mathbf{N} \in \mathbb{S}^{d \times d}$ where $\mathbf{M}^\star$ is rank-$r^\star$ and $\mu$-incoherent and $\|\mathbf{N}\|_\infty \le \Delta$, $\mathbf{M} \in \mathbb{S}^{d \times d}$ given as a rank-$r$ factorization such that $\mathbf{M}, \mathbf{M}^\star$ are $d\Delta$-close on a $\gamma$-submatrix, $\delta \in (0,1)$

2 $S \leftarrow \text{Sparsify}(\mathcal{O}_{[0,1]}^{\text{sr}}(\widehat{\mathbf{M}} - \mathbf{M}), 1000\Delta\mu r^{1.5}\log d, \frac{1}{300\mu r\log d}, \frac{10^8\mu^2 r^2\log^2(\frac{d}{\delta})}{d}, \frac{\delta}{10})$

3 $p \leftarrow \frac{100\mu r}{d}\log(\frac{d}{\delta}), q \leftarrow \frac{3000\mu r}{d}\log(\frac{d}{\delta}), \tau \leftarrow 1000\Delta\mu r^{1.5}\log d$

4 $\widetilde{\mathbf{M}} \leftarrow \text{Fix}(\mathcal{O}_p^{\text{sr}}(\widehat{\mathbf{M}}_{S:}), \mathbf{M}_{S:}, S, \sqrt{q d}r\tau, q, \frac{\delta}{10})$

5 $r \leftarrow \text{rank}(\widetilde{\mathbf{M}})$

6 $p \leftarrow \frac{100\mu r}{d}\log(\frac{d}{\delta}), q \leftarrow \frac{3000\mu r}{d}\log(\frac{d}{\delta}), \tau \leftarrow 10^{14}(\mu r)^{6.5}\log^2(\frac{d}{\delta})\Delta$

7 $\widetilde{\mathbf{M}} \leftarrow \text{Fix}(\mathcal{O}_p^{\text{sr}}(\widehat{\mathbf{M}}), \widetilde{\mathbf{M}}^\top, S, \sqrt{q d}r\tau, q, \frac{\delta}{10})$

8 **Return:** $\frac{1}{2}(\widetilde{\mathbf{M}} + \widetilde{\mathbf{M}}^\top)$

---

## E  Semi-random matrix completion

In this section, we finally combine the two main components of our algorithm, Algorithm 3 and Algorithm 7, to give our final result, Theorem 2. We begin with a preprocessing step to estimate an initial distance bound, which follows straightforwardly from an analogous result in [43].

**Lemma 34.** *There is an algorithm,* EstimateFrobNorm, *which takes as input* $\mathcal{O}_p^{\text{sr}}(\widehat{\mathbf{M}})$ *and parameters* $\Delta \ge 0$ *and* $\delta \in (0,1)$ *where* $\widehat{\mathbf{M}} = \mathbf{M}^\star + \mathbf{N}$ *where* $\mathbf{M}^\star \in \mathbb{S}^{d \times d}$ *is rank-$r^\star$ and $\mu$-incoherent and*

$\|\mathbf{N}\|_\infty \leq \Delta$ *and* $p \geq \frac{120\mu r^\star \log(d/\delta)}{d}$. EstimateFrobNorm *runs in time* $O(m)$ *where* $m$ *is the number of observed entries and outputs an estimate* $V \in \mathbb{R}$ *such that with probability* $\geq 1 - \delta$,

$$\|\mathbf{M}^\star\|_F \leq V \leq 10d^2(\|\mathbf{M}^\star\|_F + \Delta).$$

*Proof.* Let $\Omega \subseteq [d] \times [d]$ be the set of observed entries. Our estimator for $V$ will be

$$V = \sqrt{2d \sum_{(i,j)\in\Omega} (|\widehat{\mathbf{M}}_{ij}| + \Delta)^2}\,.$$

Now we analyze this estimator. For each row $i \in [d]$, by Lemma 30, with probability at least $1 - \frac{\delta}{d}$, if we let $S_i$ be the set of observed entries in row $i$,

$$\sum_{j\in S_i} (\mathbf{M}^\star_{ij})^2 \geq \frac{p}{2} \sum_{j\in[d]} (\mathbf{M}^\star_{ij})^2\,.$$

Summing over all rows, we have by the triangle inequality $|\mathbf{M}^\star_{ij}| \leq |\widehat{\mathbf{M}}_{ij}| + \Delta$ that

$$V^2 \geq dp\,\|\mathbf{M}^\star\|_F^2 \geq \|\mathbf{M}^\star\|_F^2$$

which proves one side of the claim. To prove the other direction, by the triangle inequality,

$$V \leq \sqrt{2d}\left(\|\widehat{\mathbf{M}}\|_F + d\Delta\right) \leq 10d^2(\|\mathbf{M}^\star\|_F + \Delta)\,.$$

$\square$

Now, we present the full algorithm for semi-random matrix completion.

---

**Algorithm 8:** SRCompletion($\mathcal{O}^{sr}_{[0,1]}(\widehat{\mathbf{M}}), \Delta, r^\star, \mu, \delta$)

---

1 **Input:** $\mathcal{O}^{sr}_{[0,1]}(\widehat{\mathbf{M}})$ where $\widehat{\mathbf{M}} = \mathbf{M}^\star + \mathbf{N} \in \mathbb{S}^{d\times d}$ where $\mathbf{M}^\star$ is rank-$r^\star$ and $\mu$-incoherent and $\|\mathbf{N}\|_\infty \leq \Delta, \delta \in (0,1)$
2  $\mathbf{M} \leftarrow \mathbf{0}_{d\times d}$
3  $p_0 \leftarrow \frac{200\mu r^\star \log(d/\delta)}{d}$
4  $D \leftarrow$ EstimateFrobNorm($\mathcal{O}^{sr}_{p_0}(\widehat{\mathbf{M}}), \frac{\delta}{2}, \Delta$)
5  $P \leftarrow \mathrm{poly}\left(r^\star, \mu, \log\left(\frac{dD}{\delta\Delta}\right)\right)$
6  $Q \leftarrow \mathrm{poly}(P)$
7  $p \leftarrow \frac{Q}{d}$
8  $K \leftarrow \lceil \log \frac{D}{\Delta} \rceil$
9  **while** $D \geq \Delta dQ$ **do**
10  $\quad$ $\mathbf{M} \leftarrow$ SRPartialCompletion $\left(\mathcal{O}^{sr}_{[0,1]}(\mathbf{M}^\star + \mathbf{N}), \mathbf{M}, D, r^\star, P, \frac{1}{10^6\mu P^2}, \frac{\delta}{4K}\right)$
11  $\quad$ $\widetilde{\mathbf{M}} \leftarrow$ Postprocess $\left(\mathcal{O}^{sr}_{[0,1]}(\mathbf{M}^\star + \mathbf{N}), \mathbf{M}, \frac{1}{10^6\mu P^2}, \frac{D}{dP}, P^{o(1)}, r^\star, \mu, \frac{\delta}{4K}\right)$
12  $\quad$ $\mathbf{M} \leftarrow$ PCA$_{r^\star}(\widetilde{\mathbf{M}})$
13  $\quad$ $D \leftarrow 0.1D$
14 **end**
15 **Return:** $\widetilde{\mathbf{M}}$

---

**Remark 4.** *In Line 12 of Algorithm 8,* PCA$_{r^\star}(\cdot)$ *denotes computing the top-$r^\star$ PCA of a matrix. We ensure that the input* $\widetilde{\mathbf{M}}$ *is a rank* $r = \mathrm{poly}(r^\star, \mu, \log(\frac{dD}{\delta\Delta}))$ *matrix given as a low-rank factorization, and then this step can be implemented in time* $d \cdot \mathrm{poly}(r)$.

Now we prove our main theorem.

**Theorem 2.** *Let* $\mathbf{M}^\star \in \mathbb{S}^{d\times d}$ *be rank-$r^\star$ and $\mu$-incoherent, let* $\mathbf{N} \in \mathbb{S}^{d\times d}$ *have* $\|\mathbf{N}\|_\infty \leq \Delta$, *and let* $\delta \in (0,1)$. *Let* $Q = \mathrm{poly}(r^\star, \mu, \log(\frac{d\|\mathbf{M}^\star\|_F}{\delta\Delta}))$ *for a sufficiently large polynomial. Algorithm 8 can*

*be implemented with one call to $\mathcal{O}_p^{\mathrm{sr}}(\mathbf{M}^\star + \mathbf{N})$ for $p = \frac{Q}{d}$ and outputs $\widetilde{\mathbf{M}} \in \mathbb{S}^{d \times d}$ with rank at most $Q$ (represented in terms of its low-rank factorization) such that with probability $\geq 1 - \delta$,*

$$\left\| \widetilde{\mathbf{M}} - \mathbf{M} \right\|_\infty \leq Q\Delta \,.$$

*The algorithm runs in time $O(mQ)$ where $m$ is the total number of observed entries.*

*Proof.* We inductively verify the conditions of Corollary 1 are satisfied whenever SRPartialCompletion is called and the conditions of Proposition 3 are satisfied whenever Postprocess is called. The base case for SRPartialCompletion is clear from the guarantees of Lemma 34. Now assuming that the hypotheses of Corollary 1 are satisfied when SRPartialCompletion is called, we know the output $\mathbf{M}$ has rank $r \leq r^\star \cdot P^{o(1)}$ and is $\frac{D}{\alpha}$-close to $\mathbf{M}^\star$ on a $\frac{1}{10^6 \mu P^2}$ submatrix. Now this implies that the hypotheses of Proposition 3 are satisfied when we call Postprocess and thus we get that

$$\left\| \widetilde{\mathbf{M}} - \mathbf{M}^\star \right\|_\infty \leq \mathrm{poly}(r^\star) P^{o(1)} \frac{D}{dP} \leq \frac{D}{100d} \,.$$

Since $\mathbf{M}^\star$ is rank-$r^\star$, we have

$$\|\widetilde{\mathbf{M}} - \mathbf{M}^\star\|_{\mathrm{F}} \leq 2\|\mathsf{PCA}_{r^\star}(\widetilde{\mathbf{M}}) - \mathbf{M}^\star\|_{\mathrm{F}} \leq 0.1D \,.$$

This now implies that the hypotheses of Corollary 1 are satisfied at the beginning of the next iteration (unless $D \leq \Delta dQ$ in which case the loop ends). By the way we set $K$, the loop ends in at most $K$ iterations and thus the overall failure probability is at most $\delta$. As long as none of the subroutines fails, the above argument implies the output must satisfy

$$\left\| \widetilde{\mathbf{M}} - \mathbf{M}^\star \right\|_\infty \leq \Delta Q$$

and that $\widetilde{\mathbf{M}}$ has rank at most $Q$. Note that by Lemma 2, we can simulate all of the observations used in the algorithm with a single sample $\mathcal{O}_p^{\mathrm{sr}}(\mathbf{M}^\star + \mathbf{N})$ with reveal probability

$$p \geq \frac{\mathrm{poly}\left( r^\star, \mu, \log\left( \frac{d\|\mathbf{M}^\star\|_F}{\delta\Delta} \right) \right)}{d} \,.$$

The overall runtime guarantee follows by combining the runtime guarantees of Corollary 1, Proposition 3, and the PCA step (see Remark 4). $\qquad\square$

