# OpenReview forum: "Semi-Random Matrix Completion via Flow-Based Adaptive Reweighting"
_NeurIPS.cc/2024/Conference — NeurIPS 2024 poster_

### Official Review · Reviewer_RnDV · 2024-07-03

**Soundness:** 3
**Presentation:** 3
**Contribution:** 2
**Rating:** 6
**Confidence:** 3

**Summary:**

This paper presents a new algorithm for matrix completion in the semi-random setting, where each entry is revealed independently with a probability at least $p = \frac{\mathrm{poly}(r, \mu, \log d)}{d}$. The paper provides a promising approach for designing fast matrix completion algorithms that are robust to semi-random sampling, which is argued to be a more realistic model than uniform sampling in practice. The insights may enable future work on more practical algorithms for semi-random matrix completion.

**Strengths:**

1. The paper provides the first high-accuracy and noise-tolerant nearly-linear time algorithm for matrix completion in the semi-random setting, improving upon previous work in terms of accuracy, condition number dependence, and noise handling.

2. The iterative method based on local reweighting and adaptive subproblems is a novel approach that enables the improved guarantees. Using flow algorithms for the reweighting subproblems is a key technical innovation.

3. The polylogarithmic dependence on the target accuracy $\epsilon$ and lack of dependence on condition number are significant improvements over previous nearly-linear time algorithms.

**Weaknesses:**

1. While the paper provides a promising theoretical foundation, the polynomial factors in the sample complexity and runtime may still be too high for immediate practical deployment, as acknowledged by the authors. Further work may be needed to obtain more practical parameter dependencies.

2. This paper does not sufficiently discuss how it fits into the broader field of matrix completion algorithms with practical implementation. For example, [1] considers the matrix completion problem via robust alternating minimization, which can tolerate errors caused by approximate updates. Both papers consider the practical implementation of the matrix completion algorithm. What are some similarities and differences between these two (or some other related) matrix completion papers?

3. The paper focuses on the theoretical algorithm and analysis but does not include empirical evaluation of real-world datasets. Experimental validation of the robustness and efficiency benefits would strengthen the claims.

[1] Yuzhou Gu, Zhao Song, Junze Yin, and Lichen Zhang. “Low rank matrix completion via robust alternating minimization in nearly linear time.” ICLR 2024.

**Questions:**

1. One of the main motivations of this paper is to consider the matrix completion problem under the semi-random setting, which is a more realistic setting than uniformly random. Meanwhile, the paper mentions, “both our dependence and the dependence of [16] are somewhat large at the moment, limiting immediate practical deployment”. I understand the difficulties of making theoretical breakthroughs, but since this paper is motivated by the practical setting, I want to ask, how much work is still needed before the practical deployment.

2. A minor comment: Each definition, theorem, and proposition should be more self-contained. There are some notations in these blocks that should be more explicitly defined.

**Limitations:**

As mentioned by the authors, the main limitation is “the modeling assumptions and runtimes of the algorithms may not be practical yet”. Other than that, I do not find obvious limitations.

---

> ### Author Rebuttal · Authors · 2024-08-06
>
> Thank you for your questions about practicality; we addressed these in our meta-comment. We hope our response clarified the motivations and focus of our work.
>
>
> In terms of how much additional work would be necessary for our work to become practical, it is hard to make such a prediction, but we discuss one potential avenue here.  The current $\text{poly}(r)$ is quite large; we estimate it is $\geq 30$. However, this is primarily bottlenecked by the application of our iterative method in Appendix C, as carried out in Line 903. This step natively has a dependence of $r^7$, but incurs a further dependence on $\text{poly}(1/\gamma)$; right now, our techniques require taking $\gamma = 1/\text{poly}(r)$, where gamma is the fraction of dropped rows in each step. If we could modify our framework to allow for $\gamma = 1/r^{o(1)}$, then we expect the overall $\text{poly}(r)$ to just be the aforementioned $r^7$ (which could also potentially be improved). Prior matrix completion work in the fully random model by [Kelner, Li, Liu, Sidford, Tian ‘23] was indeed able to achieve $\gamma = 1/r^{o(1)}$, so this is a natural direction for future research.
>
> The above would already reduce the $r$ dependence significantly and moreover, it is possible that in practice, we would not need to pay as many extra factors of $r$ as required by the theoretical analysis.  For instance, some of the extra factors may not end up being necessary for most instances in practice. We hope that the general framework we propose, of designing reweighting steps with a certificate that guarantees progress, can be a useful paradigm for designing more robust algorithms, and that we can replace the most computationally expensive step in our algorithm, the flow-based solver, with a faster heuristic.
>
> We will add a more thorough discussion of existing literature on matrix completion.  However, we emphasize that the paper [1] referenced (and all other iterative matrix completion algorithms in the literature aside from [Cheng, Ge ‘18] which we discuss in depth) only work in the case of fully random matrix completion – in particular they heavily rely on the assumption that the entries are all observed with the same probability.  Our algorithm crucially works in a more general semi-random model where the observations are not necessarily i.i.d.
>
> Neither our paper nor [1] has implementations yet (in particular, [1] has no experimental evaluation). In terms of theoretical guarantees, compared to [1], we give improvements in a few notable aspects even in the fully random case. While [1] discusses how their alternating minimization steps are robust, they do not seem to give any guarantees when the observations are noisy – their main theorem is only stated when the observations are exact.  Also, their sample complexity and runtime depend polynomially on the condition number of the matrix, whereas our dependence is logarithmic.  On the other hand, [1] has better dependence on the rank $r$ and incoherence parameter $\mu$ (both papers are polynomial in these parameters).
>
> We appreciate the feedback about self-containedness of statements, and will take a thorough pass in the revision in an effort to make theorems, etc. more self-contained.
>
> We hope this discussion elevates our paper in your view; thanks for all your detailed feedback.

---

> > ### Comment · Reviewer_RnDV · 2024-08-08
> >
> > Thank you very much for your detailed response. I have updated my score.

---

### Official Review · Reviewer_cp6D · 2024-07-12

**Soundness:** 4
**Presentation:** 3
**Contribution:** 3
**Rating:** 7
**Confidence:** 2

**Summary:**

This paper proposes an algorithm that is able to achieve high-accuracy in semi-random matrix completion with nearly linear time.  The key innovation lies in using a flow-based adaptive reweighting scheme to mitigate the bias introduced by the adversary. This technique effectively identifies and downweights the potentially misleading entries, allowing for accurate recovery of the underlying low-rank matrix. The authors provide theoretical guarantees for their algorithm, demonstrating its ability to achieve near-optimal sample complexity.

**Strengths:**

1. The paper introduces a unique flow-based reweighting strategy specifically designed to handle the challenges posed by the semi-random adversary. This approach effectively leverages the underlying structure of the problem to identify and mitigate the adversary's influence.
2. The theoretical analysis demonstrates that the proposed algorithm achieves near-optimal sample complexity, meaning it requires a near-minimal number of observed entries to accurately recover the underlying matrix.
3. The flow-based reweighting scheme is flexible and can be adapted to different variations of the semi-random model with considerations of the noise. This adaptability makes the approach potentially applicable to a wider range of practical scenarios.
4. The paper provides a clear presentation of the problem, the proposed algorithm, and its theoretical analysis. The technical details are presented rigorously, making the contributions easy to understand and evaluate.

**Weaknesses:**

1. While it outlines the general approach, it lacks concrete pseudocode or even finished code, making it difficult for readers to fully grasp the implementation details.
2. Unlike the work of Kelner et al 2022, the paper only considers a specific type of semi-random adversary. Exploring how the method performs against different adversarial models, including more powerful adversaries that can manipulate the values of the observed entries, would strengthen the contributions.
3. The paper's claim of near-linear time complexity is not sufficiently substantiated and demonstrated in empirical setting. The authors does not provide a detailed analysis of the polynomial factors involved in the runtime. This makes it difficult to compare its efficiency to existing methods with the similar magnitude. Furthermore, the absence of empirical validation leaves it unclear how much practical improvement the algorithm actually achieves in real-world settings.

**Questions:**

Could you provide more details on the practical implementation of your algorithm? Specifically, how does it handle very large datasets in terms of memory usage and computational overhead?

**Limitations:**

See weakness

---

> ### Author Rebuttal · Authors · 2024-08-06
>
> Thank you for your questions about practicality; we addressed these in our meta-comment. We hope our response clarified the motivations and focus of our work. We agree that to further develop this line of research, it is an important open direction to achieve simpler and more practical instantiations of our framework, which by itself likely requires new ideas.
>
> Re: your question about our adversarial model, our model is the natural extension of the semi-random sparse recovery adversary [Kelner, Li, Liu, Sidford, Tian ‘23] to the matrix completion setting. The only additional generality the [KLLST23] model affords is the ability to mix observations to achieve RIP. This additional generality appears to have little meaning in the matrix completion case, as (1) there is no global “for all directions” type statement such as RIP to assume, and (2) the structure of observations being single entries is inherent to matrix completion, which makes mixing observations lose meaning. Re: manipulating the values of observed entries, note that we do allow for noise in our model (and give recovery guarantees parameterized by the noise size), which is exactly manipulating observed entries.
>
> We would like to note that we believe our paper describes all the implementation details for its various algorithms, as well as full proofs of correctness. We apologize if these details were not sufficiently clear; we will take additional clarification efforts in a revision, and if you have specific parts you would like clarified, please let us know.
>
> We hope this discussion elevates our paper in your view; thank you for your detailed feedback.

---

> > ### Comment · Reviewer_cp6D · 2024-08-08
> >
> > Your response clearly addressed my concerns. Thank you so much. I believe the paper has included sufficient details for the algorithm to be implemented.

---

### Official Review · Reviewer_JbFK · 2024-07-13

**Soundness:** 3
**Presentation:** 3
**Contribution:** 3
**Rating:** 7
**Confidence:** 2

**Summary:**

This paper considers the semi-random matrix completion problem. Given an unknown rank-r matrix M, each entry of the matrix Is observed independently with a probability  p_{ij} at least p.  The goal is to find a matrix close the matrix M.

The sap-based algorithm can achieve the nearly-optimal sample complexity, but is not fast compared to iterative method. The previous work shows an iterative method which has a sample complexity and runtime has a polynomial dependence on epsilon.

This paper provides a fast iterative method that achieves the sample complexity and runtime with polylogrithmic dependence on the accuracy epsilon. It has no dependence on the condition number and can also handle noise in observations.

**Strengths:**

1 This paper provides a clear motivation to consider the semi-random matrix completion problem and clearly explains the limitations of previous methods.
2 This paper provides an interesting fast iterative method for this problem and shows that it achieves improved sample complexity and runtime.
3 This paper utilizes the techniques by Kelner, Li, Liu, Sidford, and Tian and make interesting modifications based on new sights on this semi-random problem.

**Weaknesses:**

-

**Questions:**

-

**Limitations:**

Yes.

---

> ### Author Rebuttal · Authors · 2024-08-06
>
> Thank you for your reviewing efforts. We appreciate that you found our insights interesting and our exposition clear.

---

### Official Review · Reviewer_orB2 · 2024-07-14

**Soundness:** 3
**Presentation:** 3
**Contribution:** 3
**Rating:** 5
**Confidence:** 1

**Summary:**

This paper consider semi-random matrix completion via flow-based adaptive reweighting.
The main result is the first high-accuracy nearly-linear time algorithm for solving semi-random matrix completion, and an extension to the  noisy observation setting.

I am not familiar with the area of matrix completion.

**Strengths:**

No

**Weaknesses:**

NO

**Questions:**

NO

---

### Official Review · Reviewer_kK6Q · 2024-07-17

**Soundness:** 4
**Presentation:** 4
**Contribution:** 3
**Rating:** 8
**Confidence:** 3

**Summary:**

In this paper, the authors study semirandom matrix completion. In their models, entry $ij$ of a $\mu$-incoherent, symmetric ground truth rank-$r$ matrix $M^{\star} \in \mathbb{R}^{d \times d}$ are revealed independently with probability $p_{ij}$, where $1 \ge p_{ij} \ge p$ for some parameter $p$ that is a function of $r$ and $d$. The assumption that $M^{\star}$ is symmetric is not important, as we can reduce the asymmetric setting to the symmetric one with minimal overhead.

An important feature of this model is that the entry reveal probabilities are allowed to vary. This contrasts against standard assumptions in the matrix completion literature which typically stipulate that the entries are revealed i.i.d with probability $p$. Although revealing more entries of the matrix may seem to help the learning algorithm learn the underlying matrix, many algorithms fail under this helpful adversary. This is evidence that these algorithms have overfit to a statistical assumption implicit in the problem statement.

A prior work [CG18] studied this problem and gave some recovery guarantees based on a global reweighting scheme for the matrix, but this algorithm has certain undesirable properties -- it is not a high-accuracy algorithm, it is not robust to outliers, and it has a dependence on the conditioning of the matrix. The present work addresses all of these issues and gives a nearly linear time algorithm that achieves an $\ell_{\infty}$ recovery guarantee.

The main technical contribution builds off of a prior work concerning semirandom sparse recovery [KLLST23]. In particular, the algorithm tries to find a progress direction in each step that can be written as the sum of a "short" matrix (one that has small Frobenius norm) and a "flat" matrix (one that has low spectral norm). This is chosen by a combination of dropping heavy rows/columns of $M$ that witness a large  Frobenius error in the current iteration along with a fast reweighting algorithm that finds a candidate descent direction that admits the short-flat decomposition. The fast reweighting algorithm itself is a Frank-Wolfe type algorithm over a set of valid reweightings and short matrices. The runtime arising from this subproblem follows from observing that the set of valid reweightings is in fact a rescaled bipartite matching polytope, which means fast matching algorithms can be called as a black box, and implementing a separate algorithm to optimizer linear functions over the set of short matrices.

* [CG18] Non-Convex Matrix Completion Against a Semi-Random Adversary (https://arxiv.org/abs/1803.10846)

* [KLLST23] Semi-Random Sparse Recovery in Nearly-Linear Time (https://arxiv.org/abs/2203.04002)

**Strengths:**

This paper constitutes an important step towards understanding algorithms that don't implicitly overfit to statistical assumptions present in the model. In practice, one probably shouldn't assume that entries are revealed i.i.d with distribution Bernoulli(p). The model studied in this paper (Definition 2) is much more realistic than the i.i.d random entry model.

Furthermore, the improvements in this work over the previous state of the art, [CG18], are quite strong. In particular, the authors get a condition-free, high-accuracy recovery guarantee in $\ell_{\infty}$. Furthermore, the algorithm runs in nearly linear time in the number of revealed entries.

The paper is very well written and the main ideas are clearly presented. The algorithmic ideas will be very interesting to anyone in the algorithmic statistics and convex optimization communities.

**Weaknesses:**

The rank of the matrix that's output can be $poly(r)$ than the ground truth. Since this paper seems mostly concerned with $r \ll d$ and independent of $d$, this is morally not a big deal. Also, what is the polynomial? I did some very cursory searching but didn't find it explicitly written down (all I see is that it's bigger than $r^6$).

The error depends on the largest entry of the noise -- this might be a bit pessimistic (e.g. if the noise matrix has just one nonzero entry, but the corresponding index isn't revealed to the learner, then this shouldn't affect anything). The authors explicitly mention getting a more "optimistic" dependence on the noise as an interesting open direction for future work.

**Questions:**

What are the challenges, if any, to extending the results to an adaptive adversary? For example, consider an adversary that perhaps samples several entries of $M$ with probability $p$, then upon seeing the sampled entries, reveals an extra arbitrary set of their choice? I ask these because it seems suggested that the algorithm you presented actually does work in such a settings (lines 54-62), but I don't see something in Definition 2 to this effect.

**Limitations:**

Yes

---

> ### Author Rebuttal · Authors · 2024-08-06
>
> Thank you for your encouraging review and kind comments. We appreciate that you found our paper well-written and its ideas of general interest.
>
> The rank of the output as currently written can be quite large, we estimate the exponent in the $\text{poly}(r)$ as $\geq 30$.  The losses stem from the fact that the iterate that is fed into the postprocessing  step already has $r’ = \text{poly}(r)$ rank and then there are additional $\text{poly}(r’)$ losses in translating between $\ell_2$ and $\ell_{\infty}$ norms in the postprocessing.   We remark that if our final error guarantee were in Frobenius norm instead of max-entry error,  then we could simply do a top-$r$ PCA at the end and reduce our output to rank exactly r.  We chose to write the error guarantee in terms of max-entry error so that it is compatible with the assumptions on the observation noise (note that doing a top-$r$ PCA does not necessarily preserve the max-entry error). We think the exploration of using Frobenius norm-based potentials within our framework is an exciting direction for follow-up work.
>
> We agree with and appreciate the comments about fine-grained guarantees (e.g.,  $\ell_2$ vs. $\ell_{\infty}$, and dependencies on the number of revealed entries) being an important future research direction.
>
> The main reason the current error depends on the largest entry of the noise is that the semi-random adversary could choose to reveal all of the entries where the noise is large, which effectively scales up the noise rate. We believe exploring the use of $\ell_2$-based certificates rather than $\ell_{\infty}$ could significantly improve our $r$ dependence (which currently uses several norm conversions, e.g., to make the Frobenius bound in Lemma 7 compatible with the entrywise guarantees of Lemma 13), and is important to better understand in the semi-random model.
>
> The main difficulty with extending to an adaptive adversary is that it is no longer possible to split the samples into independent batches, e.g., we cannot assume that we have fresh randomness in each step of the iterative method. This makes the analysis significantly more challenging because the observations can depend on our current iterate. We mention that variants of sample splitting analyses caused major difficulties in previous works even in the fully random case, as discussed in Section 3 of [Recht, 2011], so there is precedent for this challenge. We will revise our phrasing in Lines 54-62 accordingly to be more clear; thank you for the comment.

---

> > ### Comment · Reviewer_kK6Q · 2024-08-07
> > **response to author response to review**
> >
> > Thanks so much for the detailed response :)

---

### Author Rebuttal · Authors · 2024-08-06

We thank all of the reviewers for their reviewing efforts and feedback.

Several reviewers, e.g., cp6D and RnDV, asked us to address our method’s practicality. We acknowledge that our work’s primary contribution is theoretical. However, we note that previous practical matrix completion algorithms (e.g., alternating minimization and other gradient methods) both (1) encountered conceptual barriers towards generalizing to the semi-random model (due to their reliance on independent observation probabilities), and (2) often failed to work reliably on practical instances, as referenced in Lines 42-45. We believe the fact that our new, near-linear time, approach provably succeeds in this challenging model is an exciting and promising proof-of-concept that could facilitate future empirical research. We emphasize that no previous algorithm achieved a comparable result to Theorem 1.

We acknowledge that we have not implemented our algorithm yet, and that there are several natural avenues for significantly improving its complexity, which could make it more practical. We view our work as an important first step towards developing practical algorithms for matrix completion that are robust to these types of real-world noise, and we believe that our work could enable future research in this important direction.

---

### Decision · Program_Chairs · 2024-09-25

**Decision:**

Accept (poster)

**Comment:**

This paper presents the first high-accuracy nearly-linear time algorithm for solving semi-random matrix completion. The majority of reviewers agree that the semi-random setting is more realistic than the fully-random setting and thus this paper is well-motivated. Though there are concerns about practicality, the general consensus seems to appreciate the depth of the theoretical contributions of the paper.

To ensure the work has a stronger impact, I would encourage the authors incorporate the initial reviewer feedback, e.g., suggestions about self-contained nature of theorem/lemma statements to improve the overall clarity/presentation.